# Modelling glacier mass-balance and climate sensitivity in a context of sparse observations: application to Saskatchewan Glacier, western Canada

Christophe Kinnard[1], Olivier Larouche[1,2], Michael N. Demuth[3], Brian Menounos[4]

[1] Centre de Recherche sur les Interactions Bassins Versants—Écosystèmes Aquatiques (RIVE), Département des Sciences de l'Environnement, Université du Québec à Trois-Rivières, Québec, G8Z 4M3, Canada
[2] Centre d'Études Nordiques (CEN), Québec, G1V 0A6, Canada
[3] Geological Survey of Canada; currently, University of Saskatchewan Cold Water Laboratory, Canmore, T1W 3G1, Canada
[4] Natural Resources and Environmental Studies Institute and Geography, University of Northern British Columbia, Prince George, British Columbia, Canada

*Correspondence to*: Christophe Kinnard (christophe.kinnard@uqtr.ca)

**Abstract.** Glacier mass balance models are needed at sites with scarce long-term observations to reconstruct past glacier mass balance and assess its sensitivity to future climate change. In this study, North American Regional Reanalysis (NARR) data were used to force a physically-based, distributed glacier mass balance model of Saskatchewan Glacier for the historical period 1979-2016 and assess it sensitivity to climate change. A two-year record (2014-2016) from an on-glacier automatic weather station (AWS) and historical precipitation records from nearby permanent weather stations were used to downscale air temperature, relative humidity, wind speed, incoming solar radiation and precipitation from NARR to the station sites. The model was run with fixed (1979, 2010) and time-varying (dynamic) geometry using a multi-temporal digital elevation model dataset. The model showed a good performance against recent (2012-2016) direct glaciological mass balance observations as well as with cumulative geodetic mass balance estimates. The simulated mass balance was not very sensitive to the NARR spatial interpolation method, as long as station data was used for bias correction. The simulated mass balance was however sensitive to the biases in NARR precipitation and air temperature, as well as to the prescribed precipitation lapse rate and ice aerodynamic roughness lengths, showing the importance of constraining these two parameters with ancillary data. The glacier-wide simulated energy balance regime showed a large contribution (57%) of turbulent (sensible and latent) heat fluxes to melting in summer, higher than typical mid-latitude glaciers in continental climates, which reflects the local humid 'icefield weather' of the Columbia Icefield. The static mass balance sensitivity to climate was assessed for prescribed changes in regional mean air temperature between 0 to 7 °C and precipitation between -20 to +20%, which comprise the spread of ensemble RCP climate scenarios for the mid (2041-2070) and late (2071-2100) 21st century. The climate sensitivity experiments showed that future changes in precipitation would have a small impact on glacier mass-balance, while the temperature sensitivity increases with warming, from -0.65 to -0.93 m w.e. a$^{-1}$ °C$^{-1}$. The mass balance response to warming was driven by a positive albedo feedback (44%), followed by direct atmospheric warming impacts (24%), a positive air humidity feedback (22%) and a positive precipitation phase feedback (10%). Our study underlines the key role of albedo and

air humidity in modulating the response of winter-accumulation type mountain glaciers and upland icefield-outlet glacier settings to climate.

## 1 Introduction

Global warming is expected to cause reduced precipitation as snowfall in cold regions, earlier snowmelt in spring and a longer ice melt period in summer (e.g., Barnett et al., 2005; Aygün et al. 2020a). Even if precipitation remains unchanged, warming alone will reduce snow and ice storage in catchments, affecting the seasonality of river streamflow regimes and accelerating water losses to the ocean (Escanilla-Minchel et al., 2020; Huss et al., 2017; Huss and Hock, 2018). The transition from a nivo-glacial to a more pluvial river regime in response to warming will change the timing and magnitude of floods, leading to altered

patterns of erosion and sediment deposition and impacting biodiversity and water quality downstream (Déry et al., 2009; Huss et al., 2017). The impacts of the progressive loss of ice and snow surfaces and resulting alterations of the hydrological cycle can reach well beyond the glacierized catchments, affecting agriculture (Barnett et al., 2005; Comeau et al., 2009; Milner et al., 2017; Schindler and Donahue, 2006), fisheries (Dittmer, 2013; Grah and Beaulieu, 2013; Huss et al., 2017), hydropower and general ecological integrity (Huss et al., 2017).

The surface mass balance is the prime variable of interest to monitor and project the state of glaciers and their hydrological contribution under global warming scenarios (Hock and Huss, 2021). However, only a few glaciers around the world have long term direct mass balance observations, because these measurements are time consuming and logistically complicated. For example, only 30 glaciers have uninterrupted mass balance records since 1976 (Zemp et al., 2009). Geodetic estimates provide

a complementary picture of cumulative mass changes for a greater number of glaciers worldwide, but their coarser sampling interval (typically > 5years) makes their link with climate less direct (Cogley, 2009; Cogley and Adams, 1998; Menounos et al., 2019). For this reason, models are often used to extrapolate scarce measurements, estimate unsampled glaciers and to assess glacier mass balance sensitivity to climate. Temperature-index models, which use air temperature as sole predictor of ablation (Hock, 2003), have been extensively used to project regional and global glacier mass balance under climate change

scenarios, due to their simple implementation and readily available global precipitation and temperature forcing data (Hock et al., 2019; Huss and Hock, 2015; Marzeion et al., 2012; Radić et al., 2014). Enhanced temperature-index models, which include additional predictors such as potential (Hock, 1999) or net (Pellicciotti et al., 2005) solar radiation, have also been shown to improve glacier melt simulation and to be more transferable outside their calibration interval (Gabbi et al., 2014; Réveillet et al., 2017). These empirical models contain few parameters which simplifies their application, but they must be calibrated on

observations, which makes model extrapolation in a different climate questionable (Carenzo et al., 2009; Gabbi et al., 2014; Hock et al., 2007; Wheler, 2009). Hence, spatially-distributed, energy balance models that better represent the physical processes driving glacier ablation are more suited to simulate glacier mass balance outside of present-day climate conditions (Hock et al., 2007; MacDougall and Flowers, 2011), given that accurate forcing data is available (Réveillet et al., 2018).

Energy-balance glacier models require several input observations and contain multiple parameters that are sometimes difficult to measure or estimate (e.g. Anderson et al., 2010; Anslow et al., 2008; Arnold et al., 1996; Ayala et al., 2017; Gerbaux et al., 2005; Hock and Holmgren, 2005; Klok and Oerlemans, 2002; Marshall, 2014; Mölg et al., 2008). Glacier mass balance models have been mostly forced with observations from automatic weather stations (AWS) on, or near glaciers. However, the management of weather stations networks in mountainous areas poses financial and logistical challenges. At sites with scarce

or missing data, outputs from meteorological forecasting models (Bonekamp et al., 2019; Mölg et al., 2012; Radic et al., 2018), regional climate models (Machguth et al., 2009; Paul and Kotlarski, 2010) and reanalysis data (Clarke et al., 2015; Hofer et al., 2010; Østby et al., 2017; Radić and Hock, 2006) have been used to force glacier models. In particular, climate reanalyses provide consistent and readily available gridded estimates of past atmospheric states at sub-daily intervals, which constitute a useful alternative to drive glaciological and hydrological models in data-scarce regions (Hofer et al., 2010). Reanalyses are

produced by retrospective numerical weather model simulations that assimilate long-term and quality-controlled observations. Regional products like the North American Regional Reanalysis (NARR) have been developed to enhance the spatial and temporal resolution of reanalyses at the continental scale (Mesinger et al., 2006). Statistical downscaling of reanalysis data using on- or near-glacier meteorological observations is necessary to reduce biases resulting from this temporal and spatial scale mismatch as well as from structural and parameterizations errors in the reanalysis model (Hofer et al., 2010). Several

methods can be used to correct those errors, such as a simple bias shift toward observations (scaling or delta method) or the matching of two probability distributions (e.g., quantile mapping) (Radić and Hock, 2006; Rye et al., 2010; Teutschbein and Seibert, 2012). This step is crucial, as uncertainties in climate forcing can be the main source of error in mass balance modelling (Østby et al., 2017).

Forcing physically-based glacier models with global or regional gridded climate data introduces additional uncertainties which add up to the structural and parameter uncertainties of the glacier model. In a context of sparse in situ observations, the combination of poorly constrained model parameters, biases in meteorological forcings and limited validation data can result in biased long-term mass balance reconstructions and an incorrect appraisal of glacier-climate relationships (Anslow et al., 2008; Machguth et al., 2008; Zolles et al., 2019). A careful application, validation and sensitivity analysis of the model becomes

crucial in these situations. Paradoxically, glaciers with sparse or no observations are typically those where longer-term model reconstructions of mass balance are often most sought (e.g. Kinnard et al., 2020; Kronenberg et al., 2016; Sunako et al., 2019). Saskatchewan Glacier (52.15 °N, -117.29 °E), one of the main outlet glaciers of the Columbia Icefield in the Canadian Rocky Mountains, is such a glacier with sparse mass balance observations, which challenges the application of physically-based mass balance models. The Canadian Rocky Mountains support many glaciers which provide several ecosystem services, such as

water provision for hydropower production and agriculture, and constitute iconic features highly valorized for tourism (Anderson and Radić, 2020; Comeau et al., 2009; Moore et al., 2009; Petts et al., 2006; Schindler and Donahue, 2006). However, only a few glaciers have been directly and continuously monitored for mass balance. Peyto Glacier (51.67 °N, -116.53 °E) is the only reference site with a long mass balance record (since 1966), and with the exception of 1996 and 2000,

exhibits a consistent trend of negative annual balance beginning since the mid-1970's (Demuth, 2018; Demuth et al., 2006;
Demuth and Pietroniro, 2002). Menounos et al. (2019) recently used multisensor digital elevation models from spaceborne
optical imagery to calculate a mean mass balance of -0.410 ± 0.213 m w.e. a$^{-1}$ for the 2000-2018 period in the Canadian Rocky
Mountains, with accelerated mass loss between 2000-2009 and 2009-2018. A large-scale modelling study by Clarke et al.
(2015) showed that the volume of western Canada's glaciers could decrease by more than 90% from 2005 to 2100 in the
Canadian Rockies. Clarke et al. (2015) concluded that the main source of uncertainty in their simulations of glacier evolution
at the mountain range scale was not the parameterisation of glacier flow but rather the simulation of surface mass balance.
Thus, accurate models of surface mass balance are still needed at the scale of individual glaciers, to extend, and give context
to, sparse mass balance observations as well as to characterize the mass balance sensitivity to climate change.

Well-validated glacier models are an ideal tool to estimate glacier climate sensitivity, i.e., the mass balance response to a
change in climate conditions (Braithwaite and Raper, 2002; Che et al., 2019; Ebrahimi and Marshall, 2016; Engelhardt et al.,
2015; Gerbaux et al., 2005; Hock et al., 2007; Klok and Oerlemans, 2004; Mölg et al., 2008; Oerlemans et al., 1998; Yang et
al., 2013). These and other studies have reported on the varying sensitivity of mass-balance to warming air temperatures,
however often without unraveling the respective contributions of atmospheric warming, surface feedbacks and precipitation
phase feedbacks on the temperature sensitivity. Distributed energy-balance models offer the ability to resolve the changes in
energy fluxes that underpin the sensitivity of mass balance to warming air temperatures, shedding light on the driving processes
of ablation under a changing climate (e.g. Anderson et al., 2010; Rupper and Roe, 2008).

This study uses a physically-based, distributed mass balance model in the context of sparse observations to reconstruct long-
term glacier mass changes and spatiotemporal patterns of energy and mass fluxes, and to investigate the glacier mass balance
sensitivity to climate change. The main issues addressed in this study are (i) how to constrain a physically-based mass balance
model forced by reanalysis data in a context of sparse observations; and (ii) quantify the respective contributions of energy
balance, precipitation phase and air humidity feedbacks to the mass balance climate sensitivity under various warming
scenarios.

## 2 Study area

The Columbia Icefield is located in the Canadian Rocky Mountains and straddles the border between Alberta and British
Columbia (Figure 1a). The Columbia Icefield is accessible via the Icefields Parkway which is surrounded by two national
parks (Jasper, and Banff), which makes the Columbia Icefield a highly-valued cultural and touristic site (Sandford, 2016).

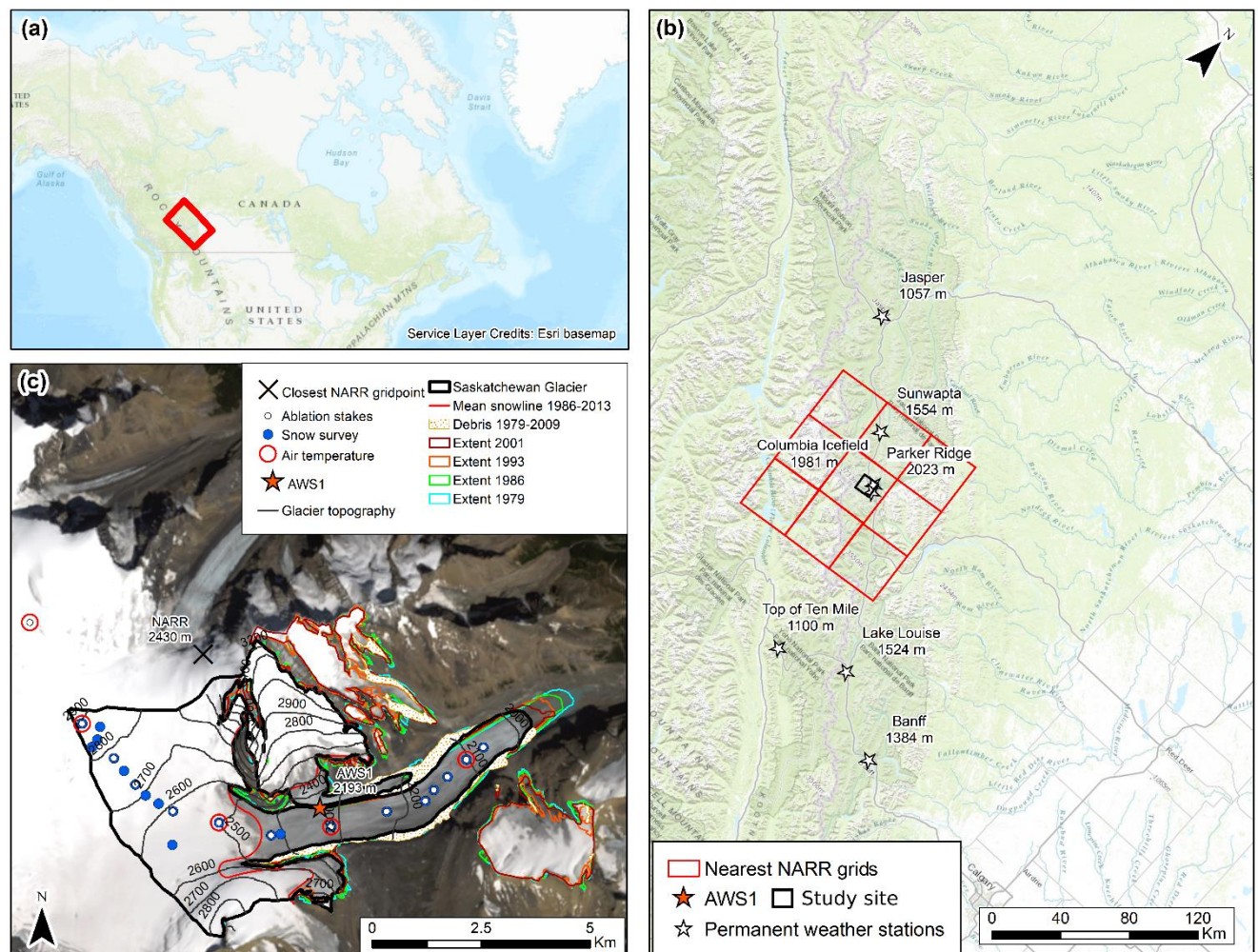

**Figure 1. Study area map. (a) Location of the Columbia Icefield in the Canadian Rockies; the red rectangle shows the area of panel b. (b) Weather stations from the permanent network used to calculate temperature and precipitation lapse rate. The nine NARR gridcells closest to Saskatchewan Glacier are shown as red squares. (c) Map of Saskatchewan Glacier showing the location of ablation stakes and additional snow survey points, and air temperature sensors used to determine the diurnal lapse rate over the glacier. The mean end of summer snow line position (1986-2013) is shown by a red line. A Landsat 8 scene from 22 August 2013 is used for map background.**

The plateau lying at ~2800 meters above sea level (m a.s.l) intercepts moist air masses originating from the Pacific Ocean, which results in large snow accumulation and the formation of glacial ice flowing downward through several outlet glaciers (Demuth and Horne, 2018). The Columbia Icefield is of crucial importance to the region's water budget, as it feeds three different continent-scale watersheds flowing towards the Arctic, Pacific and Atlantic oceans (Figure 1a). The main and largest outlet glaciers are located east of the icefield (Saskatchewan and Athabasca Glacier), draining ~60% of the eastern Columbia Icefield to the North Saskatchewan River (Hudson/Atlantic) and the Sunwapta-Athabasca River (Arctic) (Marshall et al.,

2011). Tennant and Menounos (2013) used historical aerial photographs and satellite images to reconstruct the extent and volume changes of the Columbia Icefield. The area of the Columbia Icefield was estimated to be 265.1 ± 12.3 km² in 1919. By 2009 the icefield had declined by 59.6 ± 1.2 km² (-22 ± 0.5 %). Saskatchewan Glacier is the largest outlet glacier of the icefield and the source of the North Saskatchewan River; its area was 23 km² in 2017 with elevations ranging from 1784 to 3322 m, the summit of Mount Snow Dome - the hydrological apex of western Canada (Ednie et al., 2017). Saskatchewan Glacier experienced the greatest absolute area loss among the icefield glaciers, at -10.1 ± 0.6 km² since 1919 (Tennant and Menounos, 2013). At the catchment scale, Demuth et al. (2008) reported glacier area-wise losses of -22% for the whole North Saskatchewan River Basins between 1978 and 1998.

## 3 Data and Methods

### 3.1 Topographic data

The main topographic data used in this study is a 1-meter resolution digital elevation model (DEM) derived from two WorldView-2 (WV2) satellite stereo images acquired on 31 July 2010, covering the lower glacier, and 18 September 2010, covering the upper glacier. The DEM was mosaiced with tiles from the Canadian Digital Surface Model (CDSM) (20-meter resolution) to include all adjacent topography that could cast shadows on the glacier. The merged DEM was resampled to 100 meters resolution to allow faster calculation with the mass balance model. The firn area was delimited by a mean snowline delineated from Landsat satellite images from year 1986 to 2013 (Figure 1c). Eighteen cloud-free images were chosen near the end of the hydrological season (September 30) and used to map the mean transient snowline position at the end of summer, which was used as a proxy for the equilibrium line altitude (ELA). Image dates ranged between August 22 and October 2, necessary to find cloud-free images capturing the transient snow line near the end of the ablation period.

To take into account historical glacier contraction in mass balance simulations, multi-temporal DEMs and glacier boundaries from Tennant and Menounos (2013) (hereafter 'TM2013') were used to update the glacier geometry over time in the mass balance model. TM2013 derived DEMs and glacier extents from aerial stereo photographs from 1979, 1986 and 1993. For 1999, they used the Shuttle Radar Topography Mission (SRTM) DEM of February 2000, which they attributed to best represent the glacier surface at the end of the 1999 summer ablation season, due to the penetration of the radar wave in the following year's winter snowpack. The glacier extent in 1999 was derived from the closest cloud-free, 30 m resolution Landsat 5 Thematic Mapper (TM) image in September 2001. The 2009 DEM and glacier extent from TM2013 were derived from Satellite Pour l'Observation de la Terre 5 (SPOT 5) stereo images with a resolution of 2.5 m. Points matched on stereoscopic image pairs were gridded to a 100 m resolution in the ablation area and to 200 m in the accumulation area where low contrasts resulted in a smaller number of elevation points, and varying amounts of data gaps. We re-interpolated all TM2013 DEMs to continuous 100 m resolution using shape-preserving linear interpolation. The 2010 WV2 DEM was used instead of the 2009 DEM from

TM2013, which particularly suffered from extensive gaps in the accumulation zone, but the glacier extent of August 2009 was conserved as boundary for the 2010 WV2 DEM. The slope, aspect and sky-view factors were derived from all DEMs to be used as inputs for the mass balance model. A more recent, 2-m resolution DEM was built from a stereo pair of Pleiades Satellite panchromatic images acquired in September 2016 and using the NASA Ames Stereo Pipeline (ASP) (Shean et al., 2016). This DEM was used to update the geodetic mass balance from TM2013 (Supplementary Material). Since the 2010 WV2 DEM has the highest resolution and few gaps, it was considered the most reliable and used for model calibration and climate sensitivity experiments.

Two static balance simulations were performed, one using the 1979 DEM as initial boundary condition, and the other with the 2010 DEM. These were compared with a dynamical simulation in which the glacier geometry was adjusted with the multitemporal DEMs, to consider the impact of glacier recession on mass balance. The TM2013 glacier boundaries were used but two ice masses, disconnected from Saskatchewan Glacier since 1979, were excluded from the original TM2013 outlines (see Figure 1c). The lateral, debris-covered moraines were also excluded from the glacier outlines (see Figure 1c). The term 'reference mass balance' ($B_{a\_r}$) is used hereafter to refer to glacier-wide mass balance simulated with a fixed reference geometry while the term 'conventional mass balance' ($B_{a\_c}$) is used for the simulation with adjusted glacier geometries (Huss et al., 2012). The effect of dynamical adjustment on $B_{a\_c}$ was obtained by subtracting the reference balance using the 1979 geometry ($B_{a\_r1979}$) from $B_{a\_c}$.

## 3.2 Meteorological data

### 3.2.1 On-glacier automatic weather station

An automatic weather station (AWS) was deployed in August 2014 on the medial moraine of Saskatchewan Glacier at an elevation of 2193 m a.s.l., collecting near-continuous hourly data for a two-year period, until June 2016 (Figure 1c). Recorded variables include air temperature ($T_a$), relative humidity ($RH$), incoming global ($G$) and reflected ($SW\uparrow$) solar radiation, wind speed ($WS$) and direction ($WD$), and snow depths from an ultrasonic sensor. $HOBO^{TM}$ air temperature sensors were installed by the Geological Survey of Canada (GSC) on five ablation stakes (Figure 1c) and operated between May to August 2015. The $HOBO$ sensors were shielded from solar radiation using naturally ventilated gill shields.

### 3.2.2 Meteorological data from permanent weather monitoring network

Seven weather stations were chosen from the permanent weather monitoring network maintained by Environment and Climate Change Canada, in order to calculate temperature and precipitation lapse rates. The stations ranged in elevation from 1050 to 2025 m a.s.l. (Figure 1b). As precipitation was not measured at the AWS site, a historical precipitation record was produced using data from the two weather stations closest to Saskatchewan Glacier and highest in elevation (Parker Ridge, 2023 m a.s.l. and Columbia Icefield, 1981 m a.s.l., see Figure 1b). The Columbia Icefield station was only operated between May and

November while Parker Ridge was operated mostly in winter and sometimes all year-round depending on road accessibility. Both discontinuous records were merged by averaging them.

### 3.2.3 Reanalysis data

While the precision of the on-glacier AWS data is useful to characterize the glacier microclimate, the short and discontinuous record is not adequate to drive a physically-based, distributed glacier mass balance model for periods of a decade or more. Meteorological reanalysis data were thus used to force the mass balance model over the period 1979-2016, and the AWS data was used to apply a first-order bias correction to the reanalysis data. Data from the North American Regional Reanalysis (NARR) (Mesinger et al., 2006) were chosen for this study because of its higher temporal (3 h) and spatial (32 km) resolution compared to other commonly used products, such as ERA interim (6-hourly, ~80 km resolution) and NCEP (6-hourly, ~ 600 km resolution) reanalyses. NARR precipitation have been found to be superior to other global reanalysis products in the US (Bukovsky and Karoly, 2007) and to represent well air temperature and humidity at high elevation sites in southern BC, Canada (Trubilowicz et al., 2016). Chen et al. (2017) also showed that NARR reproduced well the seasonality of precipitation and temperature for 12 catchments across US and Canada.

NARR data were acquired from the National Center for Environmental Prediction (NCEP) at the National Centers for Atmospheric Research (NCAR) for the nine gridcells closest to the on-glacier AWS (see Figure 1b). The NARR gridpoint whose center point is closest to the on-glacier AWS has an elevation of 2430 m a.s.l., i.e. 237 m higher than the AWS. The following NARR variables were used: (i) instantaneous values of air temperature and relative humidity at 2 m above the surface (TMP2m-ANL, RH2m-ANL), (ii) wind speed vectors at 10 m above the model surface (U and V wind components: UGRD10m-ANL, VGRD10m-ANL), (iii) surface 3-hourly accumulated precipitation (APCPsfc-ACC), and (iv) 3-hourly averaged surface downward shortwave radiation fluxes (DSWRFsfc-AVE).

Three-hourly NARR variables were interpolated to the center of the hourly averaging interval used by the AWS datalogger. For instantaneous variables (ANL) the concurrent time tag was used for the interpolation while for averages (AVE) the time at the center of the averaging interval was used. Linear interpolation was used for relative humidity and wind speed. However, both incoming solar radiation and air temperature have strong diurnal cycles at the AWS site. Over the year, solar noon varies between 12 h 41 to 12 h 56 and sunshine duration varies between 7.75 to 16.75 hours. The 3-hourly NARR data could thus underestimate the daily peaks in solar radiation and air temperature, especially since the midday NARR 3-hourly average value spreads between 11 h 00 and 14 h 00. However, given that solar noon occurs near the middle of this interval, the NARR midday solar radiation average may in fact well approximate the peak mid-day value, while the 14 h 00 instantaneous temperature value is close to the time of maximum daily temperature. Nevertheless, to reduce the probability of the diurnal cycle being attenuated in the interpolated NARR data, a shape-preserving piecewise cubic interpolation was used to interpolate air

temperature and solar radiation to an hourly interval. The 3-hourly accumulated (ACC) precipitation totals were disaggregated to hourly values by dividing the 3-hour totals into three exact quantities.

### 3.2.4 Downscaling NARR to weather stations

Downscaling the NARR variables to the glacier model grid involved two steps: (1) interpolation of the NARR gridded data to the reference weather stations; (2) bias correction of the interpolated NARR data. Two interpolation methods were used and compared to extract NARR time-series. The first one is a simple nearest neighbour interpolation, i.e., the NARR grid point whose center point is closest to the reference stations (the on-glacier AWS and the merged Parker Ridge/Columbia precipitation station: see Figure 1 for locations) was used. The second method used bilinear interpolation from the nine NARR grid points closest to the weather stations.

A simple bias correction procedure (Teutschbein and Seibert, 2012) was used to correct NARR biases. Air temperature, relative humidity, wind speed and solar radiation from the interpolated NARR time series were corrected relative to the on-glacier AWS. Since precipitation was not measured at the glacier AWS, the NARR precipitation were corrected with the merged historical precipitation record from the Parker Ridge/Columbia stations. Several data gaps remained in the merged record, and no observations were available after 2008. Hence only days with observations were used for bias correction over the period when NARR overlapped the merged precipitation record (1980-2008). Two simple bias correction methods were tested and compared, namely scaling and empirical quantile mapping (EQM) (e.g. Teutschbein and Seibert, 2012; Wetterhall et al., 2012). The scaling method is the simplest, in which the NARR outputs are scaled with the difference (additive correction) or quotient (multiplicative correction) between the mean NARR and mean of observations. An additive correction was used for unbounded variables ($T_{a,NARR}$) and a multiplicative correction for strictly positive variables ($RH_{NARR}$, $WS_{NARR}$, $G_{NARR}$ and $P_{NARR}$) as it also preserves the frequency. Because errors in incoming solar radiation can originate from improper representation of the atmospheric transmissivity and cloud cover in NARR and/or shading differences between the NARR smoothed topography and the real topography surrounding the AWS, a time-varying scaling method was used to correct the NARR global shortwave radiation data ($G_{NARR}$). A mean diurnal multiplicative correction factor was calculated by scaling the mean observed diurnal $G$ cycle with that of the hourly-interpolated NARR. A separate diurnal correction factor was calculated for each month of the year, to account for the seasonality in sun angle and related errors between NARR and observations.

The bias correction methods were evaluated against the glacier AWS data using split sample cross-validation, and compared with the baseline performance, i.e., without corrections to the NARR variables. The AWS data was split into two one-year sub-periods on which downscaling methods were respectively calibrated and validated; then both sub-periods were inverted, and the mean validation statistics calculated. For precipitation the entire historical record was used, so validation sub-periods are longer than for other variables. The cross-validated Pearson correlation coefficient ($r$), mean error (bias) and root mean

square error (RMSE) were used for performance assessment. The performance of bias-correction was evaluated at both hourly and daily time intervals.

### 3.2.5 Extrapolation of NARR data to the glacier DEM

The downscaled NARR data was extrapolated from the reference stations to the glacier DEM. Because data gaps remained in the merged Parker Ridge/Columbia precipitation record, the downscaled NARR precipitation record was used to force the mass-balance model. As the glacier mass balance model only considers a constant precipitation lapse rate, a mean lapse rate of 15.6% 100 m$^{-1}$ was calculated from the weather station network for the months of November to March, when snow precipitation is most abundant on the glacier and the relation between precipitation and elevation is strongest (Supplementary Material). The extrapolated total precipitation was split between rain and snowfall according to a threshold temperature ($T0$) of 1.5 °C, at which 50% of the precipitation falls as snow and 50% as rain. This value corresponds to a typical rain-snow temperature threshold for continental mountain ranges and was inferred from the relative humidity at the AWS site (83%) following Jennings et al. (2018). A linear interpolation of the rain/snow fraction is performed between $T0$-1 °C (100% snow) and $T0$+1 °C (100% rain).

A mean monthly air temperature lapse rate was calculated from the permanent weather station network. Lapse rates were calculated by linear regression of mean temperature against elevation, using a minimum of five stations for each month, depending on available data. Since diurnal lapse rate variations can affect glacier melt simulations (Petersen and Pellicciotti, 2011), the on-glacier *HOBO* sensors were used to calculate a mean diurnal air temperature lapse rate cycle on the glacier. Diurnal anomalies were produced by subtracting the mean on-glacier lapse rates from this diurnal cycle, and were then added to the mean monthly lapse rates estimated from the permanent weather station network. Hence, the lapse prescribed to the model varied on a diurnal as well as on a seasonal (monthly) scale, and was used to extrapolate air temperature to the glacier DEM.

In the absence of constraining data, wind speed and relative humidity were assumed spatially invariant, as done in earlier modelling studies of mountain glaciers (e.g. Anderson et al., 2010; Anslow et al., 2008; Arnold et al., 1996; Arnold et al., 2006; Hock and Holmgren, 2005; Mölg et al., 2008). Wind speed can be expected to be relatively constant down-glacier due to the presence of a katabatic wind or an 'ice field breeze' wind, as found on the neighboring Athabasca outlet glacier (Conway et al., 2021); however, it is possible that the more open accumulation zone of Saskatchewan Glacier could have higher winds than measured at the mid-glacier AWS (Figure 1). Global solar radiation from the downscaled NARR ($G_{NARR}$) was separated into direct ($I$) and diffuse ($D$) components, which were then extrapolated individually to each gridcell considering terrain effects of the multitemporal DEMs. Further details are given in the model description in section 3.3.

### 3.3 Mass balance model

The physically-based, distributed glacier mass balance model DEBAM (Hock and Holmgren, 2005) was used to simulate the mass balance of Saskatchewan Glacier over the period 1979-2016. The surface mass balance is expressed as:

$$b(t) = P_s(t) - M(t) - S(t) \tag{1}$$

Where $b(t)$ is the point surface mass balance at time $t$, $P_s$ is snow precipitation, $M$ is melt and $S$ is sublimation. The model calculates the distributed mass and energy balance on each 100 x 100 m grid cell from the hourly downscaled NARR

meteorological forcing data including air temperature, relative humidity, precipitation, wind speed and incoming shortwave global radiation. The energy at the surface available for melt on the glacier, $Q_M$ (W m$^{-2}$), was calculated according to Eq. (2) and converted into meltwater equivalent $M$ (m w.e. h$^{-1}$) using the latent heat of fusion:

$$Q_M + Q_G + Q_R + Q_L + Q_S + LW_S \downarrow + LW_T \downarrow + LW \uparrow + (1 - \alpha)(I + D_S + D_T) = 0 \tag{2}$$

where $I$ is the direct (beam) incoming shortwave solar radiation, $D_S$ and $D_T$ are the diffuse sky and terrain shortwave radiation,

respectively, $\alpha$ is the albedo, $LW_S \downarrow$ and $LW_T \downarrow$ are the longwave sky and terrain irradiance, respectively, $LW \uparrow$ is longwave outgoing radiation, $Q_S$ is the sensible-heat flux, $Q_L$ is the latent-heat flux and $Q_R$ is the energy supplied by rain (Hock and Holmgren, 2005). The ground heat flux in the ice or snow, $Q_G$, is often small for temperate glaciers and was neglected (e.g., Hock, 2005; Yang et al., 2021). Fluxes are positive towards the glacier surface and measured or calculated in W m$^{-2}$. The model allows for different parameterizations for calculating energy balance components, depending on the availability of

forcing data. The parameterizations used in this work are detailed in the next sections.

### 3.3.1 Shortwave incoming radiation

Following Hock and Holmgren (2005), the separation of the downscaled NARR global radiation ($G_{NARR}$) into direct ($I$) and diffuse ($D$) radiation is based on an empirical relationship between the ratio of measured global radiation to top-of-atmosphere radiation, $G_{NARR}/I_{TOA}$, and the ratio of diffuse to global radiation, $D/G_{NARR}$. Total diffuse radiation $D$ calculated at the AWS

is then subtracted from the global radiation to yield the direct solar radiation at the AWS site, $I_S$. Topographic shading is calculated at each hour and for each gridcell from the path of the sun and the effective horizon. If the AWS is shaded by surrounding topography, any measured global radiation is assumed diffuse. Direct radiation $I$ is obtained at each gridcell following Hock and Holmgren (2005) as:

$$I = \frac{I_S}{I_{SC}} I_C \tag{3}$$

where the subscript $s$ refers to the location of the climate station and $c$ denotes clear-sky conditions. $I_C$ is the potential clear-sky direct solar radiation which accounts for the effects of slope and aspect of each grid cell, as well as shading from surrounding topography. The ratio $I_S/I_{SC}$ measured at the AWS accounts for deviations from clear-sky conditions, expressing

the reduction of potential clear-sky direct solar radiation mainly due to clouds. The ratio is assumed to be spatially constant, which is reasonable given the large (~400 km) correlation length scale of cloud cover (Jones, 1992). Eq. (3) can not be applied

when the AWS is shaded, since $I_C = 0$. In this case and for glacier grid cells that remain illuminated, the last ratio that could be obtained before the AWS grid cell became shaded is applied, which assumes that cloud conditions remain constant until the climate station is illuminated again (usually the next morning). The constant ratio was applied to 57% of the glacier surface which was sunlit while the AWS was shaded, for a mean and maximum duration of 0.73 and 2.16 hours, respectively. The impact on the radiative balance is thus considered to be small because this situation occurs in the mornings and end of days at

low sun illumination angles, and also because the temporal correlation length scale of cloud cover is a few hours (Jones, 1992). The total diffuse radiation ($D$) is calculated as:

$$D = D_0 F + \alpha_m G_{NARR}(1 - F) \tag{4}$$

where the first righthand term represents sky radiation ($D_S$) and the second term terrain radiation ($D_T$). $D_0$ is diffuse radiation from an unobstructed sky calculated at the AWS and is considered spatially constant. $F$ is the grid cell sky-view factor defined

by Oke (1987) and $G_{NARR}$ is the downscaled NAAR global radiation at the AWS. The mean albedo ($\alpha_m$) of the surrounding terrain obtained for every hour is the arithmetic mean of the modelled albedo of all grid cells for the entire glacier (Hock and Holmgren, 2005).

### 3.3.2 Albedo

The albedo parameterisation of Oerlemans and Knap (1998) was used to simulate the albedo ($\alpha$):

$$\alpha_{snow}(t) = \alpha_{firn} + \left(\alpha_{frsnow} - \alpha_{firn}\right) exp\left(\frac{s-t}{t^*}\right) \tag{5}$$

$$\alpha(t) = \alpha_{snow}(t) + \alpha_{ice} - \alpha_{snow}(t)\, exp\left(\frac{d}{d^*}\right) \tag{6}$$

where $\alpha_{snow}(t)$ is snow albedo, $\alpha(t)$ the final glacier albedo at time $t$, $\alpha_{firn}$ is the characteristic albedo of firn, $\alpha_{frsnow}$ is the characteristic albedo of fresh snow and $\alpha_{ice}$ is the characteristic albedo of ice. The time scale ($t^*$) determines how fast the snow albedo decays over time (days) and approaches the firn albedo after a fresh snowfall, $d^*$ is a characteristic snow depth

scale (cm) controlling the transition from snow albedo to ice albedo, $s$ is the day of the last snowfall and $d$ is snow depth (cm). The constant, characteristic albedo values were set to $\alpha_{frsnow} = 0.9$ for fresh snow based on observations at the AWS. Ice albedo was mapped using 17 of the 18 cloud-free, end-of-summer Landsat images used to delineate the mean snowline position. Atmospherically corrected surface reflectance from the Landsat 5 ETM and Landsat 7 ETM+ sensors were converted to broadband albedo following Knap et al. (1999). A median albedo map was produced, from which the distribution of ice

albedo values was extracted in a region of interest extending below the mean snowline and excluding the glacier margins where shade effects were noticed (Supplementary Material). The median of the distribution (0.24) was used as the representative ice albedo ($\alpha_{ice}$). The characteristic time ($t^*$) and depth ($d^*$) scales were calibrated using snow depth and albedo

measurements at the AWS. Since the AWS was on a moraine the value for $\alpha_{ice}$ was set instead to the measured soil albedo for calibration purposes. The optimum values used in the model, found by minimizing the RMSE of the simulated albedo, were $t^* = 14$ day and $d^* = 3$ cm.

### 3.3.3 Longwave calculation

Since no observations of incoming longwave radiation were available at the AWS for bias-correction, NARR longwave radiation was not used for model forcing because it would carry an elevation bias and would not account for terrain effects. Instead, the incoming sky longwave radiation ($LW_S\downarrow$) was calculated based on the Stefan-Boltzmann equation, which relies on independent variables of air temperature and atmospheric emissivity according to Eq. (7):

$$LW_S \downarrow = F\varepsilon_{NARR}\sigma T_{a,NARR}{}^4 \tag{7}$$

Where $F$ is the skyview factor, $\varepsilon_{NARR}$ is the atmospheric emissivity from NARR, $\sigma$ is the Stefan-Boltzmann constant ($5.67 \times 10^8$ m$^{-2}$ K$^{-4}$) and $T_{a,NARR}$ is the downscaled NARR air temperature in Kelvin. We adjusted the $LW_S\downarrow$ calculation in DEBAM to include the spatial variability in air temperature, i.e., by using $T_{a,NARR}$ extrapolated to the glacier DEM, since the default parameterisation only used air temperature measured at the AWS location for the entire glacier area. This led to an overestimation of melt in the accumulation zone and an underestimation in the ablation zone – both corrected when including the distributed $T_{a,NARR}$ in Eq. 7. Terrain longwave irradiance, $LW_T\downarrow$, was calculated using the parameterization by Plüss and Ohmura (1997) for snow covered alpine terrains:

$$LW_T \downarrow = (1 - F)\pi(L_b + aT_{a,NARR} + bT_s) \tag{8}$$

where $F$ is the sky view factor, $L_b = 100.2$ Wm$^{-2}$ sr$^{-1}$ is the emitted radiance of a $0°$ black body, $T_s$ is the temperature of the emitting surface and $a = 0.77$ Wm$^{-2}$ sr$^{-1}$ and $b = 0.54$ Wm$^{-2}$ sr$^{-1}$ are coefficients calibrated for snow-covered alpine environments (Plüss and Ohmura, 1997).

Outgoing longwave radiation ($LW\uparrow$) is calculated from the Stefan Boltzmann equation and the simulated surface temperature ($T_s$). $T_s$ is obtained in an iterative process by lowering surface temperatures in case of negative energy balances until the energy balance equals zero (Hock and Holmgren, 2005).

### 3.3.4 Turbulent heat fluxes

The turbulent sensible ($Q_S$) and latent ($Q_L$) heat fluxes were calculated from the bulk aerodynamic method (Hock and Holmgren, 2005) based on air temperature ($T_{a,NARR}$), wind speed ($WS_{NARR}$) and vapour pressure ($e$) at height $z = 2$ m above the surface:

$$Q_S = \rho C_p \frac{k^2}{\left[\ln\left(\frac{z}{z_{0w}}\right)-\psi_M\left(\frac{z}{L}\right)\right]\left[\ln\left(\frac{z}{z_{0T}}\right)-\psi_M\left(\frac{z}{L}\right)\right]} WS_{NARR}(T_{a,NARR} - T_s) \tag{8}$$

$$Q_L = L_v \frac{0.623\rho_0}{P_0} \frac{k^2}{\left[\ln\left(\frac{z}{z_{0w}}\right)-\psi_M\left(\frac{z}{L}\right)\right]\left[\ln\left(\frac{z}{z_{0e}}\right)-\psi_M\left(\frac{z}{L}\right)\right]} WS_{NARR}(e - e_s) \tag{9}$$

where $\rho$ is air density at sea level (1.29 kg m$^{-3}$), $P_0$ is the mean atmospheric pressure at sea level (101 325 Pa), $C_p$ is the specific heat capacity of air (1005 J kg$^{-1}$ K$^{-1}$), $k$ is the von Kármán's constant (0.4), $T_s$ is the surface temperature in Kelvin, $e$ is the air vapour pressure as defined before, $e_s$ is the surface vapour pressure in Pa and $z_{0w}$, $z_{0T}$ and $z_{0e}$ are the roughness lengths for the logarithmic profiles of wind speed, temperature and water vapour, respectively, $\psi_M$, $\psi_H$ and $\psi_E$ are the stability functions, $L$ is the Monin–Obukhov length, and $L_v$ is the latent heat of evaporation (2.514 × 10$^6$ J kg$^{-1}$) or sublimation (2.849 × 10$^6$ J kg$^{-1}$), depending on surface temperature and the direction of the latent heat flux. If $Q_L$ is positive, condensation occurs if the surface is melting, or deposition if the surface is frozen. Sublimation occurs when $Q_L$ is negative. The aerodynamic roughness length ($z_0$) for snow and ice influences the intensity of turbulent fluxes at the glacier surface. Typical $z_0$ values for glacier snow ($z_{0\_snow}$) range between 0.5 and 6 mm (Brock et al., 2006; Fitzpatrick et al., 2019; Munro, 1989), while $z_0$ for smooth glacier ice surfaces ($z_{0\_ice}$) typically range between 0.1 and 6 mm (Brock et al., 2006). Munro (1989) measured $z_0$ values between 0.67 and 2.48 mm along and across the grain of the ice, respectively, and 5-6 mm for snow on nearby Peyto Glacier, which has a similar ice facies morphology as the Saskatchewan Glacier, based on our field observations. A mean $z_{0\_ice}$ of 1.58 mm and $z_{0\_snow}$ of 5.5 mm was thus used in the model. The roughness length for temperature and water vapour were both considered to be two orders of magnitude less than roughness lengths for wind (Hock and Holmgren, 2005).

### 3.4 Model validation and uncertainty analyses

The simulated mass balance was validated at the point-scale against available seasonal and annual glaciological mass balance observations since 2012, and at the glacier scale using the reconstructed geodetic mass balance from 1979 to 2016. These data are described in detail in the Supplementary Material. The sensitivity of the reconstructed mass balance was tested with respect to (i) the NARR interpolation method; (ii) the NARR bias-correction method; (iii) replacing NARR forcings by their AWS counterpart; (iv) uncertain model parameters. For (i), the model was run with NARR forcings respectively interpolated with the nearest neighbour and bilinear methods. For (ii), the model forced with the bias-corrected NARR forcings was compared with a model forced with raw NARR, but correcting for the elevation difference between NARR and the reference stations using the mean measured temperature and precipitation lapse rates (e.g. Fiddes and Gruber, 2014). For (iii), the NARR forcings ($G_{NARR}$, $T_{a,NARR}$, $WS_{NARR}$, $RH_{NARR}$) were replaced one-at-a-time by the AWS observations and the simulated point mass balances compared with stakes observation for 2015, the only year with continuous AWS data and concurrent glaciological observations. For (iv), while the physical nature of the model did not require formal calibration, four uncertain model parameters were subjected to a sensitivity analysis to characterize their impact on the modelled mass balance. The precipitation lapse rate was varied within ± 4% 100 m$^{-1}$, which corresponds to the standard deviation of the precipitation lapse rate calculated

from the permanent weather network (see Sect. 3.2.5 and Supplementary Material). The ice albedo ($\alpha_{ice}$) was varied within ± 0.03, which corresponds to the spatial standard deviation of ice albedo observed from satellite images (see Sect. 3.3.2). The aerodynamic roughness lengths for ice ($z_{0\_ice}$) and snow ($z_{0\_snow}$) were varied within ± 1 mm, which covers the range of values by Munro (1989) on nearby Peyto glacier (see Sect. 3.3.4). The sensitivity to roughness lengths was also extended to ± one order of magnitude, as an extreme case.

## 3.5 Climate sensitivity

The validated DEBAM model was used to perform a climate sensitivity analysis of the reference mass balance (with respect to the 2010 glacier hypsometry: $B_{r,\,2010}$) to potential changes in air temperature ($\Delta T_a$) ranging between 0 to 7 °C (1 °C interval) and precipitation ($\Delta P$) ranging between -20 to +20% (5% interval). These warming and precipitation changes scenarios encompass mean annual changes projected by ensemble general circulation models (GCM) simulations for the mid (2041-2070) and late (2071-2100) 21st century relative to the most recent 30-year climatological period (1981-2010) and under different Representative Concentration Pathway (RCP) scenarios (IPCC, 2013). The ensemble climate projections from the Coupled Model Intercomparison Project Phase 5 (CMIP5) were obtained from the KNMI Climate Change Atlas (Trouet and Van Oldenborgh, 2013) for RCP 2.6 (n=32), 4.5 (n=42), 6.0 (n=25) and 8.5 (n=39) for the gridpoint closest to the ELA of Saskatchewan Glacier (Figure 1). The number of simulations (n) depended on the availability of the CMIP5 models for each scenario (IPCC, 2013). The *IPCC AR5 Atlas* subset was used, which uses only a single realisation of each model and weights all models equally, where model realisations differing only in model parameter settings are treated as different models (IPCC, 2013). The DEBAM model was run 63 times for every combination of $\Delta T_a$ and $\Delta P$ perturbation imposed on the $T_{a,NARR}$ and $P_{NARR}$ records over the 30-year reference period 1981-2010. Changes in mass balance for each sensitivity run were plotted as response surfaces, which provide a simple way to assess climate sensitivity across a range of possible climate change scenarios (e.g. Aygün et al., 2020b; Prudhomme et al., 2010). Mean temperature and precipitation changes along with their 95% confidence intervals were overlained onto the response surfaces to show the most likely future climate trajectories given by the GCM projections.

## 4 Results

### 4.1 Meteorological observations

Daily and monthly averages of air temperature ($T_a$), relative humidity ($RH$), incoming solar radiation ($G$) and wind speed ($WS$) measured at the glacier AWS show notable differences between the two years of observation (Figure 2). The winter of 2014-2015 was, overall, colder than 2015-2016, with frequent cold excursions below -15 °C and a winter absolute minimum of -27 °C vs. -17 °C in 2015-16, although conditions were warmer in December. Relative humidity was generally high throughout the year (mean = 79%), illustrating the predominantly humid climate of the Columbia Icefield, but decreases noticeably in

summer. The variability in daily *RH* is similar between the two years of measurements. The incoming solar radiation shows pronounced seasonality, varying between ~50 W m$^{-2}$ in winter and ~300 W m$^{-2}$ in summer, with daily variations between 50 W m$^{-2}$ in winter and 150 W m$^{-2}$ in summer caused by variable cloud cover. A gentle breeze blows on average on the glacier (mean wind speed = 4.46 m s$^{-1}$), but wind speed shows significant day-to-day variations as well as higher values in winter. A

450 gradual increase in wind speed is notably observed from the lowest monthly mean value in May 2015 (1.76 m s$^{-1}$) to a maximum in February (7.13 m s$^{-1}$). The historical precipitation records from the Columbia Icefield and Parker Ridge stations contain several gaps but still portrays the seasonal and interannual variability in precipitation near the glacier (Figure 2e). The mean annual accumulated precipitation throughout the historical period with complete data was 874 mm a$^{-1}$ but varied between 276 mm a$^{-1}$ and 1704 mm a$^{-1}$. Precipitation are more abundant in winter, with 58% of precipitation falling between October to

455 March, mostly as snow, and 42% falling during April-September, mostly as rain.

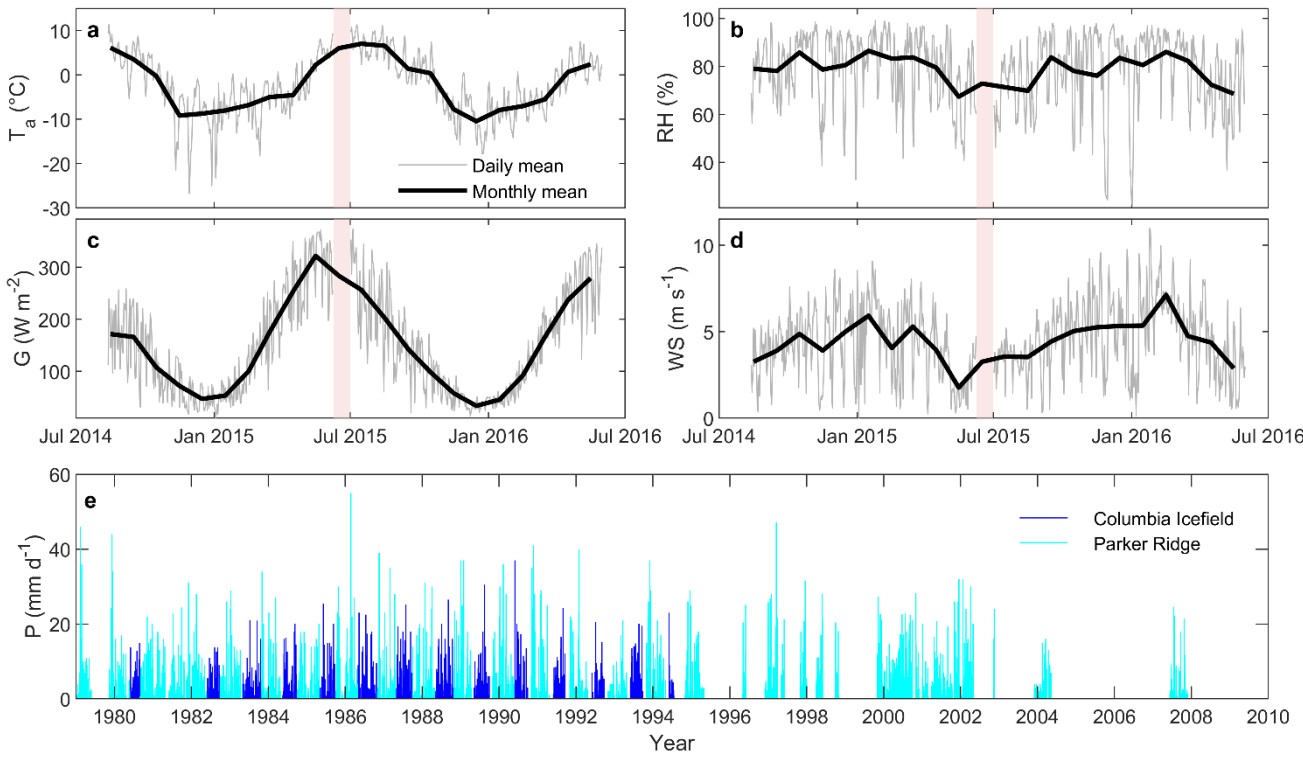

**Figure 2. Two-year record from the Saskatchewan Glacier AWS (2014-2016). (a) Air temperature (*T$_a$*); (b) relative humidity (*RH*); (c) incoming global solar radiation (*G*); (d) wind speed (*WS*). Pink shades delineate the data gap caused by the fall of the AWS (11**
**to 30 June 2015). (e) Daily precipitation records from Parker Ridge and Columbia Icefield permanent stations. Note the several gaps after 1995 when the Columbia Icefield station was interrupted.**

## 4.2 NARR downscaling

The NARR meteorological variables used to drive the glacier mass balance model were compared with data from the glacier AWS (2014-2016) and the 29-year long merged daily precipitation record (Figure 3). Even prior to applying bias correction, $T_{a,NARR}$, $RH_{NARR}$ and $G_{NARR}$ show a good correlation with AWS observations on a daily scale, for both NARR spatial interpolation methods. As expected, the correlation is poorer for $WS_{NARR}$, likely because the local glacier katabatic wind recorded by the AWS is not well represented in NARR due to its coarse grid resolution. The NARR precipitation is also rather poorly correlated with observations ($r = 0.30$). Biases in raw NARR variables are relatively small compared to the mean and range of values recorded (blue dots in Figure 3), except for $G_{NARR}$ (30.4 W m$^{-2}$) and $P_{NARR}$ (0.55 mm d$^{-1}$), which represent 15% and 25% of their mean measured values over their period of observation, respectively. The cold bias (-1.26 °C) observed for $T_{a,NARR}$ from the closest gridcell is consistent with the elevation difference between the AWS (2193 m) and the NARR gridcell (2430 m) ($\Delta Z = 237$ m), which results in an expected temperature difference of -1.19°C using the mean observed lapse rate of -0.5 °C 100 m$^{-1}$ (see Sect. 4.3). Neither the scaling nor the EQM correction methods improved the Pearson correlation coefficient ($r$) – primarily since it is a relative measure of the synchronicity between two-time series and is unaffected by the mean values. The EQM method was found to improve $T_{a,NARR}$ best, closely followed by the scaling method, while the scaling method was slightly superior for $RH_{NARR}$, $WS_{NARR}$ and $P_{NARR}$. However, scaling only slightly reduced the errors for $WS_{NARR}$ and $P_{NARR}$ and had no effect on $RH_{NARR}$, which had an initial low error. The diurnal scaling correction applied to $G_{NARR}$ also reduced its errors. Overall, the scaling bias correction method was globally the more efficient approach across all variables and both NARR spatial interpolation methods, and was thus applied to all variables for consistency except for relative humidity, which was left uncorrected. Similar results, although with expectedly higher errors, were found for the interpolated NARR hourly data (Supplementary Table S3).

The bilinear NARR interpolation method resulted in slightly lower RMSE and bias values for the raw variables, i.e., before bias correction (blue bars and dots in Figure 3), except for the slightly higher bias for $RH$. However, after applying the station-based bias correction, the bias and RMSE values were very similar among the two methods. As such, the NARR forcings downscaled from the nearest NARR gridcell were used as primary model forcings for the mass balance reconstruction and climate sensitivity experiments, and the sensitivity of the reconstructed mass balance to NARR spatial interpolation method was further investigated in Section 4.6.

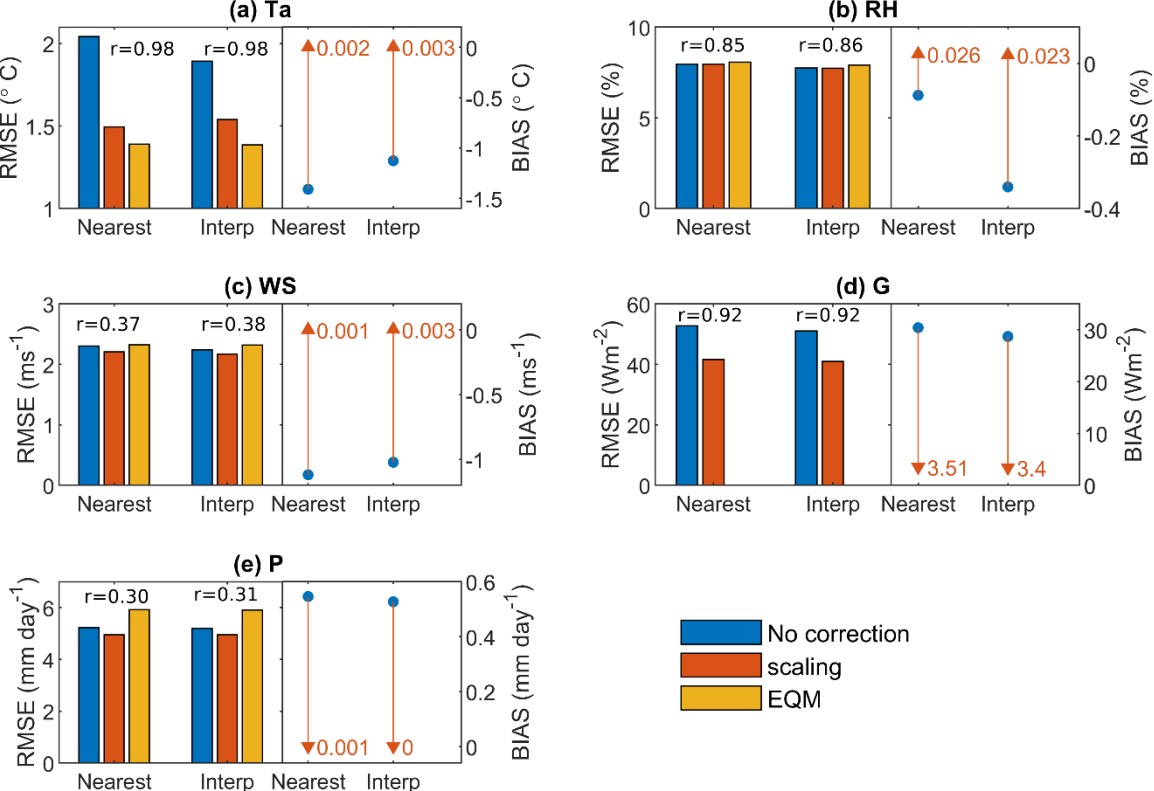

**Figure 3. Comparison between NARR reanalyses and automated weather station (AWS) meteorological variables (2014-2016). Each panel shows the cross-validated root mean square error (RMSE) between daily NARR and AWS variables, before (blue) and after bias correction (red: scaling method; yellow: EQM method), for the two NARR spatial interpolation methods (*Nearest*: nearest gridcell; *Interp*: bilinear interpolation). The cross-validated correlation coefficient (*r*), which changes little after bias correction, is shown on top of bars for each NARR interpolation method. Quivers on the left-hand side of panels show the cross-validated bias before (blue dot) and after (red triangle) applying the scaling bias correction method. (a) Air temperature; (b) relative humidity; (c) wind speed; (d) incoming global solar radiation; (e) precipitation.**

Monthly and annual averages of the downscaled NARR variables from the nearest NARR gridcell are displayed in Figure 4. There is no visible trend in mean annual $T_{a,NARR}$ over the 30 years period except since 2010, but there is a noticeable increase in minimum temperatures, with e.g., only two years with a monthly mean colder than -15 °C in 2000-2015 compared to seven years prior to 2000. The positive trend seen in mean annual $RH_{NARR}$ is driven by increasing annual minima while annual maxima show no trend, and so the seasonal amplitude decreases over time. The monthly $RH_{NARR}$ averages decrease in July and August (mean = 72%) while winter months have higher values (mean = 80-82%) (Figure 4b). No noticeable trends occur in atmospheric emissivity ($\varepsilon_{NARR}$) and $G_{NARR}$, despite the observed trend in $RH_{NARR}$ (Figure 4d, f). A progressive decline in $WS_{NARR}$ occurs from 1984 onward, reaching the lowest annual value of the period in 1995 (~4.3 m s$^{-1}$) (Figure 4c). A more subdued increase in $WS_{NARR}$ occurs afterward until 2010, followed by a decline. Finally, mean monthly precipitation shows no

long-term trend but significant seasonal and interannual variability (Figure 4e). A slight increasing trend in $P_{NARR}$ is noted in the last part of the record, since ~2000.

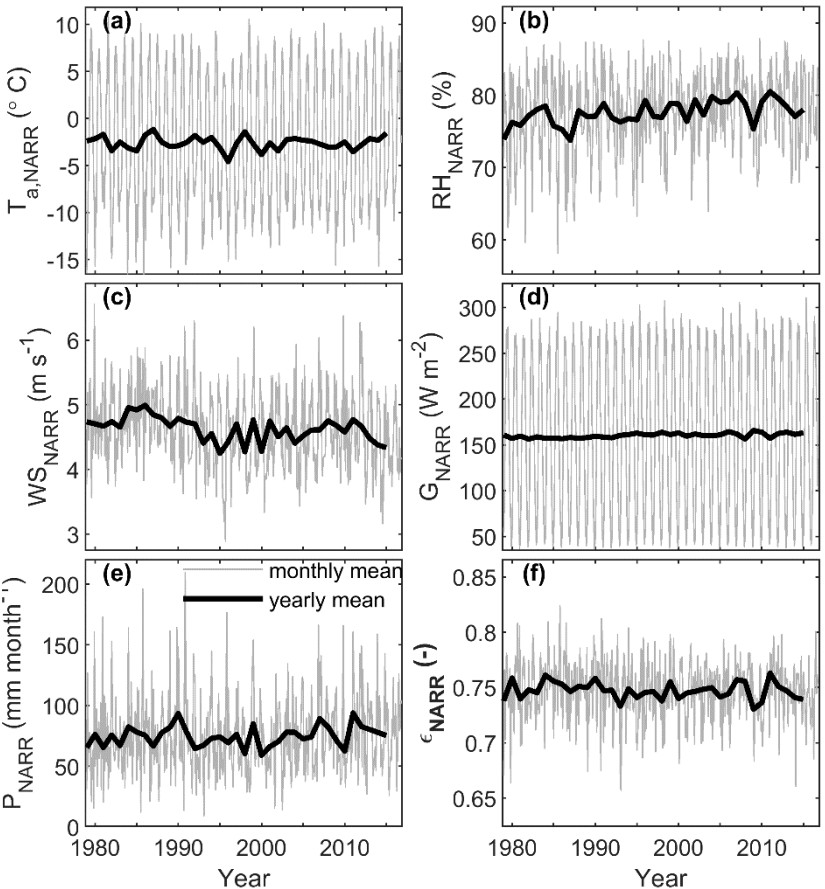

**Figure 4. Downscaled NARR variables from the nearest NARR gridcell and used to drive the DEBAM model. Grey solid lines represent monthly means and black solid lines represent annual averages. (a) Air temperature; (b) relative humidity; (c) wind speed; (d) incoming global solar radiation; (e) total precipitation; (f) atmospheric emissivity.**

### 4.3 Air temperature lapse rates

On-glacier diurnal air temperature lapse rates were found to vary between -0.55 °C 100 m$^{-1}$ at night and to increase during the day, reaching a maximum of -0.34 °C 100 m$^{-1}$ at midday (Figure 5a). The strength of the linear relationship between air temperature and elevation, as measured by the correlation coefficient ($r$), is generally high ($r > 0.95$) but decreases slightly during daytime hours ($r = 0.92$). While wind speed increased during the day, down-glacier winds prevailed, with little deviation of the wind direction within the day (Figure 5a). The wind blows dominantly down-glacier, with the relative wind direction showing a mixed contribution of the main accumulation area upwind of the AWS and the glacierized plateau North of the

AWS. Stronger daytime down-glacier winds, possibly driven by a larger thermal gradient between the lower ice-free valley and the glacier, could result in down-glacier cooling and correspondingly shallower near-surface lapse rates or even inverted lapse rates, as shown on neighbouring Athabasca glacier (Conway et al., 2021). Closer inspection of hourly lapse rates revealed that inversions only occurred 1.7% of the time between May and August on Saskatchewan Glacier. On a monthly scale, the lapse rate, calculated from seven stations from the permanent network, varied between -0.58 °C 100 m$^{-1}$ and -0.42 °C 100 m$^{-1}$ without any systematic seasonal pattern (Figure 5b). The correlation for the monthly lapse rates is also more variable than for the diurnal lapse rates, varying between low values ($r = 0.6$) in winter to higher values ($r = 0.94$) in summer. The mean on-glacier summer (May-August) lapse rate (-0.46°C 100 m$^{-1}$) was very close to that calculated from the permanent weather station network for the same period (-0.49 °C 100 m$^{-1}$), which gives confidence in extrapolating the monthly lapse rates from the network to the glacier surface. Superimposing the on-glacier diurnal lapse rates anomalies onto the mean monthly lapse rates allowed to better represent the diurnal changes associated with the glacier wind.

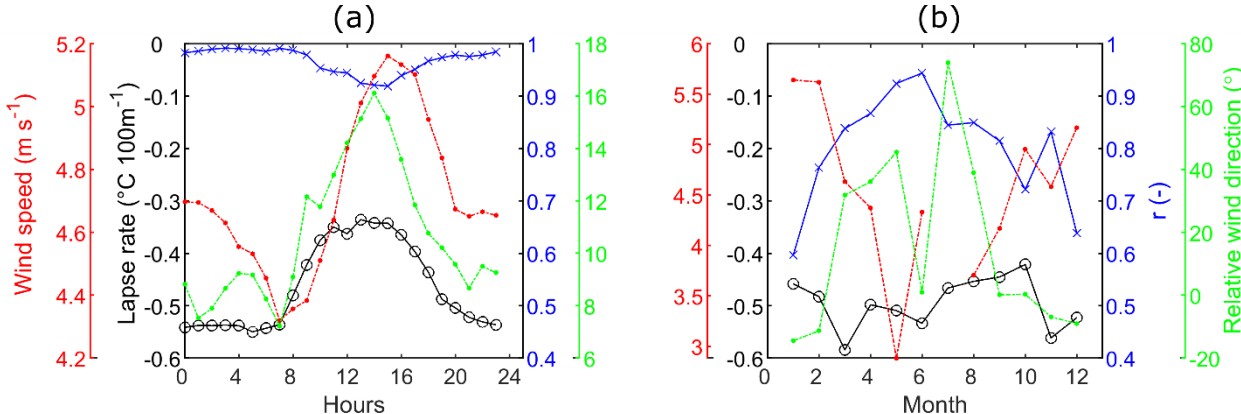

**Figure 5. Calculated air temperature lapse rates. The black axis represents the air temperature lapse rate in °C 100 m$^{-1}$, the blue axis represents the correlation ($r$) between air temperature and elevation, the red axis represents wind speed and the green axis the wind direction relative to the main glacier axis (0° = down-glacier, 180° = up-glacier). (a) Diurnal temperature lapse rate from the five HOBO$^{TM}$ microloggers installed on ablation stakes from May to August 2015 (See Figure 1). (b) Seasonal variation of the lapse rate derived from the permanent weather stations (see Figure 1). Wind speeds and directions on both panels are from the glacier AWS.**

### 4.4 Model performance

#### 4.4.1 Comparison with glaciological mass balance

The mass balance simulated with DEBAM was compared with point glaciological mass balance observations available between 2012 and 2016. Overall, the seasonal and annual mass balance components are well simulated by the model, with most observations lying near the 1:1 line and with Nash-Sutcliffe Efficiency (NSE, Nash and Sutcliffe, 1970) coefficients of 0.84 for the winter balance ($b_w$, n=49), 0.83 for the summer balance ($b_s$, n=12) and 0.91 for the annual balance ($b_a$, n=12) (Figure 6). Before the adjustment of the atmospheric emissivity calculation in the $LW\downarrow$ equation (see Sect. 3.3.3), the model

tended to overestimate melt in the accumulation zone and underestimate it in the ablation zone. The NSE was increased by 0.04 for $b_w$, 0.07 for $b_s$ and 0.06 for $b_a$ after modifying the parametrisation. The modelled $b_w$ was underestimated in 2016 in the upper part of the glacier and overestimated in the lower part, suggesting that the precipitation gradient for that year significantly differed from the other years. This shows one limitation of the current model configuration, which uses a constant, average precipitation lapse rate to distribute precipitation over the glacier surface. 2016 was a dry year, with the ultrasonic gauge on the glacier AWS recording a small amount of snow accumulation during winter (25 cm in 2016 vs. 135 cm in 2015). Observations from ablation stakes are more limited, and despite the overall good model performance as seen by the linear relationship between observed and simulated $b$ and the high NSE values, modelled $b_s$ and $b_a$ were slightly underestimated in 2014 and 2016 and overestimated in 2015 compared to observations (Figure 6).

The simulated mass balance gradient compares generally well with observations for the three years with available $b_a$ measurements (Figure 6d). Overestimation of ablation at the two ablation stakes from 2014 is apparent, however, leading to underestimated mass balance ($b_a$) in the upper glacier for that year. The equilibrium line altitude (ELA) was ~2600 m for 2014-2016, which is near the average ELA of 2587 m simulated for the entire 1979-2016 period. The mean simulated mass balance gradient for the three validation years (2014-2016) was 0.98 m w.e. 100 m$^{-1}$ in the ablation zone, with a steeper inflection below the ELA, and decreasing to 0.32 m w.e. 100 m$^{-1}$ in the accumulation zone (up to 2900 m, where the model is constrained by observations, see Figure 6d). Long-term values were 0.96 m w.e. 100 m$^{-1}$ and 0.31 m w.e. 100 m$^{-1}$ for 1979-2016, yielding a balance ratio (BR: the ratio of ablation to accumulation area balance gradients) of 3.10. A higher BR value implies that a smaller ablation area is needed to balance inputs in the accumulation area (Benn and Evans, 2014). The BR value simulated for Saskatchewan Glacier is rather high, i.e. triple that computed by Rea (2009) for the 'North America – Eastern Rockies' region (mean BR ± std = 1.11 ± 0.1). The simulated BR is within the range, but still on the high side, of values found for 'North America West Coast' glaciers (mean BR ± std = 2.09 ± 0.93) which have a more humid climate (Rea, 2009).

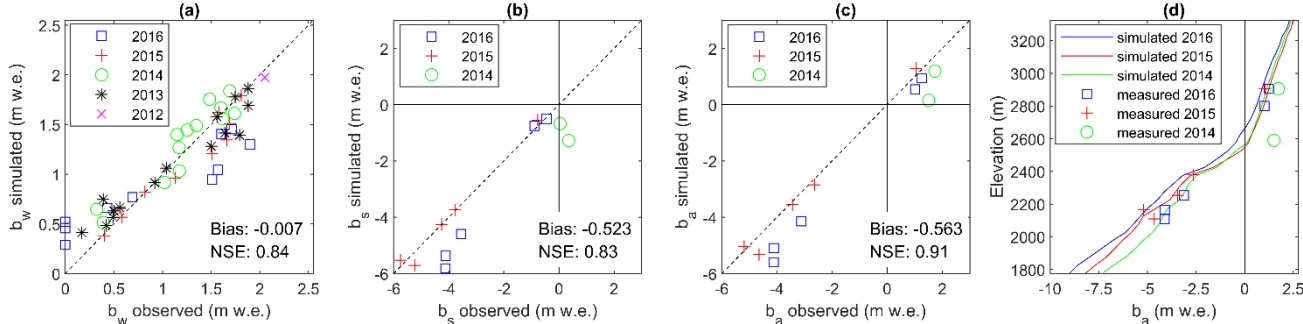

**Figure 6. Simulated mass balance compared with point mass balance observations available between 2012 and 2016. (a) Winter balance ($b_w$); (b) summer balance ($b_s$, only available since 2014); (c) annual balance ($b_a$, since 2014). The dashed line is the 1:1 relationship. (d) Simulated vs. observed annual mass balance gradient between 2014 and 2016.**

## 4.5 Mass balance reconstruction and comparison with geodetics estimates

The simulated annual specific (glacier-wide) conventional mass balance ($B_{a\_c}$) was overall negative throughout the period (mean = -0.72 m w.e. a$^{-1}$) with pronounced interannual variability (std = 0.57 m w.e. a$^{-1}$) (Figure 7a). The cumulative conventional mass balance simulated with the multitemporal DEMs agrees well with the geodetic estimates (Figure 7b). The simulated and geodetic cumulative mass balance were -26.79 m w.e. and -25.59 ± 8.44 m w.e., respectively, for 1979-2016. The cumulative error in the geodetic estimates increases in 1999 due to the large error in the SRTM DEM, even though it was coregistered to the high-quality WV2 2010 DEM (Supplementary Material).

The simulation with the 1979 reference DEM ($B_{a\_r1979}$), when the glacier was thicker and larger, results in a larger cumulative mass loss (~ -3 m w.e. over 37 years) than when using the 2010 DEM ($B_{a\_r2010}$) with the smallest historical extent (Figure 7b). The difference essentially arises from the larger extent in 1979 which provides more area available for melting at lower elevations. The conventional mass balance simulation remains between the limits of the two-endmember reference simulations, with a difference in cumulative mass loss of ~ ± 1.4 m w.e at the end of the period. The effect of dynamical adjustment was overall small from 1986 (first DEM update) onward (mean = 0.06 m w.e. a$^{-1}$), but accelerated over the last 15 years (Figure 7a).

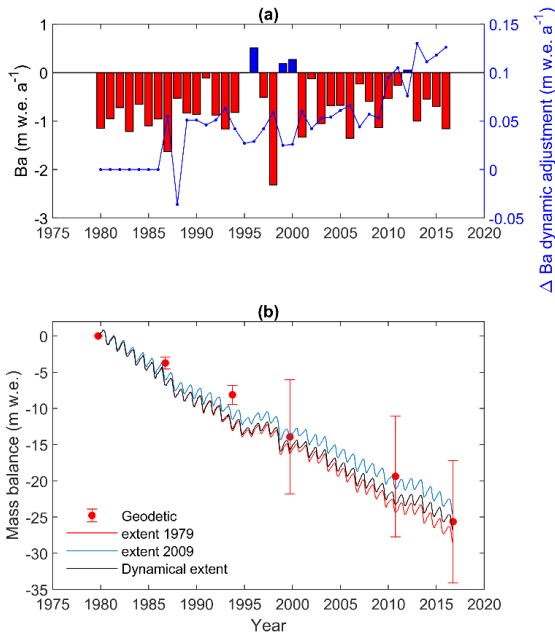

**Figure 7. Simulated mass balance compared with geodetic estimates. (a) Conventional annual glacier-wide mass balance ($B_{a\_c}$) from the dynamical simulation (multitemporal DEMs). The blue curve represents the effect of dynamical adjustment on $B_{a\_c}$. (b) Cumulative mass balance from reference (red: 1979, blue: 2010) and conventional (black: multitemporal DEM) simulations. Error bars represent one-sigma cumulative confidence intervals around the cumulative geodetic mass balance.**

**4.6 Model sensitivity to uncertainties in parameters and NARR forcings**

**4.6.1 Sensitivity to NARR interpolation and bias correction method**

Forcing the mass balance with the nearest NARR grid cell or with the bilinearly-interpolated NARR forcings resulted in negligible differences on the simulated cumulative balance, when both types of NARR forcings were bias-corrected by station observations (Figure 8a). However, when station data were not used for bias correction and the NARR precipitation and air temperature were only lapsed to the station elevations using the mean observed lapses rates, the simulated mass loss was overestimated relative to geodetic observations. However, the lapse rate-corrected and bilinearly interpolated NARR forcings resulted in a closer agreement with the geodetic observations than when using the lapse rate-corrected NARR forcings from the nearest gridcell.

**4.6.2 Sensitivity to NARR forcings**

The model sensitivity to the type of NARR variable used for forcing was investigated for the glaciological year 2014-2015, when both complete on-glacier AWS data and point mass balance were available (Figure 8b, c). Results show that the model was most sensitive to air temperature, whereas replacing the other NARR forcings ($RH_{NARR}$, $WS_{NARR}$, $G_{NARR}$) by their AWS counterparts had a comparatively small effects on the mass balance validation against observations. Hence, despite the good correlation between NARR and AWS air temperatures and the low errors following bias-correction (Figure 3), the model remains most sensitive to air temperature while it is less sensitive to other variables that showed comparatively higher errors with respect to AWS observations, such as wind speed (Figure 3).

**4.6.3 Sensitivity to model parameters**

The model parameter sensitivity analysis shows that the simulated mass balance was most sensitive to the uncertainty in the precipitation lapse rate ($\pm4\%$ $100$ $m^{-1}$) followed by the ice aerodynamic roughness length ($z_{0\_ice}$: $\pm1$ mm) (Figure 8d). The sensitivity to uncertainties in ice albedo ($\alpha_{ice}$: $\pm0.03$) and the snow aerodynamic roughness length ($z_{0\_snow}$: $\pm1$ mm) were smaller and of similar magnitude. Large changes in simulated cumulative mass balance occurred when considering an order of magnitude change on aerodynamic roughness lengths. While spatial variability in $z_0$ of that order is possible across a single glacier due to heterogeneous snow, and even more so on rougher ice surface morphology (Chambers et al., 2020), the resulting uncertainty on the glacier wide average $z_0$ would be much lower (Brock et al., 2006; Chambers et al., 2020; Munro, 1989). Nonetheless, these results clearly show that a careful assessment of the precipitation lapse rate and ice aerodynamic roughness length are crucial to derive a reliable long-term mass balance reconstruction. Constraining these two parameters as well as the

ice albedo and the snow aerodynamic roughness length against observations and ancillary information is thus pivotal to reliably

simulate the recent direct mass balance observations (Figure 6) and long-term geodetic estimates (Figure 8).

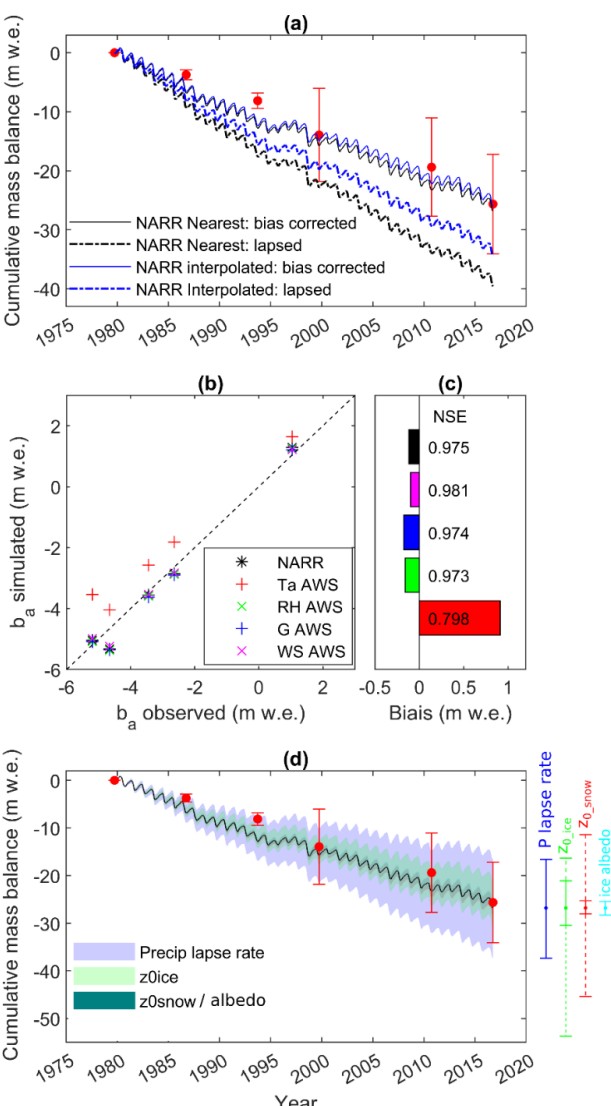

Figure 8. Model sensitivity to NARR forcings and model parameters uncertainty. (a) Sensitivity to NARR interpolation method (nearest gridcell: black, vs. bilinear interpolation: blue) and bias correction method (continuous line: bias corrected with AWS, vs.

stippled line: $P_{NARR}$ and $T_{a, NARR}$ lapse rate-corrected to the DEM). (b) Sensitivity to NARR forcings: measured vs. simulated point mass balance for glaciological year 2014-2015 after replacing NARR forcings ($T_{a, NARR}$, $RH_{NARR}$, $G_{NARR}$, $WS_{NARR}$) one at a time by AWS observations; (c) corresponding mean error (bias, m w.e., coloured bars) and Nash-Sutcliffe efficiency scores (labels). (d) Sensitivity to model parameter uncertainty. Coloured envelopes represent the cumulative uncertainty; coloured error bars on the right show the effect of parameter uncertainty on the cumulative mass balance in 2016. Error bars for ice (green) and snow (red)

roughness lengths correspond to a ± 1mm measurement uncertainty; the dotted error bars extend the uncertainty to ± one order of magnitude.

### 4.7 Energy and mass fluxes

Monthly energy balance shows that the sensible heat flux ($Q_S$) dominates energy gains throughout most of the year (Figure 9).
The contribution of $Q_S$ is fairly constant throughout the year, increasing only slightly in July-August and decreasing slightly in spring (March-May). The contribution of the net solar radiation flux ($SW^*$) increases systematically from low values in winter (November-February) when the sun angle is low and the glacier is covered by highly reflective snow, to peak values in July-August when the sun angle is high and low-albedo ice is exposed in the ablation area. Only in July and August does the net solar radiation ($SW^*$) becomes the dominant energy source. The latent heat flux ($Q_L$) is small over Saskatchewan Glacier, due to the generally high relative humidity (see Figure 2). $Q_L$ is positive on average and highest in summer, reflecting the predominance of deposition and condensation processes over sublimation. $Q_L$ represents a small, but non-negligible (7%) heat gain throughout the year, which reaches 11.5% in July-August. Energy loss occurs mainly by radiative cooling, i.e. through a negative net longwave radiation flux ($LW^*$). Lower air and surface temperature respectively reduce the incoming atmospheric longwave radiation and outgoing longwave emissions from the glacier surface, thereby reducing $LW^*$ in winter. $LW^*$ increases somewhat in summer (June-August), mainly because the glacier surface is near its melting point, limiting longwave radiation losses. The energy supplied by rain ($Q_R$) has a negligible influence on the energy balance. Melting ($Q_M$) predominantly occurs between May and October and peaks in July-August, due to the elevated $SW^*$, $Q_S$ and $Q_L$ fluxes, and radiative cooling ($LW\uparrow$) limited by the melting surface.

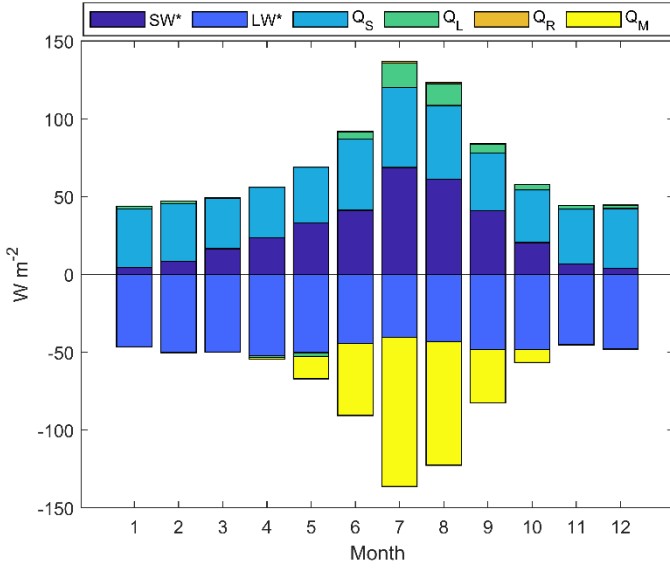


**Figure 9. Mean seasonal cycle of simulated surface energy balance on Saskatchewan Glacier between 1979-2016 from the multi-temporal DEM simulation. SW*: net shortwave radiation, LW*: net longwave radiation, $Q_S$: sensible heat flux, $Q_L$: latent heat flux, $Q_R$: rainfall heat flux, $Q_M$: energy used for melting.**

Four processes influence mass balance during the year (Figure 10a). Snowfall and snow accumulation dominate during the accumulation season (October-April). Melt mainly occurs from May to October and peaks in July-August in response to the positive surface energy balance (Figure 9). Deposition/condensation and sublimation fluxes are small. Net deposition predominates while net sublimation occurs in the spring (April-June), when there is high incoming radiation and the upper reaches of the glacier have not yet reached the melting point (Figure 10a). Although the $Q_L$ heat flux was found to be non-

negligible during summer (Figure 9), the resulting mass loss is itself negligible compared to melting because the latent heat of sublimation/deposition is seven times larger than that for melting. Moreover, the latent heat flux has a pronounced diurnal cycle, switching from deposition at night when cooling of moist air causes the vapour pressure to increase relative to the melting glacier surface, while daytime heating reverses the vapour gradient between the glacier surface and the atmosphere, causing sublimation (Figure 10b). Hence the two regimes tend to compensate each other but nighttime deposition slightly

dominates daytime sublimation, leading to a net positive deposition/condensation flux on average to the glacier surface.

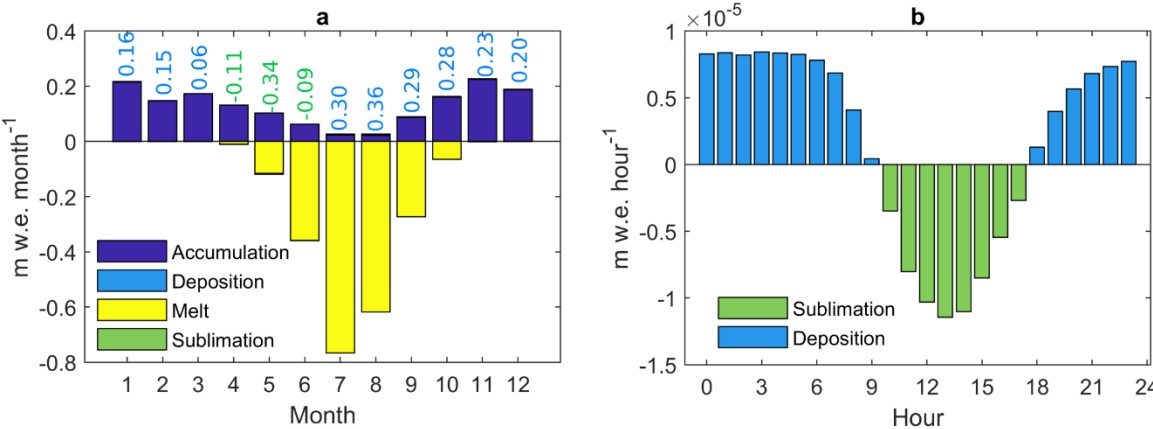

**Figure 10. Mean simulated mass fluxes on Saskatchewan Glacier between 1979-2016 using the multi-temporal DEMs. (a) Mean monthly fluxes; deposition and sublimation fluxes being much smaller, they are indicated as numbers in cm w.e. month$^{-1}$. (b) Mean**
**diurnal cycle in deposition/condensation and sublimation.**

Spatial patterns of the simulated reference mass balance ($B_{a\_r2010}$) (Figure 11) show an annual average snowfall of 1.54 m w.e over the glacier with a minimum of 0.30 m w.e near the toe, to ~3 m w.e. over the upper reaches. Annual melt can reach 7.86 m w.e. a$^{-1}$ at the glacier margin and 0.54 m w.e. a$^{-1}$ in the upper accumulation zone. Net deposition/condensation predominates
on average over the glacier, but fluxes are small (< 0.03 m w.e. a$^{-1}$), while net sublimation only occurs on the upper reaches of the glacier, mostly in the spring (Figure 11c, Figure 10a), corresponding to areas with high incoming solar radiation (Supplementary Figure S4). On average, melting losses (mean = -2.22 m w.e. a$^{-1}$) exceed snow precipitation gains (1.54 m w.e. a$^{-1}$) and the small condensation gain (mean = 0.01 m w.e. a$^{-1}$), yielding a mean negative reference annual balance ($B_{a\_r2010}$) of -0.67 m w.e. a$^{-1}$.


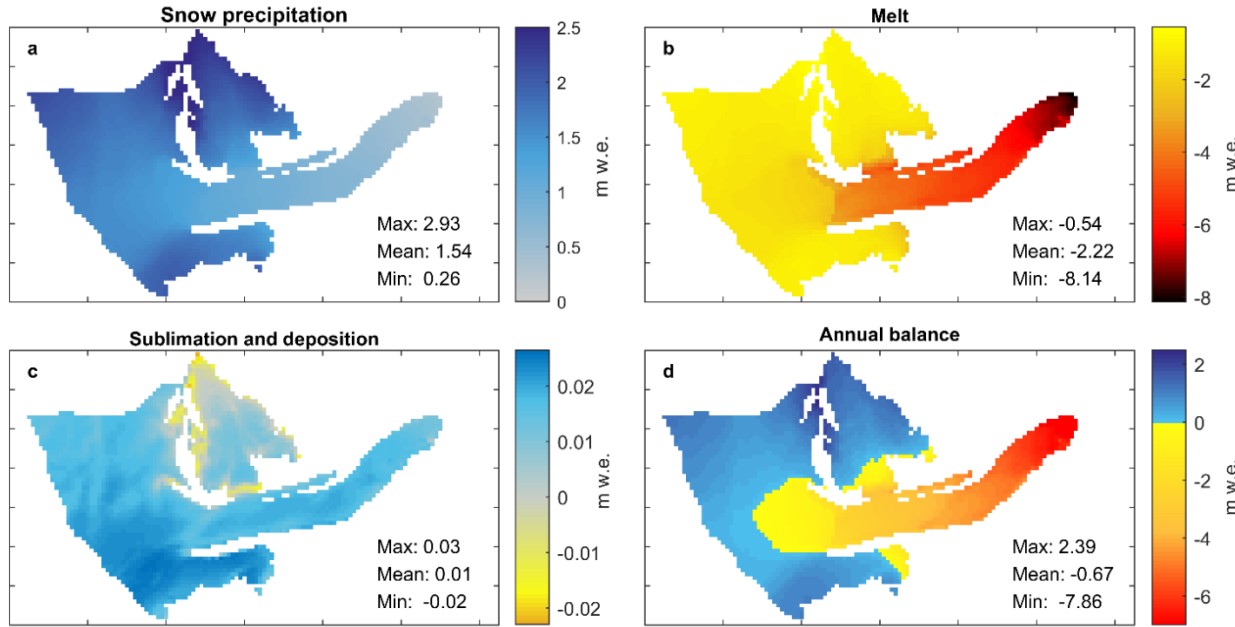

**Figure 11. Simulated average spatial patterns of reference annual mass balance ($B_{a\_r2010}$, in m w.e.) on Saskatchewan Glacier between 1979-2016. (a) Snow accumulation; (b) melt; (c) sublimation and deposition; (d) annual balance. The accumulation zone on (d) is delineated by the positive blue colour scale, the ablation zone by the negative yellow/red scale.**

**4.8 Climate sensitivity analysis**

The static sensitivity of mean mass balance ($B_{a\_r2010}$) components to climate perturbations ($\Delta T_a = 0$ to +7 °C and $\Delta P = -20$ to +20%) is shown in Figure 12. The reference scenario (1981-2010) yields an average annual mass loss of -0.68 m w.e. a$^{-1}$ (Figure 12c). The response surface for $B_{a\_r2010}$ shows that the glacier-wide mass balance is sensitive to changes in air temperature, and much less sensitive to changes in precipitation (Figure 12c). The $\Delta B_a$ contours also become steeper and narrower with increased warming, which indicates a reduced sensitivity to precipitation and increased sensitivity to temperature, respectively. The seasonal mass balance response sufaces help to understand the $B_{a\_r2010}$ sensitivities (Figure 12a, b). The $B_{w\_r2010}$ response surface shows that a precipitation increase of +20% can buffer the negative impact of warming on $B_w$ up to +3 °C of warming, but only up to +0.5 °C for $B_{a\_r2010}$. Moreover, a warming of more than +6 °C with no change in precipitation would supress net accumulation in winter, given the current glacier extent (2010) (Figure 12a). The sensitivity of winter mass balance to temperature changes also increases markedly with warming, as seen by the progresive tightening of the contours in Figure 12a. This is interpreted to result from decreasing accumulation due to the increasing shift from snowfall to rainfall and increased ablation during winter (Oct.-April) due to earlier disapearance of the snow cover under more pronouced warming. Conversely, the temperature sensitivity of summer mass balance ($B_{s\_r2010}$) increases only slightly with the warming scenario, and the steep contours in Figure 12b suggest a small sensitivity to precipitation changes. The increased temperature sensitivity of $B_{a\_r2010}$ with warming indicated in Figure 12c is therefore mainly attributed to decreasing accumulation from

reduced snowfall fraction and increased winter ablation as the climate warms and the snow cover retreats up-glacier earlier in the Spring (Figure 12a).

The IPCC RCP scenarios for the mid (2041-2070) and late (2071-2100) 21st century were overlain onto the response surfaces to show the most likely future climate trajectories. The RCP projection have significant uncertainities, as shown by their wide confidence intervals, and the annual mass balance change can vary by as much as ±3 m w.e. a$^{-1}$ within a single scenario. This illustrates the usefulness of scenario-free response surfaces to assess glacier mass balance sensitivity to climate as a background to evolving climate projections (Aygün et al., 2020b; Prudhomme et al., 2010). Nonetheless, given the current ensemble climate scenarios, the reference mass balance could decrease by -0.5 to -2.0 m w.e a$^{-1}$ by the mid-century, and by -0.5 to -4 m w.e a$^{-1}$ by the end of the century, relative to baseline conditions ($B_{a\_r2010}$ = -0.68 m w.e. a$^{-1}$) and depending on the RCP scenario considered.

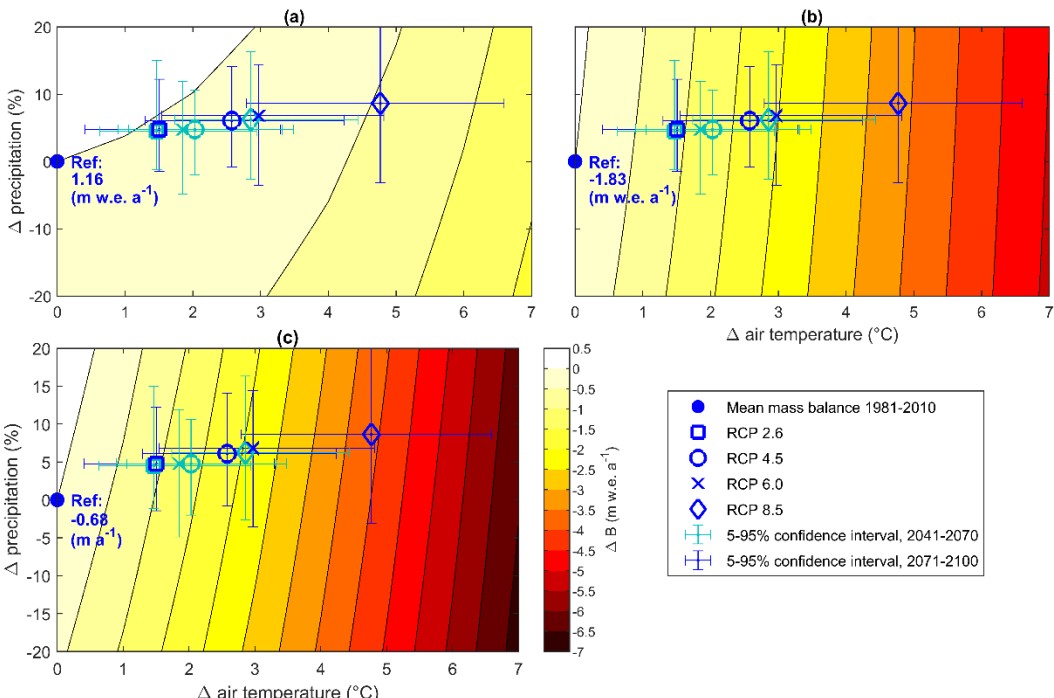

**Figure 12. Reference (2010) mass balance sensitivity to prescribed changes in regional mean air temperature between 0 to 7 °C and precipitation between -20 to +20%, which encompass IPCC RCP ensemble scenarios 2.6, 4.5, 6.0 and 8.5 for the mid (2041-2070: dark blue) and late (2071-2100: light blue) 21st century. The mean seasonal and annual mass balance are shown for the reference period 1981-2010. (a) Winter balance ($B_{w\_r2010}$); (b) summer balance ($B_{s\_r2010}$); (c) annual balance ($B_{a\_r2010}$).**

Since mass balance displays a large sensitivity to temperature and because glacier melt is the outcome of complex glacier-atmosphere energy exchanges, the sensitivity of energy and mass fluxes to warming alone was further investigated in Figure 13. The increasingly more negative mass balance in response to warming is dominated by increased melting (~93%), while increasing condensation/deposition accounts for ~ -3% (mass gain) of the net annual mass changes in response to warming (Figure 13a, b). Warming alters the precipitation phase, with the snowfall ratio decreasing non-linearly from 0.80 under present climate to 0.47 at $\Delta T_a$ +7°C (Figure 13c). This progressive conversion of snowfall to rainfall accounts for ~10 % of the mass changes in response to warming (Figure 13b).

The total energy input to the glacier surface increases with warming temperatures, and this energy surplus is predominantly used for melting ($Q_M$), which shows a non-linear increase with respect to warming (Figure 13d, e). Interestingly, the increase in energy supply with warming is mainly driven by an increase in net solar radiation ($SW^*$) and latent heat flux ($Q_L$), with more subdued increases in the temperature-dependent sensible heat ($Q_S$) and net longwave radiation fluxes ($LW^*$) (Figure 13e). Since cloud cover remained unchanged in the sensitivity experiments, the increase in $SW^*$ with warming is entirely driven by the decreasing albedo, as snow cover duration on the glacier decreases (Figure 13c). Since the relative humidity also remained constant in our sensitivity analyses, warming leads to higher atmospheric vapor pressures since the saturated vapor pressure of the air increases with warming. Since the glacier surface is contrained to the melting temperature (0 °C) during a large part of the year, the increase in surface saturated vapor pressure in response to warming will, on average, be less than that of the atmosphere, causing the vapor pressure gradient to increase and boost $Q_L$ fluxes (condensation/deposition) to the surface. Similar reasoning applies to $Q_S$, i.e., the near surface temperature gradient will increase in response to atmospheric warming. While the rainfall ratio increases with warming, its influence on the energy balance is insignificant (Figure 13e), but the reduced snowfall greatly impacts winter accumulation (Figure 12a). Increasing net solar radiation ($SW^*$) contributes from 51% to 42% of the increase in $Q_M$ ($\Delta Q_M$), with this contribution decreasing with warming. The contribution of $Q_L$ to $\Delta Q_M$ increases from 27 to 29% in response to warming, while that of $LW^*$ increases from 5 to 9%. The contributions of $Q_S$ (~19%) and $Q_R$ (~1%) are more constant across the warming spectrum (Figure 13f).

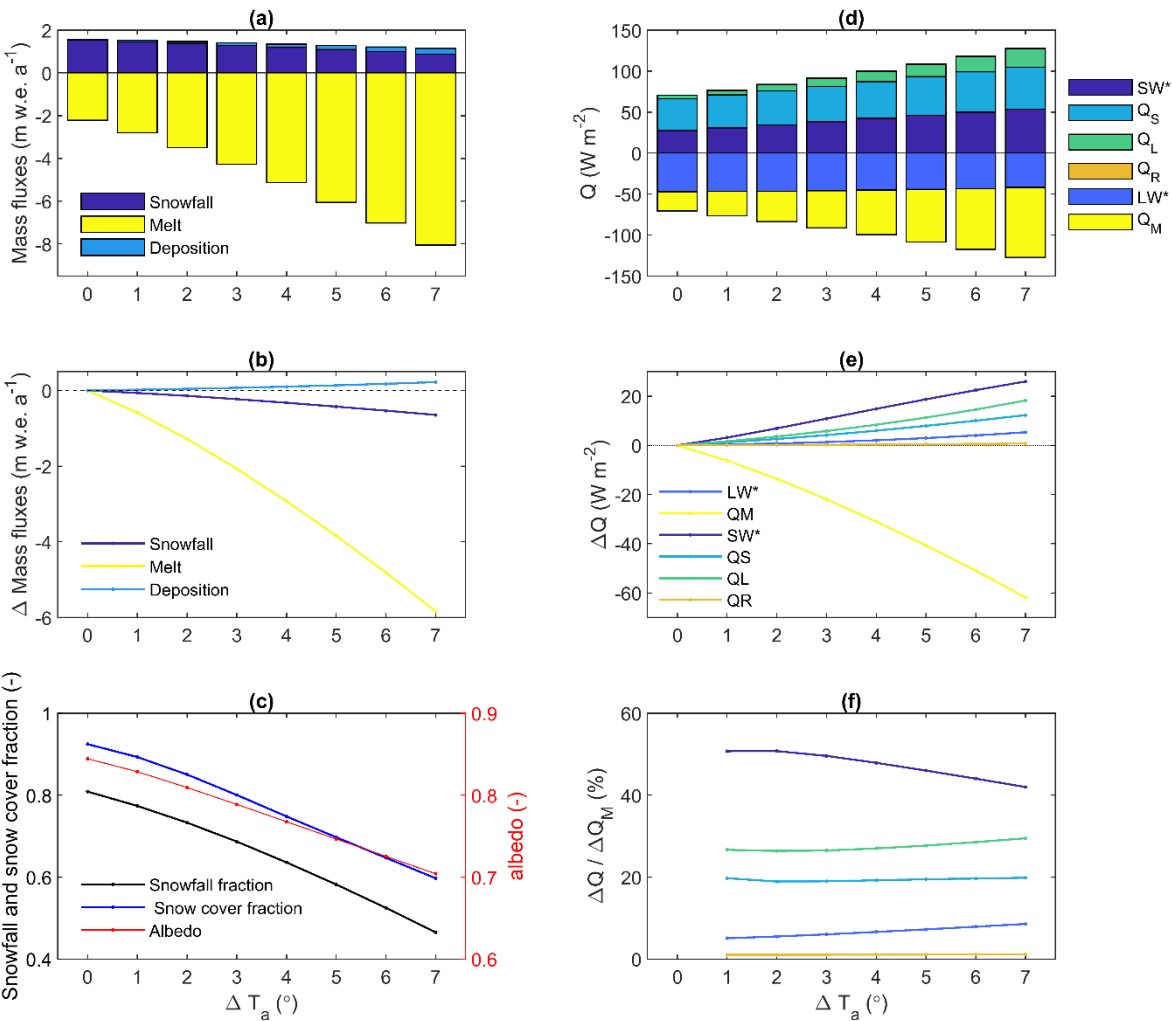

**Figure 13. Reference (2010) mass and energy balance sensitivity to changes in regional mean air temperature between 0 to 7 °C. (a) Annual mass balance; (b) changes in mass balance relative to baseline ($\Delta T_a = 0$); (c) changes in snowfall to total precipitation ratio, snow cover and albedo; (d) energy balance; (b) changes in energy balance relative to baseline ($\Delta T_a = 0$); (e) Changes in energy fluxes scaled by the changes in melt energy ($Q_m$). All fluxes and variables represent mean annual values averaged over the whole glacier**
**surface and over the baseline 1981-2010 period with mean air temperature perturbed from 0 to 7 °C.**

The results in Figure 13 allow apportioning the mass balance sensitivity to warming to four different processes (Table 1): (i) atmospheric warming, which causes an increase in the temperature-dependent fluxes ($\Delta LW^* + \Delta Q_S$) and contributes on average 24.3% to the mass balance sensitivity to warming; (ii) a precipitation phase change feedback, which contributes

10.3%; (iii) an albedo feedback, which contributes 44%, and (iv) a humidity feedback which contributes 22.3%. While the contributions from atmospheric warming and the humidity feedback increase with the level of warming, the precipitation phase feedback remains constant, while the albedo feedback decreases over time (Table 1).

**Table 1. Contribution of different processes to the sensitivity of glacier mass balance to warming from +1°C to +7°C.**

| Process | Equation | Relative contribution to $\Delta B_a$ | | | | | | | |
|---|---|---|---|---|---|---|---|---|---|
| | | +1°C (%) | +2°C (%) | +3°C (%) | +4°C (%) | +5°C (%) | +6°C (%) | +7°C (%) | **Mean (%)** |
| Atmospheric warming | $-(\Delta LW^* + \Delta Q_S)/(L_f \Delta B_a)$ | 23.0 | 22.7 | 23.3 | 24.0 | 24.9 | 25.7 | 26.5 | **24.3** |
| Precipitation phase change | $\Delta P_S/\Delta B_a$ | 10.4 | 10.3 | 10.3 | 10.3 | 10.3 | 10.3 | 10.3 | **10.3** |
| Albedo | $-\Delta SW^*/(L_f \Delta B_a)$ | 47.1 | 47.1 | 46.1 | 44.5 | 42.8 | 41.0 | 39.2 | **44.0** |
| Humidity | $\left(\dfrac{-\Delta Q_L}{L_f} + \Delta S\right)/\Delta B_a$ | 21.6 | 21.4 | 21.5 | 21.9 | 22.5 | 23.2 | 24.0 | **22.3** |

$L_f$ = latent heat of fusion, $P_S$ = snowfall, $S$ = deposition/condensation

## 5 Discussion

### 5.1 Suitability of NARR for model forcing

This study focused on reconstructing the mass balance of a glacier using a physically-based model constrained by a sparse set of glacio-meteorological data without calibration. This situation is common to many mountain glaciers of the world where logistical and financial constrains preclude continuous monitoring programs. In this context, the outputs of reanalysis products represent a useful alternative for driving glaciological models. Several previous studies have used reanalyses to force hydrological and glaciological models in mountainous region using statistical downscaling. Among the downscaling strategies used, some did not rely on situ observations, such as such as the linear theory of orographic precipitation used by Jarosch et al. (2012) and Clarke et al. (2015), and the extrapolation to the glacier surface of the vertical structure of air temperature in reanalysis products (Fiddes and Gruber, 2014; Jarosch et al., 2012). Hofer at al. (2010; 2012; 2015), on the other hand, used station-based downscaling and found that combining different types of reanalysis variables and spatial averaging of reanalyses gridcells led to superior downscaling performance. Earlier work by Radic and Hock (2006) used temperature and precipitation from ERA-40 reanalyses to force a mass balance model for Storglaciären, Sweden. They used bilinearly interpolation of the nine gridcells centered on the glacier and obtained good mass balance simulation results after correcting the ERA-40 temperature bias with a lapse rate tuned to optimise the mass balance simulation, but no correction on ERA-40 precipitation. Koppes et al (2011) also used a simple approach by regressing temperature and precipitation from the closest NCEP-NCAR reanalysis gridcell against station data in Patagonia. The station-based scaling approach used in this study to correct biases in NARR is simple compared to station-free (e.g., Jarosch et al., 2012) or multivariate regression (Hofer et al., 2010) approaches, but is similar to the station-based methods used by Radic and Hock (2006) and Koppes et al. (2011). The comparison between NARR and station observations was reasonably good (r = 0.31-0.98) given the short AWS record used for comparison. Three

variables ($T_a$, $RH$, $SW\downarrow$) showed strong correlations (r = 0.85-0.98) between NARR and AWS observations, and the simple scaling bias correction removed much of the biases present (Figure 3). Moreover, the cold bias in NARR air temperature was consistent with the elevation difference between the AWS and the NARR gridcell and the local temperature lapse rate. The low bias and high correlation for NARR air temperature and relative humidity, and solar radiation to a lesser extent (Figure 3), are consistent with previous findings from Trubilowicz et al. (2016) who showed that these variables agreed well with measured values at high-elevation stations in the southern Coast Mountains of British Columbia, Canada. Wind speed ($WS$) on Saskatchewan Glacier was however poorly represented (r = 0.37-0.38), most probably because thermal winds (katabatic and valley winds) are not represented at the coarse 32 km spatial resolution of the NARR. Trubilowicz et al. (2016) also reported lower and site-dependent accuracy for NARR wind speeds. More sophisticated wind downscaling (e.g. Vionnet et al., 2021; Wagenbrenner et al., 2016) could help improve further modelling at this site and other upland icefield-outlet valley glacier settings.

The positive bias in NARR precipitation was consistent with the higher elevation of the NARR grid point relative to the merged precipitation record (Figure 3). However, once the effect of the elevation difference was corrected using the calibrated precipitation lapse rate (15.6% 100 m$^{-1}$), the NAAR was found to underestimate observations by 10%. This is consistent with the recent study by Hunter et al. (2020) who showed that NARR underestimates precipitation in the mountain regions of British-Columbia, Canada. The NARR precipitation also correlated rather poorly with the off-glacier daily historical precipitation record (r = 0.30-0.31), showing that the daily variability in NARR precipitation is not well represented. Precipitation is notoriously more difficult to represent in reanalysis products, especially in complex terrain with steep orographic gradients and localized convective activity (Hofer et al., 2010; Mesinger et al., 2006). The station-free, linear orographic model for precipitation (LOP) method used by Jarosch et al. (2012) might perhaps be better suited than station-based downscaling in steep topography. The authors reported an improvement of the median relative error ($M$ = -3.1 to -20.9%) with respect to monthly precipitation totals in the Canadian Rockies, compared to raw NARR which underestimated station precipitation ($M$ = -9.5 to -42.6%). However, the median absolute error ($MAD$) of the relative error did not change much, and even increased in some instances, i.e. from 13.5-31.3% for the raw NARR compared with 19-29.5% for LOP (see table 3 in Jarosch et al., 2012). The station-based scaling used in this study resulted in $M$ = 3.8% and $MAD$ = 33%, compared to $M$ = 27% and $MAD$ = 41% for the raw monthly NARR precipitation. Hence the improvement seen is greater than that reported for the station-free LOP model by Jarosch et al (2012) in the Rockies.

Ebrahimi and Marshall (2016) also reported that the NARR precipitation for Haig Glacier, also in the Canadian Rocky Mountains, poorly represented the observed winter accumulation totals. Nevertheless, NARR precipitation has been found to be reliable at the monthly scale and to represent a useful input for hydrological modelling in North America generally (Chen and Brissette, 2017). Our results suggest this finding also applies for glaciological modelling, given that bias-correction is applied. The underestimation of precipitation in NARR combined with the positive bias in the raw NARR global radiation

mostly explain the exaggerated mass loss simulated by the mass balance model when forced with the lapsed NARR, i.e., when only precipitation and temperature are corrected to the elevation of the reference stations (Figure 8a). More elaborate topographic corrections of solar radiation (Fiddes and Gruber, 2014) could improve the downscaling of NARR solar radiation in the absence of ground observations, but precipitations biases remain difficult to correct in this situation.

The choice of the NARR spatial interpolation method for downscaling to stations had an overall small effect on the comparison with station data (Figure 3). The bias and RMSE were slightly reduced when using bilinear interpolation of nine gridcells, compared with the nearest gridcell method. However, following bias correction against station data, both interpolation methods resulted in similar cumulative mass balance simulation (Figure 8a).

Despite the low correlation between NARR and AWS wind speed, the simulated point mass balance in 2015 was not sensitive to using either the downscaled NARR or AWS wind speed forcings (Figure 8b, c). Replacing the downscaled NARR global radiation and relative humidity by their AWS counterparts had also a small effect on the model validation. Despite its strong correlation with AWS observations and low residual error after bias correction, the simulated point mass balance in 2015 was most sensitive to downscaled NARR air temperature (Figure 8b, c). This shows that air temperature have a large influence on
the energy balance calculations, and that the simple scaling correction could probably be improved to better represent the effect of the glacier wind on air temperature on the glacier (Shea and Moore, 2010).

The approach used in this study could be extended to other reanalysis products, especially the new global ERA5 reanalyses (Hersbach and Dee, 2016). While its spatial resolution (0.25° ~28 km) is only slightly finer than NARR (0.30° ~32 km), ERA5
has an hourly resolution compared to the 3-hourly NARR resolution.

## 5.2 Model performance and parameter sensitivity

Despite the physical nature of the model, some assumptions remain simplistic, such as a constant precipitation lapse rate and spatially invariant ice albedo, aerodynamic roughness and wind speeds. Despite these limitations, the interannual variability in mass balance was relatively well simulated by the model, with NSE values of 0.83 to 0.91 for direct point observations
(Figure 6). Point mass balance measurements with the glaciological method are affected by several uncertainties related to errors in ablation stake height measurements, stake self-drilling into the ice or firn and snow/firn density measurements (Zemp et al., 2013). Errors can range from 0.14 m w.e. a$^{-1}$ for ablation measurements on ice, 0.27 m w.e. a$^{-1}$ for ablation measurements on firn and 0.21 m w.e. a$^{-1}$ for snow measurements in the accumulation area (Thibert et al., 2008). The root mean squared error (RMSE) on the simulated $b_w$ was 0.24 m w.e. a$^{-1}$ (median relative error of 15%) – on the same order as the typical measurement
error for snow and firn.  RMSE values, however, were higher than typical measurement errors for $b_s$ (0.87 m w.e. a$^{-1}$, relative error = 22%) and $b_a$ (0.77 m w.e. a$^{-1}$, relative error = 24%), due in part to the restricted number of available observations for

validation (Figure 6). The reconstructed mass balance also compared favorably against the independent geodetic estimates (see sect. 4.5 and Figure 8). The simulated cumulative mass loss (-26.79 m w.e.) was close to the geodetic estimate (-25.59 ± 8.44 m w.e), despite the large uncertainties in the geodetic balance introduced from 2000 onward due to vertical uncertainties in the SRTM DEM. The long-term consistency between geodetic and modelled mass balance gives further confidence that the bias-corrected NARR forcings do not suffer from systematic biases.

The model sensitivity to uncertain model parameters showed that the simulated mass balance was most sensitive to uncertainties in the precipitation lapse rate, followed by the ice aerodynamic roughness, while the sensitivity to the snow aerodynamic roughness and ice albedo were lower. This demonstrates that the precipitation lapse rate must be carefully evaluated using ancillary meteorological data, which can be difficult in regions with no permanent weather station network nearby, a conclusion also reached for the Himalayas by Immerzeel et al. (2014). As the model only accepted a constant lapse rate, we used a value (15.6 ±4%) representative of the period during which most of the snow accumulation occurs, i.e. when the glacier toe is above the zero-degree isotherm (Supplementary Figure S1). Including shoulder months (April and September) which have mixed precipitations would slightly lower this gradient. The extrapolation of the gradient beyond the highest weather station (2000 m) is also a common but hazardous practice, and validation against snow courses (Avanzi et al., 2021) or winter mass balance surveys (Carturan et al., 2012) offers a way to check the validity of the gradient. The gradient used in this study resulted in accurate simulations of winter mass balance (Figure 6a), which strengthens our confidence in extrapolating the gradient to the glacier. However, there were no observations beyond 2900 m to constrain the gradient further. The area of the glacier above 2900 m represents only 8.8 % of the total area, so extrapolation errors in this unsampled area would have a small impact on the glacier-wide mass balance. Further development of the model should also consider including a time-varying precipitation lapse rates, as it was shown for example that the lapse rates was smaller during the 2016 dry year (Figure 6a).

A high sensitivity to the ice aerodynamic roughness has been reported in several studies (e.g. Brock et al., 2000; Hock and Holmgren, 1996; MacDonell et al., 2013; Munro, 1989). It remains one of the most challenging parameters to constrain in glacier mass balance models, and the assumption of a spatially and temporally constant $z_0$ value is a simplistic representation of reality (Fitzpatrick et al., 2019). This parameter is indeed often calibrated in the absence of direct observations (Hock, 2005). In this study, observations from the nearby Peyto Glacier allowed using a representative value which yielded good results; however the uncertainty range in the values reported by Munro (1989) (± 1 mm) was sufficient to induce a ± 17% error in the simulated cumulative balance (Figure 8d). Advances in deriving aerodynamic roughness from remote sensing could help in the future to improve the calculation of turbulent fluxes in distributed glacier models (Chambers et al., 2020; Fitzpatrick et al., 2019; Smith et al., 2020). The use of remotely sensed albedo maps also contributed to constrain a representative value for ice albedo (see Sect. 3.3.2) but the simulated mass balance was not very sensitive to the uncertainty around this estimate (Figure 8d). Nevertheless, only an average value was used, when in fact significant heterogeneity was found within the ablation zone

(Supplementary Material). Decreasing ice albedo can occur over the course of the melt season due to impurities of geogenic origin concentrating at the surface (Cuffey and Paterson, 2010), cryoconite development (Takeuchi et al., 2001) and more discrete events not taken into account in the model, such as algal mat development (Lutz et al., 2014) or wildfires that bring black carbon and ash onto the glacier and decrease the albedo (Marshall and Miller, 2020). Long-term darkening has also been observed on glaciers of the European Alps, which questions the use of fixed albedo values in long historical and future mass balance simulations (Oerlemans et al., 2009). Further efforts could look to assimilate such remotely sensed albedo maps within distributed models.

### 5.3 Impact of glacier recession on mass balance

The multi-temporal DEMs used in the study allowed quantifying the impact of glacier elevation changes on long-term mass balance (Figure 7). The conventional mass balance simulation with the multitemporal DEMs showed a maximum difference of ~1.5 m w.e., or 5.6% of the 1979 reference cumulative balance. This is a small difference overall, which shows that glacier recession has had a minor impact on the mass balance of Saskatchewan Glacier. This is expected for this setting in particular, since the glacier margin is at the bottom of the occupying valley and glacier retreat has occurred over a restricted elevation range – thereby limiting negative feedback effects between glacier retreat and mass balance. This study has focused on the static climate sensitivity of mass balance, which ignores future dynamical feedbacks. Static, or reference mass balances calculated over a constant glacier hypsometry have been proposed to be better suited for climatic interpretation (Elsberg et al., 2001; Harrison et al., 2009). But from a hydrological perspective, future glacier retreat towards higher elevations would mitigate an increasing portion of the simulated mass loss, gradually increasing the difference between the reference (2010) and conventional mass balance, and progressively decreasing the volume of meltwater released annually (Huss and Hock, 2018; Huss et al., 2012). An increase in dynamical adjustments effects on mass balance was already visible on Saskatchewan Glacier from 2000 onward (Figure 7a).

### 5.4 Energy balance regime

The simulated glacier-wide energy balance regime of Saskatchewan Glacier showed that energy inputs are dominated by the sensible heat flux, flowed by net radiation and latent heat fluxes. This is different than commonly reported for mid-latitude glaciers in continental climates, where net radiation dominates over turbulent fluxes (e.g., see compilation by Smith et al., 2020). However, most studies reporting energy flux partitioning relied on summer observations in the ablation zones of glaciers. Hence, the often-reported high contribution of net radiation to melting energy is biased by the season (values are commonly reported for July/August – when net radiation is high), and to the ablation zone of glaciers, where most micrometeorological studies have been done and where again net radiation is higher due the lower albedo. Year-round, glacier-wide values are rarely published, and only available from distributed energy balance models. On Saskatchewan Glacier, the glacier-wide contribution to melting energy was 26.1% for net radiation, 57% for sensible heat ($Q_S$) and 16.9% for the latent

heat flux ($Q_L$) during the ablation period (July-August). The energy partitioning was quite different when looking at the ablation zone only, with net radiation contributing 57%, $Q_S$ 32%, and $Q_L$ 11% of the melting energy at the AWS, midway up the ablation zone. The lower glacier-wide contribution of net radiation reflects the fact that much of Saskatchewan Glacier is

920 covered in snow, and later firn, in summer. This is also accordance with Klok and Oerlemans (2002), who showed that net radiation dominates over $Q_S$ in the lower part of the glacier, while $Q_S$ dominates in the higher part of Morteratschgletscher Glacier, Switzerland. Studies that reported glacier-wide energy partitioning include Storglaciären in Sweden (summer net radiation: 38-57%, $Q_S$: 42% $Q_L$: up to 17%) (Hock and Holmgren, 2005), Brewster glacier, New-Zealand (annual net radiation: 45%, $Q_S$ + $Q_L$: 52%, turbulent fluxes dominating in summer) (Anderson et al., 2010), Haut Glacier d'Arolla, Switzerland

(summer net radiation : 82%, $Q_S$: 25%) (Arnold et al., 1996), Donjek range glaciers in the Yukon, Canada (summer net radiation : 60-83%, $Q_S$: 20-45%, $Q_L$: -4 to -9%) (MacDougall and Flowers, 2011), and Haig Glacier in Alberta, Canada (summer net radiation: 70%, $Q_S$: 30%) (Marshall, 2014). Point measurements in late June/early July on nearby Peyto Glacier showed that net radiation contributed 63% and 42% of the melt energy over ice and snow surfaces, respectively, while sensible heat contributed 34% (ice) and 50% (snow) (Munro, 2006). Hence, the contribution of sensible and latent heat flux to summer

melting on Saskatchewan Glacier is higher than common values for mid latitude temperate glaciers with a continental climate, and closer to that encountered for glaciers in more humid climates (Anderson et al., 2010; Smith et al., 2020). The contribution of turbulent fluxes to melting energy was however not so different from the earlier measurements reported by Munro (2006) at Peyto: 32-57% at Saskatchewan vs. 34-50% at Peyto for $Q_S$, and 11-17% vs. 3-9% for $Q_L$. The higher contribution of turbulent fluxes to melting on Saskatchewan Glacier, together with a simulated balance ratio (BR = 3.10, see Sect. 4.4.1) that

is closer to values from more humid climates (Rea, 2009), may thus reflect the locally wetter and cloudier climate, and high accumulation rates resulting from the efficient interception of moist polar maritime air masses from the west by the high and extensive plateau of the Columbia Icefield (Demuth and Horne, 2018; Tennant and Menounos, 2013). This 'icefield weather' frequently wraps the Columbia icefield in clouds while surrounding valleys are cloud-free.

## 5.5 Climate sensitivity

The simulated mass balance sensitivity to a +1°C warming was -0.65 m w.e. $a^{-1}$ °$C^{-1}$. This value is comparable to other mid-latitude glaciers: -0.60 m w.e. $a^{-1}$ °$C^{-1}$ for the Illecillewaet Glacier in the Selkirk Mountains of British Columbia (Hirose and Marshall, 2013), -0.66 m w.e. $a^{-1}$ °$C^{-1}$ for the Haig Glacier in the Canadian Rocky Mountains (Ebrahimi and Marshall, 2016), -0.65 ±0.05 m w.e. $a^{-1}$ °$C^{-1}$ for small (<0.5 km$^2$) glaciers in Switzerland (Huss and Fischer, 2016), -0.60 m w.e. $a^{-1}$ °$C^{-1}$ for the larger Morteratschgletscher, Switzerland (Klok and Oerlemans, 2004), and -0.61 m w.e. $a^{-1}$ °$C^{-1}$ for Storglaciären, Sweden

(Hock et al., 2007). Higher sensitivities are found in more humid climates, e.g. -0.86 m w.e. $a^{-1}$ °$C^{-1}$ for the South Cascade Glacier, Washington (Anslow et al., 2008) and up to -2.0 m w.e. $a^{-1}$ °$C^{-1}$ on Brewster Glacier, New Zealand (Anderson et al., 2010), and lower sensitivities in drier climate, e.g. -0.44 m w.e. $a^{-1}$ °$C^{-1}$ on Urumqi River Glacier No.1 in the Chinese Tien Shan (Che et al., 2019). Earlier work by Braithwaite (2006), Oerlemans and Fortuin (1992) and Oerlemans (2001) showed that

the mass balance sensitivity to temperature scales with mean annual precipitation, due to larger albedo and precipitation phase feedbacks and longer melt seasons on glaciers in wetter climates.

We found that the albedo feedback is the main contributor (mean = 44%) to the temperature sensitivity of mass balance on Saskatchewan Glacier (Figure 13). Increases in net shortwave radiation caused by a reducing snow cover and ensuing reduced glacier albedo account for 39-47 % of the increase in melt energy across the various warming scenarios (Table 1). A similar finding was reported on Haig Glacier by Ebrahimi and Marshall (2016), who found that introducing albedo feedbacks doubles the net energy sensitivity to warming. This value (44%) is significantly high, but less than the 80% reported recently by Johnson and Rupper (2020) for the summer-accumulation type Chhota Shigri Glacier in High Mountain Asia. As shown by Fujita (2008), higher sensitivities are found for glaciers located in a summer-precipitation climate, where albedo feedbacks on ablation are stronger, than for glaciers located within a winter-precipitation climate. Atmospheric warming itself contributed only 24.3% to the mass balance sensitivity to temperature across all warming scenarios, through sensible heat and longwave radiation transfer to the glacier (Table 1). A significant air humidity feedback was also found, with latent heat fluxes representing an average of 22% of the temperature sensitivity across all warming scenarios. Keeping the relative humidity constant under warming scenarios may be plausible for the high elevation Columbia Icefield. The icefield receives moist air masses from the British Columbia interior and the Pacific Ocean uplifted onto the icefield, as the region is subject to upslope conditions derived from convergent upper air masses as low-pressure systems spin by south of the region. Under a stable atmospheric moisture regime, increasing atmospheric warming would lead to an increasing humidity feedback on ablation (Table 1). Other glaciers subjected to subsiding air masses could experience drier weather in the future, which would decrease their melt sensitivity to warming (Ebrahimi and Marshall, 2016). The large contribution of latent heat fluxes to melting under warming scenarios points to the necessity of considering changes in specific air humidity when simulating glacier melt under future climates. This conclusion is in line with the recent findings by Harpold et al. (2018) who showed that atmospheric humidity plays a critical role in local energy balance and snowpack ablation under warmer climates, with latent and longwave radiant fluxes cooling the snowpack under dry conditions and warming it under humid conditions. The precipitation phase feedback, on the other hand, contributed the remaining 10% of the mass balance temperature sensitivity (Table 1).

The mass balance sensitivity of Saskatchewan Glacier to a ±10% change in precipitation under the current temperature regime was 1.01 (unitless: m w.e. of mass change per m w.e. of precipitation change). A value of 1 would occur if all precipitation were snowfall and there were no albedo feedbacks on $B_a$. The snowfall fraction being 0.81 under the present climate (Figure 13c), the albedo feedback on ablation contributes 19% to the mass balance sensitivity to precipitation. With the mean annual precipitation on the glacier being 1880 mm for the 1981-2010 reference period, the maximum +20% precipitation increase projected from ensemble climate scenarios for the end of the century would add a maximum of 0.4 m w.e. a$^{-1}$ if all new precipitation falls as snow, which is small compared to the mean temperature sensitivity of -0.65 m w.e. °C$^{-1}$. As such, precipitation increases can only buffer up to +0.5 °C$^{-1}$ of warming on Saskatchewan Glacier. As warming causes snowfall to

shift to rainfall at a rate of ~5% °C$^{-1}$ (Figure 13c), the buffering effect of a +20% increase precipitation would decrease accordingly, i.e. from 0.29 m w.e. a$^{-1}$ for a +1°C warming to 0.17 m w.e a$^{-1}$ for a +7°C warming.

## 6 Conclusions

Despite their physical basis, energy-balance models often struggle to replicate mass-balance observations, due to the difficulty in constraining their numerous parameters and obtaining reliable meteorological forcings (Gabbi et al., 2014; Réveillet et al., 2018). Our study showed that a physically-based, distributed mass balance model forced by regional reanalysis data can adequately reproduce the recent and long-term evolution of glacier mass balance when forcings and key model parameters are judiciously constrained with available observations and ancillary data. This is a key requirement for the effective application of such models, since parameters from distributed energy balance models dot not necessarily transfer well between sites (MacDougall and Flowers, 2011). While reanalysis data can provide realistic climate forcings for glacier models, bias-correction with in situ observations remains ideal when such measurements are available. Adopting this approach, however, entails a significant amount of work, which would be hard to implement at the mountain range scale. While ancillary data were key to constraining key model parameters, model sensitivity analyses showed that the precipitation gradient and the aerodynamic roughness lengths were sensitive parameters that need to be carefully prescribed.

The reconstructed mass balance of Saskatchewan Glacier shows a cumulative loss of -26.79 m w.e. over the period 1979-2016, in good agreement with independent geodetic estimates (-25.59 ± 8.44 m w.e). Glacier retreat has had a small impact overall on glacier mass balance, but the effect of dynamical adjustment has been increasing in recent years. Climate sensitivity experiments showed that future changes in precipitation would have a small impact on glacier mass-balance, while the temperature sensitivity increases with warming, from -0.65 to -0.93 m w.e. °C$^{-1}$. Increased melting accounted for 90% of the temperature sensitivity while precipitation phase feedbacks accounted for 10%. Close to half (44%) of the mass balance response to warming was driven by reductions in glacier albedo as the snow cover on the glacier thins and recedes earlier in response to warming (positive albedo feedback). Atmospheric warming directly accounted for about one quarter (24%) of the mas balance sensitivity to warming. The remaining mass balance response to warming was driven by latent heat energy gains (positive humidity feedback) and conversion of snowfall to rainfall (positive precipitation phase feedback). Our study therefore underlines the key role of albedo and air humidity in modulating the response of winter-accumulation type mountain glaciers and upland icefield-outlet glacier settings to climate.

## 7 Code availability

The glacier mass balance model code is available at https://regine.github.io/meltmodel/

**8 Data availability**

Downscaled NARR forcings, geodetic mass balance estimates and reconstructed mass balance are available from the corresponding author.

**9 Author contribution**

Conceptualisation: CK, MND. Formal analysis: CK, OL; Supervision: CK; Data Curation: MND, BM; Writing – original draft preparation: OL, CK; Writing – review & editing: CK, OL, MND, BM.

**10 Competing interests**

The authors declare that they have no conflict of interest.

**11 Acknowledgements**

CK was supported and funded by the Fonds de recherche du Québec – Nature et Technologies, the National Sciences and Engineering Research Council of Canada and the Canada Research Chair Program. MND was supported and funded by Natural Resources Canada's Climate Change Geoscience Program and the NSERC-CCAR Changing Cold Regions Network (CCRN). The authors thank Steve Bertollo, Eric Courtin, May Guan, Gabriel Meunier-Cardinal and Anthony Pothier-Champagne for assisting with field data collection. Jasper and Banff National Parks of Canada are thanked for supporting this work under Permit JNP-2010-4694.

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
