# Peer review of "Modelling glacier mass-balance and climate sensitivity in a context of sparse observations: application to Saskatchewan Glacier, western Canada"

_The Cryosphere, 2021_

## Referee Comment (RC1)

**General comment:**

The paper presents the use of reanalysis data to drive a distributed, physically-based modelling endeavor of the Saskatchewan Glacier. The authors cleverly combine a very limited on-site AWS record with longer term regional AWS records, and NARR product for multiple variables to model the long term (1979-2016, and 21st century climate perturbations) mass balance of the glacier. The analysis includes geodetic estimates of glacier change and point-scale estimates of mass balance, that are used to further constrain the model and investigate its performance. These results are used to assess the glacier's 21st century climate sensitivity across projected temperature and precipitation changes from the IPCC RCPs.

I found it challenging to follow the organization of the scientific information and I believe the manuscript should be significantly reworked. The text contains grammar and spelling mistakes that need to be addressed. The authors should clarify their use of terminology (i.e. 'altitude' vs 'elevation' are used interchangeably), homogenize variable names throughout the text (i.e. Ta for all air temperatures derived from NARR, from the regional AWS network, and the downscaled model input), and use consistent units (i.e. melt reported in mm w.e., cm w.e., and m w.e. at different stages). The information pertaining to the methods is disorganized causing difficulty in deciphering the approach (with omissions of details about downscaling key variables such as temperature and wind speed to the distributed glacier grid), and at some points systematic misuse of terminology ('downscaling' vs 'bias correction' vs 'extrapolation to model grid') obfuscates the underlying novelty of the approach to designing a spatially-distributed application of a physically-based melt model in a setting with limited input data.

Additionally, the use of reanalysis data from a single NARR grid node as model forcing is not adequately justified. The authors convincingly demonstrate that these data perform well in comparison to observational records of climatological variables. However, without additional information on the performance of either nearby nodes or an aggregate of these, it is impossible to determine whether this convincing performance is due to a modelling workflow which leverages a reanalysis dataset that is transferrable to other glaciers (especially those that are large enough to intersect multiple grid cells, or small enough and less fortunate in their geography to be equidistant from many grid nodes), or simply due to the coincident position of Saskatchewan Glacier relative to the nodes of the reanalysis product used. Some form of evaluation pertaining to the regional potential of the NARR product for direct incorporation into this melt-model workflow would help to better highlight this paper's relevance to cryospheric research.

For further context, I would highlight the interesting use of the melt modelling results to assess the sensitivity of Saskatchewan Glacier, specifically in extracting meaningful process-based explanations from the glacier's energy balance. These are obviously locally significant, and perhaps regionally so, but it is difficult to address the true scale of relevance of these results without significant improvements to the clarity and organization of writing, and rigor of the analysis. These improvements, combined with a more focused introduction presenting exactly the purpose of the study would greatly help in highlighting the potential use of the method for data-poor glacier settings. Additionally, improvements to clarity and organization would ensure the approach is reproducible, and as such, transferable to others.

**Specific comments:**

Section 1:

The introduction contains very detailed information on several topics related to the analysis that distract from a well-formulated motivation for this work. I would recommend using more direct wording to improve the readability of the text (i.e. 'the amount of precipitation remains unchanged' vs 'precipitation remains unchanged'; line 34, 'the so-called 'temperature index' or degree days' models vs 'temperature index models'; line 59). I would also recommend reducing the specificity of information on topics both tangentially and directly related to the paper topic.

With respect to tangential information: the details about GIC mass loss (lines 45-50) seem incongruous with the paper topic and the opening summary of glacier-freshwater relationships, details about the Canadian Rockies (lines 97-117) could be shortened by moving some information to the study area section.

With respect to direct information: detailed information on the temperature index to energy balance spectrum of melt models could be significantly condensed (lines 59-77), information on reanalysis products could be limited to NARR (lines 83-90), and information on their downscaling could be limited to the methods relevant to the study approach (lines 90-95).

The stated objectives highlight the 'comparatively unexplored' topics of ablation vs precipitation phase feedbacks to temperature sensitivity (lines 119-124) but offer no literature to situate this statement or the study. Please clarify what exactly is novel about this study's results and situate it in existing literature to really highlight the novelty of the study. The text at the very end of the paper (line 810-811) could be tied back to a statement here as it is more specific.

Section 3:

The data and methods section requires significant redesign. Section 3.2.3 contains most information pertaining to reanalysis data, and all pre- and post- processing associated with bias correction and downscaling across 5 climate variables with no internal divisions. Additionally, information on downscaling across the glacier model grid in a distributed manner is contained within the mass balance model section, which makes it hard to find and follow the logic of the approach. I would recommend a clear division of data (sources, types, any pre-processing strategies) and methods used in manipulating these data to create model inputs (downscaling, bias correction), followed by information specific to the mass balance model once all input variables (to the distributed model) have been presented. I like the authors strategy of presenting each element of the energy balance, but this could be shortened significantly if information relative to model input pre-processing were already made clear beforehand.

Additionally, the exact workflow used is very difficult to decipher. The authors first obtain NARR data from a node closest to the study area, then they bias correct these long-term timeseries of climatological variables (1979-2016) using a two year on-ice AWS record (excepting precipitation), then downscale the bias corrected NARR data to the model grid using lapse rates derived from regional AWS records. In the text, the terms bias correction and downscaling are used synonymously for work performed at the on-ice AWS location (lines 238-239), and the precipitation variable downscaling step is described as

'extrapolating to the model grid' in the mass balance model section (lines 262-263). See my technical comments for citations that may help in differentiating these terms. It is necessary to address the confusion in usage of terms such as 'downscaling', 'extrapolation', and 'bias correction', and distinguish between the steps for the transition of NARR climatological variables from the grid cell timeseries, to the AWS location timeseries, to the distributed model grid timeseries ultimately used to force the mass balance model for each variable, in order for this workflow to be reproducible.

Section 4:

The use of a single NARR grid node to drive the entire modelling study must be evaluated further. The authors demonstrate that temperature, solar radiation, and relative humidity perform well compared to a 2 year on-ice AWS record. My concerns and related recommendations here are two-fold:

(1) There needs to be evidence that demonstrates how the use of this grid node is justified compared to other regional grid nodes (for example, at better situated elevations), and/or a regional aggregate of NARR data, and/or using bilinearly interpolated NARR data at the AWS location (given that the nodes define a continuous field across the region of interest). Some recommendations include: commenting on differences between the coarse NARR topography (derived from surface geopotential variable in NARR for example) to credit/discredit the use of information at the AWS location, a sensitivity test of the model using a different grid node(s) and/or a regional aggregate of nearby grid nodes as forcing, and/or using AWS location bilinearly interpolated values to assess the geographic sensitivity of point-scale data as forcing (and assessing the relative cost/benefit of introducing interpolation to this approach).

(2) As it stands the novelty of this method seems limited by the possibility that it is only applicable to glaciers with nearby reanalysis grid nodes. Given that this study investigates a single glacier in the Canadian Rockies, it would greatly improve the significance of the work to demonstrate that this method can reliably inform physically-based models on other glaciers with limited AWS forcing data. To do this however, the above (1) needs to be addressed.

Throughout:

Methods details are missing:

- Temperature downscaling using lapse rates to the model grid is never explicitly detailed.
- The sensitivity of the model to poor wind forcing input is never evaluated (the downscaling of this variable to the model grid is not addressed and I am unsure if the AWS bias corrected NARR values are used to force the entire glacier surface or if these data are further downscaled to be distributed across the finer model grid). If only one wind speed timeseries is used to drive the entire distributed energy balance model, this strategy must absolutely be justified with supporting literature given a lack of distributed on-ice wind data to evaluate local variability. I do not have sufficient expertise to comment on the limitations of this in detail, but it seems important to demonstrate that this is a valid strategy for forcing wind speed, given how it fails to perform well due to the presence of katabatic and sloping wind signals in the AWS record (table 1), and the fact that WS factors heavily into the discussion of results relating to the energy balance of the glacier. See technical comment for more information.

- Smaller details of the workflow are missing (i.e. filling of precipitation record data gaps, see technical details) and are needed to ensure the study results are reproducible and the method transferable to other settings.

Presentation of results could be improved:

- Units change frequently throughout; this is distracting and must be made consistent.
- Please assign a unique term to each variable. For example, temperature that has undergone different steps of processing should have a different subscript, terms presented at one stage of the paper should be consistent throughout (i.e. Qh changes to Qs for sensible heat flux in methods/results), and variables in equations should have the same notation as those in the text that defines them. Whenever possible, variables relating to mass balance should be consistent with the mass balance glossary:
  Cogley, J. Graham, et al. "Glossary of glacier mass balance and related terms." *IHP-VII technical documents in hydrology* 86 (2011).
- Occasionally, statements are inaccurate of the literature cited for support, and the work of others is improperly cited. See technical comments.

**Technical/detailed comments:**

The abstract would benefit from being more focused on the paper's key results: removing information on the static vs dynamic extent performance (lines 20-21), details surrounding IPCC RCPs (lines 23-24), and details of the melt vs sensitivity (lines 26-27, 29-30), would both shorten the text and emphasize the key findings.

Lines 23-24: 'IPCC representative concentration pathways climate scenarios' is redundant, either climate scenarios or RCPs

Lines 44-49: Very dense information on SLR specific to GICs can be condensed to remain specific to small mountain glacier study

Line 51: Sentence lacks specific information, or citation if the importance of mass balance variable is necessary to argument.

Lines 52-53: 'Only a few' -> list how many. If this information does not exist, a citation pointing what is known would be appropriate. Time-consuming better described as resource intensive.

Lines 59-62: 'So-called 'temperature-index' or 'degree-day' models' is redundant, can just use temperature-index. More accurate would be to say these models use air temperature combined with an empirically derived melt factor, and require tuning to some observational target result(s).

Lines 62-63: Also require empirically determined melt factors and/or radiation factors that are modified by radiation inputs. This could be cut, but if kept more correct description of model formulations required.

Line 64: Please clarify what constitutes complexity, or reorder until after energy balance model is presented.

Line 67: 'is therefore questionable' is unclear, please state exactly what the issues of spatial and temporal transferability are.

Line 79: AWS is automated weather station, not 'automated meteorological station'

Lines 79-95: consider condensing information to specific reanalysis product and methods relevant to this study

Lines 104-105 and 107: redundant

Lines 97-117: Consider reordering the information about Peyto Glacier to the end of paragraph to present information from the regional scale first, then the local scale. Or redistribute some regional info to study area section.

Lines 119-121: Great and clear

Line 121: Need citations after the 'has been investigated in several previous studies' statement

Lines 122-14: Is this truly unexplored? Citations needed to understand where exactly current studies decomposing energy balance in a climate sensitivity context lie. I recommend:

Rupper, Summer, and Gerard Roe. "Glacier changes and regional climate: A mass and energy balance approach." *Journal of Climate* 21.20 (2008): 5384-5401.

Anderson, Brian, et al. "Climate sensitivity of a high-precipitation glacier in New Zealand." *Journal of Glaciology* 56.195 (2010): 114-128. (already in citation list)

Figure 1: Shaded NARR grid is misleading given that the work uses a single node, and at a glance a reader might assume the shading delineates panel C. Perhaps shade the location of panel C and show locations of adjacent NARR nodes. Additionally, satellite imagery in panel C would help in assessing debris cover, ELA, for example. Could preserve the contours or provide hillshade off-ice to preserve information on shading.

Line 174: A division or set of sub-sections here between DEM sources/manipulation, and mass balance model simulations relative to different DEMs would help clarity of the approach.

Line 186: elevation not altitude

Line 188: GSC acronym is not previously defined, this will be confusing to international readers

Line 192: use 'see section 3.2.2', not 'c.f.' as no comparison is invited, rather referring to a subsequent step of the approach.

Line: 198: How were these anomalies introduced into the monthly record? More information on method used is needed to understand and allow reproducibility.

Line 203: How were the precipitation records combined? Here I am referring to the choices made (i.e. were the winter gaps in CI AWS filled with PR AWS, or how were overlaps handled). More information on method used is needed to understand and allow reproducibility. Also, note how much gap filling with NARR record was needed overall.

Line 203: Figure two shows precipitation is not recorded between 2005-2016 at either PR or CI AWS locations. The text does not indicate a lack of coverage during the study period. If no AWS precipitation is available between 2005-2016, is the NARR precipitation bias corrected using the pre-2005 historical data

multiplicative deltas? This information is crucial and needs to be reported, even if it is just the same factors as those derived during the historical time period, given that this means mass balance computed between 2005-2016 is exclusively derived from bias corrected NARR.

Line 205: use 'see section 3.2.3'

Lines 205-206 Need to specify the strategy used for summer months.

Lines 205-207: Need to show how the constant lapse rate elevation structure differs from the actual (not mean) lapse rate elevation structure. At least state their differences.

Line 210: its should be their for data, or its for timeseries or record of said data

Line 217: Bukovsky et al. explicitly state that NARR precipitation outperforms other reanalysis products for the continental US and warn against extending this conclusion to the rest of North America. They do suggest that improvements are possible with better precipitation coverage, therefore this statement and use of the citation requires rewording and/or an additional more recent citation to be accurate.

Lines 226-257: need subsections here when methods are introduced to improve clarity. Also need divisions by variable given methods used are different in each case starting line 229.

Line 238: Variable Ta has already been assigned to AWS air temperature (line 187). Need additional subscripts to differentiate air temperatures from different sources/stages of processing. This needs to be corrected across the manuscript, and other variables as well.

Lines 238-239: The data are downscaled to the point location of the AWS, by applying a bias correction. This precedes the downscaling of NARR node information to the finer glacier grid on which the energy balance model is run. These are not necessarily the same thing, although I believe translation of variables (without a reduction in scale) can qualify as downscaling. I recommend offering more information on the Matlab algorithm used to improve clarity.

This is an old, but helpful, reference to clarify terminology:

Hewitson, Bruce C., and Robert George Crane. "Climate downscaling: techniques and application." *Climate Research* 7.2 (1996): 85-95.

Line 257: information regarding downscaling of variables to the model grid should be presented here, before the mass balance model that uses it as forcing is presented.

Lines 261-262/Equation 1: I highly recommend using up to date terms for mass balance terms from the glossary of mass balance:

Cogley, J. Graham, et al. "Glossary of glacier mass balance and related terms." *IHP-VII technical documents in hydrology* 86 (2011).

Line 261: elevation not altitude

Line 261-262: extrapolation of precipitation to the model grid is in fact downscaling (from coarse NARR resolution to finer mass balance model grid). This needs to be explained before the mass balance model is presented. Additionally, the procedure for air temperature needs to be explained explicitly as well.

Lines 264-269: This is the method for obtaining accumulation from precipitation. This is a separate topic and should be in a separate subsection.

Lines 270-271: State grid resolution, present downscaled variable names (with different subscript).

Line 273: Melt is later reported in cm w.e. not mm w.e.

Line 278: Citation needed for statement that justifies ignoring Qg component of energy balance.

Line 283: downscaled to the AWS location (not at). G is downscaled to the AWS location, then extrapolated to model grid also seems like a downscaling step (from coarser to finer resolution).

Lines 294-295: Perhaps I have misunderstood, but does the assumption of spatially constant clear-sky ratio mean that the entire glacier responds to cloud cover at the AWS. If so, a citation or justification on the scales of cloud-cover change would be appropriate and required for reproducibility.

Line 298: It would be appropriate to report the maximum/mean lengths of time that the 'constant' clear-sky ratio is applied to cells that are illuminated after the AWS becomes shaded. If the AWS location is shaded and clear-sky ratio is inferred from that location for several hours at non-shaded locations, any systematic daily patterns in cloud cover could be missed (i.e. common summer afternoon clearing that is missed because all cells are assumed cloudy after the AWS is shaded). If the duration is small it should not matter, but if more than several hours I recommend a citation justifying this approach.

Line 324: describe why these values are 'optimum'

Line 326: LWin subscript

*Line 326: Explicitly describe what assumptions are used for the ice surface temperature (i.e. fixed to 0 when Ta > 0)

Line 331: Ta is multiply defined for AWS and NARR derived air temperatures, and here is neither. Downscaling of Ta to the model grid has not been presented yet, that would be a good place to define a new Ta variable used as input following processing.

Line 336: Downscaling of the WS variable to the model grid is absent, unless the bias corrected AWS location WS is used across the entire glacier. This needs to be explicitly detailed, and justified with literature supporting the use of local, point-scale WS data to drive a distributed energy balance model.

Line 338: WS variable in text, WSz in equation

Line 351: 'similar ice facies morphology between Peyto and Saskatchewan glacier' citation needed.

Line 355: I recommend adding WS/direction to analysis to analysis given the poor performance compared to AWS due to local wind patterns, and the important of sensible heat flux to the overall energy balance.

Line 368: Oerlemans 2001 citation unneeded

371-372 'different greenhouse gases emission scenarios', could just be previously defined RCPs, assuming these are where the T and P values are derived from? Strictly speaking the range is similar but does not encompass (0-8 vs 2.6-8.5)

Line 370: encompass not encompasse, would be more accurate to say approximate given previous comment

Line 372: this IPCC citation is to an annex of AR5. If this is where the values listed are obtained please disregard. Otherwise, I would recommend a reference for the primary AR5 report where RCPs are defined.

Figure two: Adding a shading to the data gaps would help visualize them.

Figure two: See comment for line 203 on precipitation record length. What is the origin of the precipitation data gap between 2005-2016, and what data are being used bias correct the NARR precipitation? It would be helpful to indicate, even if they are simply the values from the historical time period. It would be extremely helpful to color code the precipitation curve to show what data are from each of the PR and CI AWS stations, and what data is gap-filled from the NARR product.

Line 408: the bias-corrected NARR variables, before being downscaled to the mass balance model grid

Line 415: NARR Ta needs to be its own variable

Line 420: I see only a very modest improvement to RMSE and MAE and no difference in R in table 1 for WS (RMSE/MAE/R 2.21/1.72/0.37 vs 2.30/1.85/0.37) and precipitation (RMSE/MAE/R 4.96/2.82/0.30 vs 5.22/3.10/0.30), making the sentence 'Validation statistics were significantly improved using the scaling technique for precipitation and WS' seemingly incorrect.

Lines 419-425: having a call to an equation earlier in text that demonstrates what exactly the scaling method is would be helpful here.

Line 433: Here the data is stated to be the 'bias-corrected NARR', inconsistent with previous statements of downscaled.

Line 454: This seems plausible, but additional analysis could demonstrate it more clearly. I recommend demonstrating the difference in lapse rates between clear and bad weather days as a supporting argument to the statement.

Figure 4: This is a very interesting and helpful figure! I think it would be important to clearly identify the data derived from the regional AWS network (lapse rates and R) and from the Saskatchewan Glacier AWS (WS and WD) to avoid confusion. Perhaps could use different line widths and explicitly state this in caption.

Lines 480-481: I would encourage the use of subscripts, and appropriate terms from the mass balance glossary (see citation at line 261 comment) for all mass balance terms.

Lines 482-485: As previously mentioned (line 205), I recommend showing the difference between mean precipitation lapse rate from AWS network (used to downscale precipitation onto the mass balance model grid) and the elevation dependent lapse rate as a figure. Additionally, I would recommend showing temporal variability over the years with local mass balance validation measurements.

Line 485: explicitly define what is considered 'good model performance', or remind the linear distribution of model vs measured b values if that is the case.

Line 494: m^-1 missing in units

Line 495: use exact language when comparing study balance ratios to those from previous work: the balance ratio is triple (3x) that computed for the region by Rea 2009. This seems anomalous and should be explained further.

Line 510: List the results in text that constitute 'good general agreement'.

Lines 505-516: This section needs to include the geodetic estimate results to have a number to compare the model balance to.

Line 518: Unclear if Ta and P are AWS or NARR variables because of same variable term

Lines 520-522: The difference between mean and elevation dependent precipitation lapse rates needs to be addressed here.

Figure 6: An excellent and helpful figure. Consider renaming the Y labels to cumulative MB. Additionally, choice of colors in legend does not correspond to colors in plot, likely because of overlain curves with transparency. This makes it hard to decipher Z0snow and ice albedo. Change to better color scheme.

Line 539: Sensible heat flux is now defined as QS, it was initially defined as Qh at line 277. These terms absolutely must be consistent throughout to avoid confusion.

8Line 547: Is LW* net longwave? The text refers to it as radiative cooling, which I assume means outgoing longwave exceeds incoming longwave?

Lines 576-578: This should be just melt, not ice melt if I understand correctly that we are above the snowline and therefore no ice surface is exposed in the upper accumulation regions (and all values are in cm w.e.)?

Figure 9: Scale bar needs rescaling on panel A

Line 601: Conversaly - > Conversely

Line 606: were overlapped -> were overlain

Line 607: 'given by the latest projections from climate models' redundant

Line 610: m w.e. -> cm w.e.

Line 613: Static and dynamic mass balance values should be clearly reported and defined with separate subscripts at some point in the manuscript

Figure 10: Previously stated Temperature range was 0-8 C in methods, now 0-7 C. This does not technically encompass RCPs of 2.6-8.5C

Line 645: typo

Line 631-646: A table would be extremely helpful to navigating these results

Line 658-659: reword attractive to a word that is specific to the gain (i.e. efficient, affordable, etc)

Line 660: this has been stated and cited already

Line 662: simple is misleading considering the breadth of statistical downscaling methods. I recommend specifying the exact methods referred to here. Additionally, Clarke et al. 2015 used dynamic downscaling of their surface precipitation to drive their mass balance model and is improperly cited here.

Line 666: Same as above: statistical downscaling can include non-linear regression with multiple predictors, as well as the use of scaling factors, etc, with some methods not having calibration steps. Specify exact methods.

Line 668: replace 'reasonably good' with exact numbers

Lines 674-679: With respect to wind speed, the authors convincingly argue that glacier-scale katabatics that are not captured on the coarse NARR grid account for poor performance (line 410-412), and identify the impact of poor wind speed variable performance on temperature lapse rates (section 4.3). However, the wind speed variable performance is only further discussed as suffering from katabatics (lines 674-679), and the impact of this poor performance on turbulent heat fluxes computed in the energy balance model seem absent. Furthermore the results demonstrate that the sensible heat flux plays an important role in the overall energy budget of the glacier surface. I recommend further comments on the possible consequences of poor wind speed data forcing the model, and would suggest adding this variable to the earlier forcing variable sensitivity tests. Additionally, the impact of using point scale WS data to drive an entire glacier-scale energy balance model must be explicitly justified.

Line 694: biased -> raw

Line 697: list all assumptions explicitly for clarity

Line 700-705: present errors as fraction of modelled results to give context. Discussing an error of 78 cm w.e. a-1 is different from an error representing +/- 108% (77.6/-0.72 m w.e.).

Lines 704-705: Significant figures for this result are different from previous ones in the results section (different units too), make consistent.

Line 707: list numbers corresponding to these results for clarity and context

Lines 711-713: Again, I highly recommend assessing the sensitivity of the model to Wind Speed given the poor performance of this variable and the importance of fluxes using Wind Speed to the interpretation of the results.

Line 717: glacier models -> glacier mass balance models or energy balance models or glacier surface process models given that 'glacier models' includes dynamic models from the bed to the surface.

Lines 748-757: a table would help navigating this discussion point

Line 782: m w.e -> cm w.e.

Line 782-783: 'A value of 1 would occur if all precipitation were snowfall and there were no albedo feedback.' -> specify that these values are ratios and unitless (the previous sentence does so partially). The Oerlemans 2001 reference here seems generic and unneeded.

Line 786: units are now in mm

Line 787: and now in m w.e.

Line 798: MacDougall and Flowers warn that transferring model parameters between sites in a specific setting will result in increasing error depending on the number of parameters and inputs carried over. This statement inexact resulting in an improper citation.

Line 800: I believe this has been done before? Please include citations to identify where this work improves the existing work

Line 804: please include geodetic estimate here for clarity

---

## Author Comment (AC2)

Dear RC2,

Thank you for taking time to thoroughly review our manuscript and for your constructive comments. Here is a preliminary reply to your major concerns. A more detailed version will follow with the revised manuscript.

Major Concerns

1. *I don't think it is unreasonable to use the single NARR grid cell that coincides with this glacier, given the 32-km scale of NARR grid cells and the good correlation with available ground T, RH, and shortwave radiation data. I do wonder why NARR incoming longwave radiation data was not used in this study though, for consistency. Was this considered, or compared with the longwave radiation parameterization that is used? The parameterization requires cloud cover fraction that is taken from NARR, so why not just use incoming LW from NARR, which appropriately considers the 3-hourly vertical temperature, humidity, and cloud structure, vs. a parameterization that only uses near-surface values?*

We also think this choosing the closest NARR gridpoint would be the most obvious choice, but Rev1 had some reserves and made useful suggestions, which will be included in the revised MS. Regarding the incoming longwave: thank you for spotting this…The Konzelmann parameterization described in section 3.3.3 of the methods is not what was used. This was an earlier application of the model, and as you correctly spotted this is a crucial aspect that we in fact had carefully addressed! Here is what was done:

We did not use the raw NARR incoming longwave (LWin) because these would carry an elevation bias, since the NARR gridcell is 237 m higher than the AWS. NARR LWin will therefore be too low. We did not have observations of LWin at the AWS to correct this bias directly. Instead, LWin was calculated from the bias-corrected NARR air temperature (Ta) at the AWS location and the atmospheric emissivity (E) from NARR: LWin = theta*E*Ta^4, where theta is the Stefan Boltzmann constant. The result is a LWin that is almost equal to the raw NARR Lwin, but with a slight correction for the difference in air temperature between the NARR gridcell and the AWS. The key point is to use the NARR emissivity, which reflects the vertical humidity and cloud structure. I am glad you spotted this, we will correct that section in the revised MS. Accordingly, panel f of figure 3 showing the bias-corrected NARR forcing will be changed to replace the cloud cover (CC) with the atmospheric emissivity

2. *The authors note that ERA5 was not available at the time of the study, but it has been out for more than one year now - about 1.5 years I believe, and offers 1/4 degree resolution with hourly data, which can avoid some of the complexities and assumptions in mapping the 3-hour NARR data onto hourly estimates. ERA-land is even higher resolution. I would not insist on this, as it is a lot of additional work, but I think it would be valuable and would strengthen the manuscript a lot to explore model results with ERA5 and/or ERA-land as well - this could very much help to test the robustness of the results and conclusions, and lower the reliance on NARR's relatively unproven veracity in the complex mountain terrain. This could be follow-up work, but I also have the nagging sense that this study risks being already out of date. The argument that ERA5*

*precipitation is questionable is not really valid, as all of the reanalysis output is bias-adjusted and NARR is weak in this respect.*

As you say, comparing with ERA-5 would be interesting but would entail too much work at this point, so it will have to be left for follow-up work. However, it is worth mentioning that ERA5 has a 0.25 degrees resolution (~ 28 km) vs. 0.30 degrees for NARR (~32 km). Most importantly, all atmospheric forcings for the ERA5-**land** product (0.1 degree) are interpolated from ERA5, so there is no real gain in spatial resolution for atmospheric variables, but only so for land surface variables. We think that the methodological framework used in this study could very well be re-applied with ERA5 in follow-up work, and this will be emphasized.

3. *I agree with R1 on the confusion regarding terminology in the methods. As I read it, the NARR output is not downscaled to the AWS; this study uses bias-adjusted rather than downscaled NARR output, as I understand it. Bias-adjusted NARR fields are then distributed over the glacier from a reference site (the AWS) using locally-relevant lapse rates and sophisticated methods for the incoming SW radiation. As I understand it, the reference site for the precipitation differs (e.g., Parker Ridge), but it is a similar approach. I am happy to leave it to the authors to decide how they would like to refer to this process (extrapolated, downscaled, lapsed, or distributed over the glacier), as long as it is clearly defined and consistent in the manuscript. It is not what I think of or what is commonly referred to as downscaling though.*

Yes, we agree we need to clarify the terminology. We think that bias-correction should be used to described the correction of NARR forcings, and "distribution to the glacier grid" for the subsequent step.

4. *I admire the use of the regional network of permanent weather stations to develop temperature and precipitation lapse rates, but I worry about the relevance of these values to the glacier itself. These are all off-glacier sites with a maximum elevation of 2025 m, while Saskatchewan Glacier extends from about 1800 to 3300 m. Glacier near-surface temperatures (and the surface energy balance that influences these) are very specific, as is the snow accumulation regime on glaciers and in the unsampled elevation band from 2025-3300 m. I don't have great confidence in the applicability of the lapse rates as determined by the off-glacier climate station network. For temperature, why not use the average daily or monthly lapse rates as determined by the HOBO temperature transect? I realize that these are summer-only, and the data are limited, but this is what is used for the diurnal lapse rates so this would seem relevant and consistent. Winter temperature lapse rates are not important to the glacier melt, so could be assigned an average or May value. For the precipitation lapse rates over the glacier, is there a way to use available winter mass balance data (in situ and/or LIDAR-inferred) to look at this? The current precipitation lapse rate may be appropriate, but it would be helpful to constrain and evaluate this, as well as the assumption of a sustained (and strong) linear increase in precipitation across the icefield plateau from 2800 to 3300 m.*

This is a common issue, to rely on lapse rates calibrated with lower-elevation stations. Your preoccupation is warranted, though. We thus checked the mean on-glacier lapse rates derived from the Hobos on the ablation stakes for their available period from May to August 2015, which yields a value of -0.46°C/100 m, versus the average from the permanent network for the

same months: -0.49 °C/100. This rather similar value gives us confidence that the lapse rates extrapolate well to the AWS site. When considering the mean lapse rate from the permanent network for May to October, i.e. the months during which the toe of the glacier (1784 m) is above the zero-degree isotherm (Supplementary Figure S1a), then the mean lapse rate is -0.47 °C/100. Hence replacing the mean monthly lapse rates by the mean lapse rates from the Hobos would not significantly change the results. These considerations will be included in the revised MS. We also want to recall that diurnal anomalies in the temperature lapse rate derived from the Hobos were superimposed on the seasonal lapse rate cycle derived from the permanent network, to more realistically simulate diurnal changes in the lapse rates not captured by the permanent network.

As for the precipitation lapse rate: this is harder to constrain, as in other studies, as there are no high elevation precipitation gauges. The Columbia/Parker Ridge stations (2000 m) are actually pretty high already compared to typical stations in the area. The sensitivity of the modelled mass balance to this lapse rate was tested in Fig. 6c: the sensitivity is expectedly high, which highlighted the need to constrain this value as best as possible from ancillary observations. We found that the lapse rate constrained on the permanent network gave good results. The performance, as you suggest, can be specifically assessed against the end of winter mass balance data (bw), which was done in section 4.4 and figure 5a: the lapse rate used yielded satisfactory bw simulations up to the highest stake/snowpit, which gives confidence on the lapse rate used. We did discuss the limitation of using a time-invariant lapse rate, which gave poorer results in the dry 2016 year (Fig6a), but we will also better discuss the unknown biases for the highest reaches of the glacier not covered by snowpits/stakes. This area only represents 8.8% of the total glacier area.

> 5. The precipitation lapse rate that is used is based on the reference climate station data from November to March (l.206). This does not coincide with the accumulation season on the glacier, which is more like September to May.  Is this same precipitation lapse rate used for April to October, and is there objective support for that? This needs to be discussed and addressed, perhaps with an examination of the primary data or perhaps by bringing in the winter mass balance data from the glacier, if there is some from the 2014-2016 study. November to March is relevant for the lower-elevation snow season, but not that of the glacier, where autumn and spring often bring a lot of snow.

Please refer to section 7 and figure S1 that discusses this extensively based on observations. The current model code only allowed using a single value for the precipitation lapse rate, which is a limitation. We sought the most representative winter value. The precipitation gradient is best defined (strongest relationship between precipitation and elevation) during November-March which is also the period when most snow accumulation occurs, i.e. when the whole glacier is above the zero-degree isotherm (Fig S1). The lapse rate value for this period (mean = 15.6 % per 100 m) yielded satisfactory simulations of the end of winter mass balance (Figure 5 a), but we do show that there is a high sensitivity of the simulated mass balance to the uncertainty in this lapse rates (± 4%/100 m: see figure 6c and discussion). In fact, this point was a strong one made in the paper, that it is crucial to constrain this lapse rate to obtain good mass-balance simulations, but we realize that we did not express it clearly enough, so that we will emphasize this better in the revised MS. I also agree that snow accumulation occurs from September to May, for the higher parts of the glacier, and that the lapse rate starts decreasing in these shoulder seasons (April-May and September-Oct.: Fig. S1). Yet the bulk of snow accumulation is

rather from Nov. to March and the ± 4% sensitivity used in Figure 6 captures the uncertainty of adding Sept-Oct and May in the mean value). Most importantly, the value used yielded good end of winter mass balance, which is dominated by accumulation.

6. *Wind speed results on ll.393-395. These are extremely high average wind speeds, an annual average of 16 m/s and up to 23 m/s in February. I appreciate this is likely a windy site, and there are katabatic winds here, but those are typically stronger in summer. Are the authors confident that these units are correct - is this perhaps km/hr, or are these maximum (vs. mean) wind speeds that are reported here and plotted in Figure 2? An average monthly wind speed of 23 m/s equates to 83 km/hr, which is not plausible. Values reported and used later in the manuscript (e.g., from NARR, means closer to 5 m/s) are more reasonable. I would also add that I have spent some time on this glacier, and there is a steady and reliable down-glacier wind, but not of the knock-you-over variety.*

Thanks for spotting this. Indeed, quite high… I will say I was nervous for a moment… The wind speeds on Figure 2d are in km/h, not m/s as labelled, but they were correctly converted to m/s before bias-correcting the NARR wind speeds. This is why the resulting NARR forcings (Fig.3c) are within expected values, as you spotted correctly. We will modify figure 2d to plot wind speeds in m/s rather than km/h.

7. *I am not sure what 'homogenized' means here in the context of the observational precipitation records that are spliced. Homogenized has a very specific meaning for meteorological data sets, involving corrections for discontinuities associated with station moves or changing conditions/instruments/methods at an observation site. The precipitation data also seem to have a lot of gaps, which makes me worry about the time series of mean annual values. It seems best from about 1972 to 1994, not for the full period plotted in Figure 2. What methods were used to gap-fill this data for missing months? Apologies if I missed this. My sense is that it would be best to use these data for long-term mean monthly values from 1979-1994, using all available monthly data over this period. This can then inform a bias-adjustment of NARR mean monthly values for the same period, 1979-1994. Then go with bias-adjusted monthly NARR (or ERA5) precipitation for the study. Just my surficial thoughts on inspection of the observational data in Figure 2.*

We will rephrase to 'merged' instead of 'homogenized'. Yes, there are gaps, as discussed in the MS. Recall that we used the daily observations from the 'merged' Parker Ridge/Columbia station to bias-correct the daily NARR precipitation, using all common days with available data. So we did not gap-filled the observations, we used whatever was available to bias-correct the NARR data on a daily time scale. The numerous gaps were the prime reason to use the (more uncertain) NARR precipitation as forcings. I hope this is clear, we will revise the text to make this clearer to the reader.

8. *Perhaps my most significant concern: the sensible heat fluxes seem far too high for a mid-latitude continental glacier, and compared with other data from the region (Peyto, Haig Glaciers). Also, it is surprising and unusual that latent heat fluxes are positive. I don't trust either of these results. Are the erroneously high winds speeds (point 6) the reason for this? This could explain the high values of sensible heat flux, though it is still*

*odd that latent heat flux is positive. What is the basis for determining the snow/ice surface temperature in these calculations? This is critical to the turbulent heat flux calculations, and I did not see a discussion of this in the paper - apologies if I missed it. Is a melting glacier surface assumed in the summer? What is assumed through the rest of the year? I wonder too if the snow roughness value is appropriate for winter conditions - 6 mm is high, perhaps more reflective of sun cups than the smooth winter and spring snow surface. Snow roughness values closer to 1 mm are commonly adopted in glacier modelling. The sensitivity to this variable could be more thoroughly explored, perhaps considering order of magnitude rather than \pm 1 mm variations.*

First, the wind speeds are correct, as mentioned in point 7. We were also a little surprised at first by the large sensible heat fluxes and small but positive latent heat fluxes. We will double check again that everything is correct, but we think they are correct and that the climate of the Columbia icefield (as you may have experienced) is actually quite different than Peyto and the other glaciers you refer to, which have more continental climates. The icefield is higher and has its owns weather, more humid than Peyto. This is why the Columbia icefield is often wrapped in clouds while the valley is clear (and explains why one can get stuck on the icefield waiting for a helicopter while it is sunny below the icefield…). But you are correct that this needs to be better discussed, and thank you for suggesting references. Some are already quoted, but we will discuss this point in more details in the revised MS.

You are right, we did not mention how the surface temperature (Ts) is calculated. Ts calculation follows Hock and Holmgren (2005). Ts is assumed to be zero if the computed energy balance is positive. If the energy balance is negative, Ts is lowered iteratively by steps of 0.25 K until the energy balance for the time-step and the gridcell considered becomes zero.

Regarding the roughness values we used the closest analogs on Peyto (Munto, 1989) to guide theses parameters. The snow value is indeed on the high side but still closer to 'glacier snow' during the snowmelt period. Since we used a time-invariant value a value close to metamorphized snow is better than mid winter when ablation is limited. We could expand the range for sensitivity analyses (Fig 6) to enclose a broader range of published values.

9. *AWS snow accumulation is reported on l.485. How is this recorded? Isn't this just a tipping bucket rain gauge at this site? Or is this based on measurements from site visits? It would be helpful, per the comment above, to report winter snow accumulations on the glacier and use this data to help evaluate the precipitation modelling.*

Using an ultrasonic gauge on the AWS. I see it was not mentioned, we will include this when describing the station. End of winter mass-balance (bw) at stakes and snowpits/soundings between stakes are presented in Figure 5a, we will try to better emphasize this.

10. *Agree with R1 that it will be really helpful to use mm or m w.e. throughout for mass balance, rather than a mix of mm, cm and m.*

I agree, we will homogenize the units.

11. *l.494, the balance gradients. Please give in units of m or mm w.e. per m. This is an interesting result, though I worry that the unusually high value (steep gradient) on the upper glacier is in part due to the unconstrained precipitation/accumulation gradient on the glacier. Looking at the available data in Figure 5 from 2015 and 2016 (2014 data are not sufficient), it would be hard to justify a bi-linear vs. linear relationship for ba. This is purely a model result then, as I understand it - can it be explained via mass balance processes here, since the authors describe the balance ratio as an unusual result?  Is it reflected in the geodetic mass balance profiles?  This is a significant point, as the balance gradients (values, linear vs. bi-linear) are potentially significant for regional-scale mass balance modelling - I can imagine other authors using the values that are reported in this study.  Perhaps it is early to think about this too much, as the results may change upon revisiting of the wind speeds, modelled surface temperatures, and modelled turbulent fluxes.*

It is true that the modelled gradient is not constrained by point mass balance beyond 2900 m. Geodetic gradients include ice flux, which would complicate deriving mass balance gradient from them, no? The area of the glacier above 2900 m represents only 8.8 % of the total area, so extrapolation errors in this unsampled area would have a small impact on glacier-wide balances. Yet I agree that we should perhaps not use the modelled gradient, at least not beyond the last validation point at 2900 m. Perhaps calculating the gradient from the ELA up to 2900 m would be acceptable? This slightly changes the gradient, from 0.31 cm w.e./m (whole accumulation zone) to 0.32 for the period of observation, and from 0.29 to 0.31 for the long-term (1979-2016) period. The revised balance ratio would be 3.10 in place of 3.34, still high compared eastern Rockies 'drier' glaciers, and closer to 'wetter' West coast glaciers. This point, along with the larger simulated sensible heat flux and positive latent heat flux, suggest that the weather of the Columbia icefield is indeed distinct from the more continental types glaciers of the Rockies. We see no reasons at this stage to question the validity of the model results, but we will certainly carefully review our analyses as per your comments before re-asserting this conclusion in the revised MS.

---

## Author Response (AR1)

**Response to reviewers**

**Reviewer 1**

**General comment:**

The paper presents the use of reanalysis data to drive a distributed, physically-based modelling endeavor of the Saskatchewan Glacier. The authors cleverly combine a very limited on-site AWS record with longer term regional AWS records, and NARR product for multiple variables to model the long term (1979-2016, and 21st century climate perturbations) mass balance of the glacier. The analysis includes geodetic estimates of glacier change and point-scale estimates of mass balance, that are used to further constrain the model and investigate its performance. These results are used to assess the glacier's 21st century climate sensitivity across projected temperature and precipitation changes from the IPCC RCPs.

I found it challenging to follow the organization of the scientific information and I believe the manuscript should be significantly reworked. The text contains grammar and spelling mistakes that need to be addressed. The authors should clarify their use of terminology (i.e. 'altitude' vs 'elevation' are used interchangeably), homogenize variable names throughout the text (i.e. Ta for all air temperatures derived from NARR, from the regional AWS network, and the downscaled model input), and use consistent units (i.e. melt reported in mm w.e., cm w.e., and m w.e. at different stages). The information pertaining to the methods is disorganized causing difficulty in deciphering the approach (with omissions of details about downscaling key variables such as temperature and wind speed to the distributed glacier grid), and at some points systematic misuse of terminology ('downscaling' vs 'bias correction' vs 'extrapolation to model grid') obfuscates the underlying novelty of the approach to designing a spatially-distributed application of a physically-based melt model in a setting with limited input data.

Additionally, the use of reanalysis data from a single NARR grid node as model forcing is not adequately justified. The authors convincingly demonstrate that these data perform well in comparison to observational records of climatological variables. However, without additional information on the performance of either nearby nodes or an aggregate of these, it is impossible to determine whether this convincing performance is due to a modelling workflow which leverages a reanalysis dataset that is transferrable to other glaciers (especially those that are large enough to intersect multiple grid cells, or small enough and less fortunate in their geography to be equidistant from many grid nodes), or simply due to the coincident position of Saskatchewan Glacier relative to the nodes of the reanalysis product used. Some form of evaluation pertaining to the regional potential of the NARR product for direct incorporation into this melt-model workflow would help to better highlight this paper's relevance to cryospheric research.

For further context, I would highlight the interesting use of the melt modelling results to assess the sensitivity of Saskatchewan Glacier, specifically in extracting meaningful process-based explanations from the glacier's energy balance. These are obviously locally significant, and perhaps regionally so, but it is difficult to address the true scale of relevance of these results without significant improvements to the clarity and organization of writing, and rigor of the analysis. These improvements, combined with a more focused introduction presenting exactly the purpose of the study would greatly help in highlighting the potential use of the method for data-poor glacier settings. Additionally, improvements

to clarity and organization would ensure the approach is reproducible, and as such, transferable to others.

We thank you for taking time to thoroughly revise our work. We have made several significant changes to the manuscript in response to your major comments. New analyses were carried out, and the introduction, results and discussion sections were largely reworked to take these new results into account and to respond to the comments from both reviewers.

As the major comments outlined here were also addressed in more specific form below, please see our detailed responses to them in the next section.

**Specific comments:**

Section 1:

The introduction contains very detailed information on several topics related to the analysis that distract from a well-formulated motivation for this work. I would recommend using more direct wording to improve the readability of the text (i.e. 'the amount of precipitation remains unchanged' vs 'precipitation remains unchanged'; line 34, 'the so-called 'temperature index' or degree days' models vs 'temperature index models'; line 59). I would also recommend reducing the specificity of information on topics both tangentially and directly related to the paper topic.

Well noted. The introduction was largely edited to remove superfluous information while emphasizing elements directly pertinent to the work, enriching it with new references.

With respect to tangential information: the details about GIC mass loss (lines 45-50) seem incongruous with the paper topic and the opening summary of glacier-freshwater relationships, details about the Canadian Rockies (lines 97-117) could be shortened by moving some information to the study area section.

Agreed. The whole segment about GIC and glaciers contribution to sea level was removed as it is not pertinent. The review of glacier area loss trends in the Rockies was shortened and moved to study area.

With respect to direct information: detailed information on the temperature index to energy balance spectrum of melt models could be significantly condensed (lines 59-77), information on reanalysis products could be limited to NARR (lines 83-90), and information on their downscaling could be limited to the methods relevant to the study approach (lines 90-95).

Information about model types was condensed. However, we feel that mentioning the use of other reanalyses for glacier model forcings, and giving a short overview of downscaling strategies is important ta place our study in this context. We have tried to make the text more straightforward in this sense

The stated objectives highlight the 'comparatively unexplored' topics of ablation vs precipitation phase feedbacks to temperature sensitivity (lines 119-124) but offer no literature to situate this statement or the study. Please clarify what exactly is novel about this study's results and situate it in existing literature to really highlight the novelty of the study. The text at the very end of the paper (line 810-811) could be tied back to a statement here as it is more specific.

We have reworked that section, first by reworking the previous section on glacier climate sensitivity, highlighting those past studies have not unraveled the respective contribution of atmospheric

warming, surface feedbacks and precipitation phase feedbacks on the mass balance sensitivity to warming. The objectives were then more clearly stated, i.e., addressing the questions of (i) constraining a physically-based model in the context of spare observations, and (ii) unraveling the respective contribution of changing energy fluxes vs precipitation phase feedbacks to the temperature sensitivity of mass balance.

Section 3:

The data and methods section requires significant redesign. Section 3.2.3 contains most information pertaining to reanalysis data, and all pre- and post- processing associated with bias correction and downscaling across 5 climate variables with no internal divisions. Additionally, information on downscaling across the glacier model grid in a distributed manner is contained within the mass balance model section, which makes it hard to find and follow the logic of the approach. I would recommend a clear division of data (sources, types, any pre-processing strategies) and methods used in manipulating these data to create model inputs (downscaling, bias correction), followed by information specific to the mass balance model once all input variables (to the distributed model) have been presented. I like the authors strategy of presenting each element of the energy balance, but this could be shortened significantly if information relative to model input pre-processing were already made clear beforehand.

We reworked that section based on your comments and suggestion. The new section '3.2.4 Downscaling NARR to the glacier model grid' now describe in more details how Raw NARR grids are transferred to the glacier model grid. All sections from the former model description pertaining to the extrapolation of meteorological variables to the grid have been moved there, except for the details of the solar radiation routine which is tied by the model equation. A short statement was still included to describe how radiation is spatialized.

Additionally, the exact workflow used is very difficult to decipher. The authors first obtain NARR data from a node closest to the study area, then they bias correct these long-term timeseries of climatological variables (1979-2016) using a two year on-ice AWS record (excepting precipitation), then downscale the bias corrected NARR data to the model grid using lapse rates derived from regional AWS records. In the text, the terms bias correction and downscaling are used synonymously for work performed at the on-ice AWS location (lines 238-239), and the precipitation variable downscaling step is described as 'extrapolating to the model grid' in the mass balance model section (lines 262-263). See my technical comments for citations that may help in differentiating these terms. It is necessary to address the confusion in usage of terms such as 'downscaling', 'extrapolation', and 'bias correction', and distinguish between the steps for the transition of NARR climatological variables from the grid cell time series, to the AWS location timeseries, to the distributed model grid timeseries ultimately used to force the mass balance model for each variable, in order for this workflow to be reproducible.

'Downscaling' had been used as a general term for any transfer function that translates a coarse climate field (reanalyse, RCM, GCM, etc.) to a local meteorological record (here an AWS). Bias-correction is one, and the simplest downscaling strategy (Teutschbein, C., & Seibert, J., 2012). However, we agree that the workflow may not have been clearly outlined, so we reorganized and re-segmented the methodological workflow as per your suggestions.

To clarify, the downscaling term implies two steps that are now clearly outlined in the new section 3.2.4 'Downscaling NARR to weather stations': **(**1) interpolation of the NARR gridded data to the reference weather stations; (2) bias correction of the interpolated NARR data.

A new section 3.2.5 'Extrapolation of NARR data to the glacier DEM' now describes strategies and processing steps involved in distributing the downscaled NARR to the glacier DEM. All lapse rates calculations are now included in this section.

Section 4:

The use of a single NARR grid node to drive the entire modelling study must be evaluated further. The authors demonstrate that temperature, solar radiation, and relative humidity perform well compared to a 2 year on-ice AWS record. My concerns and related recommendations here are two-fold:

(1)  There needs to be evidence that demonstrates how the use of this grid node is justified compared to other regional grid nodes (for example, at better situated elevations), and/or a regional aggregate of NARR data, and/or using bilinearly interpolated NARR data at the AWS location (given that the nodes define a continuous field across the region of interest). Some recommendations include: commenting on differences between the coarse NARR topography (derived from surface geopotential variable in NARR for example) to credit/discredit the use of information at the AWS location, a sensitivity test of the model using a different grid node(s) and/or a regional aggregate of nearby grid nodes as forcing, and/or using AWS location bilinearly interpolated values to assess the geographic sensitivity of point-scale data as forcing (and assessing the relative cost/benefit of introducing interpolation to this approach).

(2)  As it stands the novelty of this method seems limited by the possibility that it is only applicable to glaciers with nearby reanalysis grid nodes. Given that this study investigates a single glacier in the Canadian Rockies, it would greatly improve the significance of the work to demonstrate that this method can reliably inform physically-based models on other glaciers with limited AWS forcing data. To do this however, the above (1) needs to be addressed.

We followed your suggestion and compared the nearest neighbour interpolation method (NARR gridcell closest to the AWS as initially done), with NARR time series bilinearly interpolated from the 9 gridpoints closest to the AWS. We now compare the two methods against the AWS measurements (see new Figure 3) and comment on the differences. Further, we also included the NARR interpolation method in a new model sensitivity experiment, comparing the long-term reconstructed mass balance with closest NARR forcing vs. a model run with bilinearly interpolated forcings.

Throughout:

Methods details are missing:

-  Temperature downscaling using lapse rates to the model grid is never explicitly detailed.

It is now clearly stated in new section 3.2.5 (extrapolation of NARR to glacier DEM)

-  The sensitivity of the model to poor wind forcing input is never evaluated (the downscaling of this variable to the model grid is not addressed and I am unsure if the AWS bias corrected NARR values are used to force the entire glacier surface or if these data are further downscaled to be distributed across the finer model grid). If only one wind speed timeseries is used to drive the entire distributed energy balance model, this strategy must absolutely be justified with

supporting literature given a lack of distributed on-ice wind data to evaluate local variability. I do not have sufficient expertise to comment on the limitations of this in detail, but it seems important to demonstrate that this is a valid strategy for forcing wind speed, given how it fails to perform well due to the presence of katabatic and sloping wind signals in the AWS record (table 1), and the fact that WS factors heavily into the discussion of results relating to the energy balance of the glacier. See technical comment for more information.

This point is not explicitly addressed in section 3.2.5. RH and wind speed were assumed spatially invariant, in the absence of local data to constrain spatial variations. This is a common practice in model studies of valley glaciers and now we refer to the literature in support of this. There is evidence from nearby Athabasca Glacier that the wind speed is relatively constant down glacier, this information was added. But we agree that this is a lingering and possible shortcoming of the model, which is now mentioned in the discussion.

As you noted the NARR wind speed had a rather poor correlation with AWS observations. In order to examine further how NARR variables, influence the modelled mass balance, we performed a new sensitivity analysis, described in section 3.5 (Model validation and uncertainty analyses) and displayed in panels b-c of the updated Figure 8. The NARR forcings were replaced one at a time by the AWS observation for the glaciological year 2015 when continuous AWS data were available along with stake data, and the results discussed. It was found that despite the fact that NARR Ta were well correlated with AWS, it had the largest impact on point mass balance validation.

- Smaller details of the workflow are missing (i.e. filling of precipitation record data gaps, see technical details) and are needed to ensure the study results are reproducible and the method transferable to other settings.

See our responses to each technical detail

Presentation of results could be improved:

- Units change frequently throughout; this is distracting and must be made consistent.

We homogenized units, e.g. mass balance is now everywhere in m w.e.

- Please assign a unique term to each variable. For example, temperature that has undergone different steps of processing should have a different subscript, terms presented at one stage of the paper should be consistent throughout (i.e. Qh changes to Qs for sensible heat flux in methods/results), and variables in equations should have the same notation as those in the text that defines them. Whenever possible, variables relating to mass balance should be consistent with the mass balance glossary:

  Cogley, J. Graham, et al. "Glossary of glacier mass balance and related terms." *IHP-VII technical documents in hydrology* 86 (2011).

Symbols were reviewed and changed accordingly. We had indeed followed Cogley for mass balance terminology (b = point wise, B = specific, with subscript a = year, s=summer and w = winter, but omitting the surface 'srfc' subscript for simplicity). The text was reviewed to make sure this nomenclature was used throughout.

We reviewed extensively the symbology and made several changes to improve the clarity.

- Occasionally, statements are inaccurate of the literature cited for support, and the work of others is improperly cited. See technical comments.

See response there

**Technical/detailed comments:**

The abstract would benefit from being more focused on the paper's key results: removing information on the static vs dynamic extent performance (lines 20-21), details surrounding IPCC RCPs (lines 23-24), and details of the melt vs sensitivity (lines 26-27, 29-30), would both shorten the text and emphasize the key findings.

The abstract has been largely rewritten to reflect the changes made to the manuscript

Lines 23-24: 'IPCC representative concentration pathways climate scenarios' is redundant, either climate scenarios or RCPs

done

Lines 44-49: Very dense information on SLR specific to GICs can be condensed to remain specific to small mountain glacier study

This segment was removed altogether

Line 51: Sentence lacks specific information, or citation if the importance of mass balance variable is necessary to argument.

We added a reference to Hock and Huss (2021)

Lines 52-53: 'Only a few' -> list how many. If this information does not exist, a citation pointing what is known would be appropriate. Time-consuming better described as resource intensive.

We added 'i.e. 30 glaciers with uninterrupted records since 1976 (Zemp et al., 2009),..'

We corrected to time-consuming

Lines 59-62: 'So-called 'temperature-index' or 'degree-day' models' is redundant, can just use temperature-index. More accurate would be to say these models use air temperature combined with an empirically derived melt factor, and require tuning to some observational target result(s).

We only used the temperature-index now. This whole section was shortened somewhat. We do close this section by saying that empirical models need calibration and are hence not ideal for projecting mass balance under climate change.

Lines 62-63: Also require empirically determined melt factors and/or radiation factors that are modified by radiation inputs. This could be cut, but if kept more correct description of model formulations required.

See modifications to the section. 'empirical' models include temperature index models and their enhanced versions

Line 64: Please clarify what constitutes complexity, or reorder until after energy balance model is presented.

This segment was modified. The 'complexity' of EB models is explained later, due to the number of inputs needed (forcings and parameters)

Line 67: 'is therefore questionable' is unclear, please state exactly what the issues of spatial and temporal transferability are.

New sentence: 'Still, while these simple empirical models contain few parameters which simplifies their application, they must be calibrated on observations, which makes model extrapolation in a different climate questionable (Carenzo et al., 2009; Gabbi et al., 2014; Hock et al., 2007; Wheler, 2009)'

This is a well known and common issue I would say, that calibrated models are meant to be transferred outside of their calibration conditions (climate and/physiography)… We believe the point is made and the interested reader can continue exploring this issue with the cited references.

Line 79: AWS is automated weather station, not 'automated meteorological station'

Corrected

Lines 79-95: consider condensing information to specific reanalysis product and methods relevant to this study

We tried condensing the text, and removed the references to NCEP and ERA5 reanalyses to focus on NARR

Lines 104-105 and 107: redundant

Former Line 107 was removed

Lines 97-117: Consider reordering the information about Peyto Glacier to the end of paragraph to present information from the regional scale first, then the local scale. Or redistribute some regional info to study area section.

We moved some of the info to study area and removed altogether some superfluous information about area loss in the wider area.

Lines 119-121: Great and clear

thanks

Line 121: Need citations after the 'has been investigated in several previous studies' statement

See modification to that section. We have added some text before the objectives to better explain knowledge gaps in former mass balance climate sensitivity studies, with supporting reference

Lines 122-14: Is this truly unexplored? Citations needed to understand where exactly current studies decomposing energy balance in a climate sensitivity context lie. I recommend:

Rupper, Summer, and Gerard Roe. "." *Journal of Climate* 21.20 (2008): 5384-5401.

Anderson, Brian, et al. "Climate sensitivity of a high-precipitation glacier in New Zealand." *Journal of Glaciology* 56.195 (2010): 114-128. (already in citation list)

Thank you for pointing these out. Anderson was already cited; their study is indeed the closest to providing a clear partitioning of the temperature sensitivity into albedo feedback, rain-snow partition and increasing melting due to atmospheric warming. However, they do not clearly report the respective contribution of these three mechanisms to the mass balance sensitivity. We still refer to them as an example of such effort. The work of Ruper and Roe (2008) tied ELA changes in response to warming to the dominant (melt vs sublimation) mechanism. A good example of unraveling the mass balance sensitivity but not quite the same as we did. We do cite these two studies now in this section as examples of using energy-balance models to unravel the processes driving the climate sensitivity of mass balance and refer to their results in the discussion.

Figure 1: Shaded NARR grid is misleading given that the work uses a single node, and at a glance a reader might assume the shading delineates panel C. Perhaps shade the location of panel C and show locations of adjacent NARR nodes. Additionally, satellite imagery in panel C would help in assessing debris cover, ELA, for example. Could preserve the contours or provide hillshade off-ice to preserve information on shading.

We have modified the map: removed the fill from the different glacier extent, added a Landsat 8 background image (August 23, 2018). We have added the 9 closest NARR gridcells to panel b and added the inset showing panel c.

Line 174: A division or set of sub-sections here between DEM sources/manipulation, and mass balance model simulations relative to different DEMs would help clarity of the approach.

We re-organized the information in this section, introducing two divisions in the text to better separate the information

Line 186: elevation not altitude

corrected

Line 188: GSC acronym is not previously defined, this will be confusing to international readers

corrected

Line 192: use 'see section 3.2.2', not 'c.f.' as no comparison is invited, rather referring to a subsequent step of the approach.

Corrected there and hereafter

Line: 198: How were these anomalies introduced into the monthly record? More information on method used is needed to understand and allow reproducibility.

This is now better explained in new section 3.2.5 'Extrapolation of NARR data to the glacier DEM'

Line 203: How were the precipitation records combined? Here I am referring to the choices made (i.e. were the winter gaps in CI AWS filled with PR AWS, or how were overlaps handled). More information

on method used is needed to understand and allow reproducibility. Also, note how much gap filling with NARR record was needed overall.

The description was improved. We combined two nearby stations (Parker Ridge and Columbia, which have almost the same elevation). We merged by averaging both records, but data gaps remained. So the model is forced with NARR precipitation, after bias-correction against the merged record (using only days with observations)

Line 203: Figure two shows precipitation is not recorded between 2005-2016 at either PR or CI AWS locations. The text does not indicate a lack of coverage during the study period. If no AWS precipitation is available between 2005-2016, is the NARR precipitation bias corrected using the pre-2005 historical data multiplicative deltas? This information is crucial and needs to be reported, even if it is just the same factors as those derived during the historical time period, given that this means mass balance computed between 2005-2016 is exclusively derived from bias corrected NARR.

Indeed, the NARR precipitation record was bias-corrected with the merged PR/CI records using only days with available observations over the period of overlap between NARR and the precipitation record (1980-2008). This is now clearly indicated in section 3.2.4.

Line 205: use 'see section 3.2.3'

corrected

Lines 205-206 Need to specify the strategy used for summer months.

As stated, the CI station operated mainly in summer and ParkerRidge mainly in winter so both records complemented each other. In the end the NARR precipitation is used for forcing after bias correction. The merging (averaging) is now better explained which should clarify this point

Lines 205-207: Need to show how the constant lapse rate elevation structure differs from the actual (not mean) lapse rate elevation structure. At least state their differences.

I am nor sure I follow here, sorry… The mean monthly lapse rates are shown in Supplementary material, is this what you meant?

Line 210: its should be their for data, or its for timeseries or record of said data

corrected

Line 217: Bukovsky et al. explicitly state that NARR precipitation outperforms other reanalysis products for the continental US and warn against extending this conclusion to the rest of North America. They do suggest that improvements are possible with better precipitation coverage, therefore this statement and use of the citation requires rewording and/or an additional more recent citation to be accurate.

True. We modified the sentences to 'in the US' for Bukovsky, and also refer to Chen et al 2017 for the reported good performance of NARR for 12 catchments across US and Canada.

Lines 226-257: need subsections here when methods are introduced to improve clarity. Also need divisions by variable given methods used are different in each case starting line 229.

two divisions in the text were added

Line 238: Variable Ta has already been assigned to AWS air temperature (line 187). Need additional subscripts to differentiate air temperatures from different sources/stages of processing. This needs to be corrected across the manuscript, and other variables as well.

We added 'NARR' indices to indicate that these variables are from the downscaled NARR. The whole section about NARR downscaling has been significantly reworked.

Lines 238-239: The data are downscaled to the point location of the AWS, by applying a bias correction. This precedes the downscaling of NARR node information to the finer glacier grid on which the energy balance model is run. These are not necessarily the same thing, although I believe translation of variables (without a reduction in scale) can qualify as downscaling. I recommend offering more information on the Matlab algorithm used to improve clarity.

This is an old, but helpful, reference to clarify terminology:

Hewitson, Bruce C., and Robert George Crane. "Climate downscaling: techniques and application." *Climate Research* 7.2 (1996): 85-95.

Thank you for the reference. We rather refer to Teutschbein, C., & Seibert (2012) for terminology. We now explicitly use the term 'extrapolation to the glacier DEM' (or 'model grid') while downscaling involves the two steps to go from the coarse NARR to one point (AWS), i.e. NARR interpolation and bias-correction, see new sections 3.2.4. and 3.2.5. We feel that the scaling and delta methods are standard bias-correction method. The Meteolab toolbox from Cofino et al. was used for convenience but there is nothing particular about it to explain, in fact we removed this reference as it is a distracting detail.

Teutschbein, C., & Seibert, J. (2012). Bias correction of regional climate model simulations for hydrological climate-change impact studies: Review and evaluation of different methods. *Journal of hydrology*, 456, 12-29.

Line 257: information regarding downscaling of variables to the model grid should be presented here, before the mass balance model that uses it as forcing is presented.

Done

Lines 261-262/Equation 1: I highly recommend using up to date terms for mass balance terms from the glossary of mass balance:

Cogley, J. Graham, et al. "Glossary of glacier mass balance and related terms." *IHP-VII technical documents in hydrology* 86 (2011).

We added that b(t) is the point surface mass balance to be more precise and in agreement with Cogley's terminology. We however drop the 'sfc' index for surface, for simplicity. Capitalized B is used for the specific (area-wise) mass balance. No standard symbology seem to exist for melt, precipitation and sublimation in Cogley.

Line 261: elevation not altitude

corrected

Line 261-262: extrapolation of precipitation to the model grid is in fact downscaling (from coarse NARR resolution to finer mass balance model grid). This needs to be explained before the mass balance model is presented. Additionally, the procedure for air temperature needs to be explained explicitly as well.

See my earlier answer and new sections 3.2.4 and 3.2.5

Lines 264-269: This is the method for obtaining accumulation from precipitation. This is a separate topic and should be in a separate subsection.

This has been moved to section 3.2.5 Extrapolation of NARR data to the glacier DEM

Lines 270-271: State grid resolution, present downscaled variable names (with different subscript).

Rephrased to: The model calculates the distributed mass and energy balance on each 100 x 100 m grid cell from the hourly downscaled NARR meteorological forcing data including air temperature, relative humidity, precipitation, wind speed and incoming shortwave global radiation (G).

Line 273: Melt is later reported in cm w.e. not mm w.e.

Everywhere in m w.e. now

Line 278: Citation needed for statement that justifies ignoring Qg component of energy balance.

We refer to Hock (2005) and Yang et al (2021).

Line 283: downscaled to the AWS location (not at). G is downscaled to the AWS location, then extrapolated to model grid also seems like a downscaling step (from coarser to finer resolution).

This info has been moved to sections 3.2.4 and 3.2.5

Lines 294-295: Perhaps I have misunderstood, but does the assumption of spatially constant clear-sky ratio mean that the entire glacier responds to cloud cover at the AWS. If so, a citation or justification on the scales of cloud-cover change would be appropriate and required for reproducibility.

Yes. This is standard practice in glacier models, at least for valley glaciers, to use cloud cover or more practically a cloud factor derived from measured solar radiation.

We cite Jones (1992) and rephrased to: 'The ratio is assumed to be spatially constant, which is reasonable given the large (~400 km) correlation length scale of cloud cover (Jones, 1992).

Line 298: It would be appropriate to report the maximum/mean lengths of time that the 'constant' clear-sky ratio is applied to cells that are illuminated after the AWS becomes shaded. If the AWS location is shaded and clear-sky ratio is inferred from that location for several hours at non-shaded locations, any systematic daily patterns in cloud cover could be missed (i.e. common summer afternoon clearing that is missed because all cells are assumed cloudy after the AWS is shaded). If the duration is small it should not matter, but if more than several hours I recommend a citation justifying this approach.

We calculated shading times on the glacier. We modified this segment to:

'The constant ratio was applied to 57% of the glacier surface which was sunlit while the AWS was shaded, for a mean and maximum duration of 0.73 and 2.16 hours, respectively. The impact on the radiative balance is thus considered to be small because this situation occurs in the mornings and end of days at low sun illumination angles, and also because the temporal correlation length scale of cloud cover is a few hours (Jones, 1992)'

Line 324: describe why these values are 'optimum'

Rephrased to : 'The optimum values used in the model, found by minimizing the RMSE of the simulated albedo, were $t^* $ = 14 day and $d^*$ = 3 cm.'

Line 326: LWin subscript

Corrected to: $LW_S\downarrow$ for the incoming sky longwave radiation following eq. 2

*Line 326: Explicitly describe what assumptions are used for the ice surface temperature (i.e. fixed to 0 when Ta > 0)

New information has been added on the longwave radiation section to describe how terrain radiation is calculated, as well as outgoing longwave, including the surface temperature scheme (Ts). An iterative scheme is used to lower Ts when the energy balance is negative.

Line 331: Ta is multiply defined for AWS and NARR derived air temperatures, and here is neither. Downscaling of Ta to the model grid has not been presented yet, that would be a good place to define a new Ta variable used as input following processing.

Ta is now referred to in that section to: '$T_a$ is the downscaled NARR air temperature in Kelvin'

Line 336: Downscaling of the WS variable to the model grid is absent, unless the bias corrected AWS location WS is used across the entire glacier. This needs to be explicitly detailed, and justified with literature supporting the use of local, point-scale WS data to drive a distributed energy balance model.

See my earlier answer to your major comment and the new information presented in section 3.2.5 'Extrapolation of NARR data to the glacier DEM'

Line 338: WS variable in text, WSz in equation

All z subscripts have been removed for simplicity. The introductory statement of that section states that meteorological variables are at z = 2 m.

Line 351: 'similar ice facies morphology between Peyto and Saskatchewan glacier' citation needed.

This is based on our own observations after a few field visits to both glaciers. We added '…based on our field observations'

Line 355: I recommend adding WS/direction to analysis to analysis given the poor performance compared to AWS due to local wind patterns, and the important of sensible heat flux to the overall energy balance.

The mass balance model does not consider wind directions, so, although it would be interesting this ms is already overloaded… No doubts that given the rather low correlations, there will be discrepancies in the wind directions.

Line 368: Oerlemans 2001 citation unneeded

Agreed and removed

371-372 'different greenhouse gases emission scenarios', could just be previously defined RCPs, assuming these are where the T and P values are derived from? Strictly speaking the range is similar but does not encompass (0-8 vs 2.6-8.5)

Agreed and changed. However, the +2.6 to +8.5 suffixes to 'RCP' is a radiative forcing in Wm^-2 and not temperature. Refer to our former Fig 11 (now Fig 12) for the range of temperatures changes under these scenarios.

Line 370: encompass not encompasse, would be more accurate to say approximate given previous comment

corrected

Line 372: this IPCC citation is to an annex of AR5. If this is where the values listed are obtained please disregard. Otherwise, I would recommend a reference for the primary AR5 report where RCPs are defined.

They were taken form the KNMI Atlas so we changed the reference to the AR5 report

Figure two: Adding a shading to the data gaps would help visualize them.

We added shades

Figure two: See comment for line 203 on precipitation record length. What is the origin of the precipitation data gap between 2005-2016, and what data are being used bias correct the NARR precipitation? It would be helpful to indicate, even if they are simply the values from the historical time period. It would be extremely helpful to color code the precipitation curve to show what data are from each of the PR and CI AWS stations, and what data is gap-filled from the NARR product.

See my earlier answer and modification to the text regarding the merging of both records and its use for downscaling NARR.

We modified the figure: (i) we only show the 1979-2010 record relevant for the model; (ii) we show the raw daily CI and PR data and removed the monthly and annual averages which were not robust/useful.

Line 408: the bias-corrected NARR variables, before being downscaled to the mass balance model grid

Please see previous regarding the clarification of the terminology. Downscaled NARR now explicitly refer to the interpolation of the NARR gridded data and its bias correction to the station.

Line 415: NARR Ta needs to be its own variable

Done

Line 420: I see only a very modest improvement to RMSE and MAE and no difference in R in table 1 for WS (RMSE/MAE/R 2.21/1.72/0.37 vs 2.30/1.85/0.37) and precipitation (RMSE/MAE/R 4.96/2.82/0.30 vs 5.22/3.10/0.30), making the sentence 'Validation statistics were significantly improved using the scaling technique for precipitation and WS' seemingly incorrect.

This sentence was removed and the section significantly reworked to include the new results in figure 3. This statement is now formulated as: 'The EQM method was found to improve $T_a$ best, closely

followed by the scaling method, while the scaling method was slightly superior for RH, WS and P. However, scaling only slightly reduced the errors for WS and P and had no effect on RH, which had an initial low error'

Lines 419-425: having a call to an equation earlier in text that demonstrates what exactly the scaling method is would be helpful here.

The scaling method is quite standard and simple. However it was better described now in section 3.2.4. which removes the need to burden the text with another equation

Line 433: Here the data is stated to be the 'bias-corrected NARR', inconsistent with previous statements of downscaled.

Please see earlier responses regrading the clarified terminology

Line 454: This seems plausible, but additional analysis could demonstrate it more clearly. I recommend demonstrating the difference in lapse rates between clear and bad weather days as a supporting argument to the statement.

We agree that this statement was rather speculative, but at this point we do not want to add new analyses that would distract the reader from the main scope. That sentence was thus removed but in the lights of a recent study on nearby Athabasca Glacier we proposed this tentative explanation for the diurnal cycle in wind and Ta lapse rates:

' Stronger daytime down-glacier winds, possibly driven by a larger thermal gradient between the lower ice-free valley and the glacier, could result in down-glacier cooling and correspondingly lower near-surface lapse rates, as shown on neighbouring Athabasca glacier (Conway et al., 2021).'

Figure 4: This is a very interesting and helpful figure! I think it would be important to clearly identify the data derived from the regional AWS network (lapse rates and R) and from the Saskatchewan Glacier AWS (WS and WD) to avoid confusion. Perhaps could use different line widths and explicitly state this in caption.

The data provenance was already indicated: panel a (diurnal) is all from the on-glacier AWS. Lapse rates in panel b are from the permanent stations. We modified the last statement to 'Wind speed and directions on both panels are from the glacier AWS.

Lines 480-481: I would encourage the use of subscripts, and appropriate terms from the mass balance glossary (see citation at line 261 comment) for all mass balance terms.

All seasonal terms are now in subscripts in accordance with Cogley (2011)

Lines 482-485: As previously mentioned (line 205), I recommend showing the difference between mean precipitation lapse rate from AWS network (used to downscale precipitation onto the mass balance model grid) and the elevation dependent lapse rate as a figure. Additionally, I would recommend showing temporal variability over the years with local mass balance validation measurements.

The seasonality of the precipitation lapse rates derived from the permanent weather stations is shown in Figure S1. I do not understand what is meant here by elevation-dependent lapse rate. Do you mean showing a plot of the mean temperature against elevation?

As for the temporal variability over the years with local mass balance, isn't it already shown on Figure 6? The point colours refer to years of observation/simulations

Line 485: explicitly define what is considered 'good model performance', or remind the linear distribution of model vs measured b values if that is the case.

Added:

'…as seen by the linear relationship between observed and simulated *b* and the high NSE values,…'

Line 494: m^-1 missing in units

corrected

Line 495: use exact language when comparing study balance ratios to those from previous work: the balance ratio is triple (3x) that computed for the region by Rea 2009. This seems anomalous and should be explained further.

This segment was reworked, with additional interpretation given for the BR higher than typical eastern Rockies. The higher BR is thought to reflect the fact that the Columbia Icefield has a locally wetter climate, more akin to coastal western glaciers. However, this interpretation was moved to the discussion (section 5.4 on energy balance regime)

Line 510: List the results in text that constitute 'good general agreement'.

Rephrased to: 'The cumulative mass balance simulated with the multitemporal DEMs agrees well with the geodetic estimates'

Lines 505-516: This section needs to include the geodetic estimate results to have a number to compare the model balance to.

I am not sure I fully understand. Figure 7a (former Figure 6 b) clearly shows the geodetic and simulated mass balance… To add some hard numbers we added this: 'The simulated and geodetic cumulative $B_a$ were -26.79 m w.e. and -25.59 ± 8.44 m w.e., respectively, for 1979-2016.'

The order of the information has also been reorganized in a more logical way in that section

Line 518: Unclear if Ta and P are AWS or NARR variables because of same variable term

NARR subscripts added

Lines 520-522: The difference between mean and elevation dependent precipitation lapse rates needs to be addressed here.

See my earlier response. The 'mean' lapse rates refer to temporal averaging, not spatial (elevation). Mean monthly precipitation rates were first computed, then the average of the November-March values when snowfall is most abundant on the glacier was used. Does that clarify?

Figure 6: An excellent and helpful figure. Consider renaming the Y labels to cumulative MB. Additionally, choice of colors in legend does not correspond to colors in plot, likely because of overlain curves with transparency. This makes it hard to decipher Z0snow and ice albedo. Change to better color scheme.

Thank you. Labels were renamed. Note that former Fig 6 is not split into new Figure 7 (mass balance validation against geodetic and effect of dynamical adjustment and 8 (sensitivity analyses). The sensitivity envelopes for albedo and Z0_snow do overlap, so it is impossible to show both, hence we used the merged color as a single legend symbol for z0ice/ ice albedo. The cumulative uncertainty is visible on the right axis, thought.

Line 539: Sensible heat flux is now defined as QS, it was initially defined as Qh at line 277. These terms absolutely must be consistent throughout to avoid confusion.

Corrected, Q_S is used throughout now

Line 547: Is LW* net longwave? The text refers to it as radiative cooling, which I assume means outgoing longwave exceeds incoming longwave?

Yes, that it what is implied. LW* was indeed defined as net longwave, so negative LW* means the surface is cooling radiatively

Lines 576-578: This should be just melt, not ice melt if I understand correctly that we are above the snowline and therefore no ice surface is exposed in the upper accumulation regions (and all values are in cm w.e.)?

Correct, 'ice' was removed

Figure 9: Scale bar needs rescaling on panel A

Colour scale was rescaled. Units adjusted to m w.e.

Line 601: Conversaly - > Conversely

Corrected

Line 606: were overlapped -> were overlain

corrected

Line 607: 'given by the latest projections from climate models' redundant

This segment was removed

Line 610: m w.e. -> cm w.e.

All mass fluxes are now in m w.e. throughout the text

Line 613: Static and dynamic mass balance values should be clearly reported and defined with separate subscripts at some point in the manuscript

We clarified the terminology and symbols. In the method section 3.1 we added this:

The term 'reference mass balance' (Ba_r) is used hereafter to refer to glacier-wide mass balance simulated with a fixed reference geometry while the term 'conventional mass balance' (Ba_c) is used for the simulation with adjusted glacier geometries (Huss et al., 2012).

Figure 10: Previously stated Temperature range was 0-8 C in methods, now 0-7 C. This does not technically encompass RCPs of 2.6-8.5C

RCP refer to increases in radiative forcing, not temperature, see my earlier response

Line 645: typo

corrected

Line 631-646: A table would be extremely helpful to navigating these results

We added a new table, as well as a new short section above it, summarising the apportioning of different processes to the simulated mass balance sensitivity to warming (Table 1)

Line 658-659: reword attractive to a word that is specific to the gain (i.e. efficient, affordable, etc)

Changed to 'useful'

Line 660: this has been stated and cited already

Sentence removed

Line 662: simple is misleading considering the breadth of statistical downscaling methods. I recommend specifying the exact methods referred to here. Additionally, Clarke et al. 2015 used dynamic downscaling of their surface precipitation to drive their mass balance model and is improperly cited here.

The word 'simple' was removed. Clarke et al (2015) is based on Jarosch et al (2012) (which we now quote). They used linear orographic theory to downscale NARR precipitation, so while elaborate this is not to my understanding dynamical downscaling, which involves an atmospheric or climate model at higher resolution than the boundary product (here NARR).

This whole section was reworked to outline the main methods used by previous studies, in order to better place our approach and results in this context, see section 5.2

Line 666: Same as above: statistical downscaling can include non-linear regression with multiple predictors, as well as the use of scaling factors, etc, with some methods not having calibration steps. Specify exact methods.

Done

Line 668: replace 'reasonably good' with exact numbers

Correlation coefficients were added in parenthesis

Lines 674-679: With respect to wind speed, the authors convincingly argue that glacier-scale katabatics that are not captured on the coarse NARR grid account for poor performance (line 410-412), and identify the impact of poor wind speed variable performance on temperature lapse rates (section 4.3). However, the wind speed variable performance is only further discussed as suffering from katabatics (lines 674-679), and the impact of this poor performance on turbulent heat fluxes computed in the energy balance model seem absent. Furthermore, the results demonstrate that the sensible heat flux plays an important role in the overall energy budget of the glacier surface. I recommend further comments on the possible consequences of poor wind speed data forcing the model, and would

suggest adding this variable to the earlier forcing variable sensitivity tests. Additionally, the impact of using point scale WS data to drive an entire glacier-scale energy balance model must be explicitly justified.

We performed a new sensitivity analysis to NARR forcings, see new Figure 8b and related method description and description of results. In brief, the correlation of NARR wind speed with AWS observations, the simulated mass balance does not appear very sensitive to switching between WS_NARR and WS_AWS. The model is most sensitive to air temperature, despite the good correlation and low errors between Ta_NARR and T_AWS. This is now discussed in the discussion section.

Line 694: biased -> raw

Changed to:

'The underestimation of precipitation in NARR combined with the positive bias in the raw NARR global radiation mostly explain the exaggerated mass loss simulated by the mass balance model when forced with the lapsed NARR, i.e., when only precipitation and temperature are lapsed to the reference station elevations.

Line 697: list all assumptions explicitly for clarity

Changed to:

'Despite the physical nature of the model some assumptions remain simplistic, such as a constant precipitation lapse rate and spatially invariant ice albedo, aerodynamic roughness and wind speeds. Despite these limitations, the interannual variability in mass balance was relatively well simulated by the model, with NSE values of 0.83 to 0.91 for direct point observations (Figure 6).'

Line 700-705: present errors as fraction of modelled results to give context. Discussing an error of 78 cm w.e. a-1 is different from an error representing +/- 108% (77.6/-0.72 m w.e.).

Changed to:

'The root-mean-squared-error (RMSE) on the simulated bw is 0.24 m w.e. a-1 (median relative error of 15%)  – on the same order as the typical measurement error for snow and firn, which gives a RMSE values, however, are higher than typical measurement errors for bs (0.87 m w.e. a-1, relative error = 22%) and ba (0.77 m w.e. a-1, relative error = 24%)...'

Lines 704-705: Significant figures for this result are different from previous ones in the results section (different units too), make consistent.

Al units are in m w.e. for absolute metrics.

Line 707: list numbers corresponding to these results for clarity and context

Done: 'The simulated cumulative mass loss (-26.79 m w.e.) was close to the geodetic estimate (-25.59 ± 8.44 m w.e),...'

Lines 711-713: Again, I highly recommend assessing the sensitivity of the model to Wind Speed given the poor performance of this variable and the importance of fluxes using Wind Speed to the interpretation of the results.

See my previous replies

Line 717: glacier models -> glacier mass balance models or energy balance models or glacier surface process models given that 'glacier models' includes dynamic models from the bed to the surface.

Agreed. Corrected to glacier mass balance models

Lines 748-757: a table would help navigating this discussion point

See new table 1 and updated discussion for better clarity

Line 782: m w.e -> cm w.e.

Why? The mass balance sensitivity to precipitation is near 1 (1.01), which is in mass/mass, so m w.e. / m w.e.

Line 782-783: 'A value of 1 would occur if all precipitation were snowfall and there were no albedo feedback.' -> specify that these values are ratios and unitless (the previous sentence does so partially). The Oerlemans 2001 reference here seems generic and unneeded.

Oerlemans 2001 was removed. We replaced (m w.e. of mass change per m w.e. of precipitation change) simply by (unitless) in the previous sentence

Line 786: units are now in mm

All in m w.e. now

Line 787: and now in m w.e.

All in m w.e. now

Line 798: MacDougall and Flowers warn that transferring model parameters between sites in a specific setting will result in increasing error depending on the number of parameters and inputs carried over. This statement inexact resulting in an improper citation.

I respectfully disagree. From MacDougall and Flowers:

"Transferring parameters in space or in space and time together produces similar results with significantly higher errors than in the control runs (Figs. 3i–p)" (p. 1492).

"Transferring parameters in space or in space and time produces large differences in the modeled distribution of ablation and, consequently, in the calculated mass balance". (p. 1492)

They later show that by fixing some of the parameters to local values (summer snowfall, initial snow depth and ice albedo) the spatial transferability of other parameters was more successful.

From their conclusions: "Our results suggest that caution must be exercised when transferring melt models in space." (p. 1497)

So I think this citation is well suited to warn about parameter transferability in energy balance models and the need to constrain local parameters…

Line 800: I believe this has been done before? Please include citations to identify where this work improves the existing work

As far as I know mountain-range scale modelling using downscaled reanalyses did not use in situ station data for downscaling (eg Clarke et al, 2015). We have reworked the conclusion to better bring out the novel aspects of the study.

Line 804: please include geodetic estimate here for clarity

done

**Reviewer 2**

Overall assessment:

The manuscript makes the most out of a limited but valuable data record, bias-adjusting NARR climate renalyses from 1979-2016 to develop a point-scale meteorological forcing relevant to the AWS location adjacent to Saskathewan Glacier. This point 'data' is then distributed or extapolated over the glacier to force a surface energy balance model for glacier melt. Combined with NARR-based precipitation estimates, mapped onto the glacier using similar methods (bias-adjusted, then distributed over the glacier using a lapse rate), this provides an estimate of glacier mass balance from 1979-2016. Perturbation methods are used to assess energy and mass balance sensitivity to increases in temperature and precipitation, representative of future climate change scenarios.

There is good potential in this manuscript and I found the methods to be mostly well-explained, although I agree with R1 that inconsistencies in terminology, notation, and units need to be improved. Many previous studies have used bias-adjusted climate reanalyses or climate model output to force distributed surface energy and mass balance models on a glacier, so this is not particularly novel or innovative. However, there are several judicious choices that seem to me well-judged, such as the assignment of glacier albedo and its variability (uncertainty), based on satellite images, the inclusion of available precipitation data to inform the NARR bias adjustments, and the introduction of diurnal temperature lapse rates on the glacier based on the vertical transect of HOBO temperature sensors. Available in situ data is limited but is well-leveraged. The numerical experiments are well-designed and nicely visualized, and the authors reach significant and (mostly) well-supported conclusions. This is a significant and interesting outlet glacier of Columbia Icefield, an important glacier system in the Canadian Rocky Mountains which nonetheless lacks in situ surface mass balance data. I therefore think this study has potential value, and would be interested to see a revised manuscript that addresses several of the major concerns of R1. I have several additional points that also need to be addressed. Within these, it may just be that I did not understand or follow the authors' methods. In some places though, I am not confident in the results (see points 5 to 8) and would encourage a careful re-examination. In particular, the reported AWS wind speeds are implausibly high and the modelled turbulent fluxes also do not seem correct; sensible heat fluxes are extremely high and the finding of positive latent heat flux is possible but unsual. The values reported here are not typical of mid-latitude glaciers in a continental environment (point 8).

Thank you for taking time to thoroughly review our manuscript and for your constructive comments. We reply in detail to each of your comment below.

Major Concerns

1. I don't think it is unreasonable to use the single NARR grid cell that coincides with this glacier, given the 32-km scale of NARR grid cells and the good correlation with available ground T, RH, and shortwave radiation data. I do wonder why NARR incoming longwave radiation data was not used in this study though, for consistency. Was this considered, or compared with the longwave radiation parameterization that is used? The parameterization requires cloud cover fraction that is taken from NARR, so why not just use incoming LW from NARR, which appropriately considers the 3-hourly vertical temperature, humidity, and cloud structure, vs. a parameterization that only uses near-surface values?

We now compare two NARR interpolation methods based on suggestions made by Rev1: nearest neighbour (closest NARR gridcell), and bilinear interpolation for the 9 (3x3) closest gridcells. We find that when using raw NARR variables, i.e., only correcting temperature and precipitation for elevation differences using fixed lapse rates, the bilinear method gives slightly better results against AWS observations and for reconstructing mass balance against geodetic observations. However, when bias-correcting the NARR forcings with station observation, the choice of interpolation method has marginal effects.

Regarding the incoming longwave radiation: thank you for spotting this. The Konzelmann parameterization described in former section 3.3.3 of the methods is not what was used. This was an earlier application of the model, and as you correctly spotted this is a crucial aspect that we in fact had carefully addressed! Here is what was done:

We did not use the raw NARR incoming longwave (LWin) because these would carry an elevation bias, since the closest NARR gridcell is 237 m higher than the AWS. NARR LWin would therefore be too low. We did not have observations of LWin at the AWS to correct this bias directly. Instead, LWin was calculated from the bias-corrected NARR air temperature (Ta_NARR) at the AWS location and the atmospheric emissivity (E) from NARR, taking terrain effects into account: LWin = F*theta*E*Ta^4, where theta is the Stefan Boltzmann constant and F the skyview factor. The result is a LWin that is almost equal to the raw NARR Lwin, but with a slight correction for the difference in air temperature between the NARR gridcell and the AWS. The key point is to use the NARR emissivity, which reflects the vertical humidity and cloud structure. Section 3.3.3 was corrected accordingly, and panel f of figure 3 now shows the NARR emissivity instead of the cloud cover.

The authors note that ERA5 was not available at the time of the study, but it has been out for more than one year now - about 1.5 years I believe, and offers 1/4 degree resolution with hourly data, which can avoid some of the complexities and assumptions in mapping the 3-hour NARR data onto hourly estimates. ERA-land is even higher resolution. I would not insist on this, as it is a lot of additional work, but I think it would be valuable and would strengthen the manuscript a lot to explore model results with ERA5 and/or ERA-land as well - this could very much help to test the robustness of the results and conclusions, and lower the reliance on NARR's relatively unproven veracity in the complex mountain terrain. This could be follow-up work, but I also have the nagging sense that this study risks being already out of date. The argument that ERA5 precipitation is questionable is not really valid, as all of the reanalysis output is bias-adjusted and NARR is weak in this respect.

As you say, comparing with ERA-5 would be interesting but would entail too much work at this point, so it will have to be left for follow-up work. However, it is worth mentioning here that ERA5 has a 0.25

degrees resolution (~ 28 km) vs. 0.30 degrees for NARR (~32 km) so the difference in spatial resolution is rather small. Most importantly perhaps, all atmospheric forcings for the ERA5-**land** product (0.1 degree) are interpolated from ERA5, so there is no gain in spatial resolution for the atmospheric variables, but only so for land surface variables. We think that the methodological framework used in this study could very well be re-applied with ERA5 in follow-up work, and this is now mentioned in section 5.1 of the discussion.

2.  I agree with R1 on the confusion regarding terminology in the methods. As I read it, the NARR output is not downscaled to the AWS; this study uses bias-adjusted rather than downscaled NARR output, as I understand it.  Bias-adjusted NARR fields are then distributed over the glacier from a reference site (the AWS) using locally-relevant lapse rates and sophisticated methods for the incoming SW radiation. As I understand it, the reference site for the precipitation differs (e.g., Parker Ridge), but it is a similar approach. I am happy to leave it to the authors to decide how they would like to refer to this process (extrapolated, downscaled, lapsed, or distributed over the glacier), as long as it is clearly defined and consistent in the manuscript. It is not what I think of or what is commonly referred to as downscaling though.

    We addressed the terminology in response to reviewer 1:

‘Downscaling’ had been used as a general term for any transfer function that translates a coarse climate field (reanalyse, RCM, GCM, etc.) to a local meteorological record (here an AWS). Bias-correction is one, and the simplest downscaling strategy (Teutschbein & Seibert, 2012). However, we agree that the workflow may not have been clearly outlined, so we reorganized and re-segmented the methodological workflow as per your suggestions.

To clarify, the downscaling term implies two steps that are now clearly outlined in the new section 3.2.4 ‘Downscaling NARR to weather stations’: **(**1) interpolation of the NARR gridded data to the reference weather stations; (2) bias correction of the interpolated NARR data.

A new section 3.2.5 ‘Extrapolation of NARR data to the glacier DEM’ now describes the strategies and processing steps involved in distributing the downscaled NARR to the glacier DEM. All lapse rates calculations are now included in this section.

3.  I admire the use of the regional network of permanent weather stations to develop temperature and precipitation lapse rates, but I worry about the relevance of these values to the glacier itself. These are all off-glacier sites with a maximum elevation of 2025 m, while Saskatchewan Glacier extends from about 1800 to 3300 m. Glacier near-surface temperatures (and the surface energy balance that influences these) are very specific, as is the snow accumulation regime on glaciers and in the unsampled elevation band from 2025-3300 m. I don't have great confidence in the applicability of the lapse rates as determined by the off-glacier climate station network. For temperature, why not use the average daily or monthly lapse rates as determined by the HOBO temperature transect?  I realize that these are summer-only, and the data are limited, but this is what is used for the diurnal lapse rates so this would seem relevant and consistent. Winter temperature lapse rates are not important to the glacier melt, so could be assigned an average or May value.  For the precipitation lapse rates over the glacier, is there a way to use available winter mass balance data (in situ and/or LIDAR-inferred) to look at this?  The current precipitation lapse rate may be appropriate, but it would be helpful to constrain and evaluate this, as well as the

assumption of a sustained (and strong) linear increase in precipitation across the icefield plateau from 2800 to 3300 m.

This is a common issue, to rely on lapse rates calibrated with lower-elevation stations. Your preoccupation is warranted, though. We thus checked the mean on-glacier lapse rates derived from the Hobos on the ablation stakes for their available period from May to August 2015, which yields a value of -0.46°C/100 m, versus the average from the permanent network for the same months: -0.49 °C/100. This rather similar value gives us confidence that the lapse rates extrapolate well to the AWS site. When considering the mean lapse rate from the permanent network for May to October, i.e. the months during which the toe of the glacier (1784 m) is above the zero-degree isotherm (Supplementary Figure S1a), then the mean lapse rate is -0.47 °C/100. Hence replacing the mean monthly lapse rates by the mean lapse rates from the Hobos would not significantly change the results. We also want to recall that diurnal anomalies in the temperature lapse rate derived from the Hobos were superimposed on the seasonal lapse rate cycle derived from the permanent network, to more realistically simulate diurnal changes in the lapse rates not captured by the permanent network.

We added this segment to section 4.3:

*'The mean on-glacier summer (May-August) lapse rate (-0.46°C 100 m-1) was very close to that calculated from the permanent weather station network for the same period (-0.49 °C 100 m-1), which gives confidence in extrapolating the monthly lapse rates from the network to the glacier surface. Superimposing the on-glacier diurnal lapse rates anomalies onto the mean monthly lapse rates allowed to better represent the diurnal changes associated with the glacier wind.'*

As for the precipitation lapse rate: this is harder to constrain, as in other studies, as there are no high elevation precipitation gauges. The Columbia/Parker Ridge stations (2000 m) are actually pretty high already compared to typical stations in the area. The sensitivity of the modelled mass balance to this lapse rate was tested in former Fig. 6c (now Fig 8d): the sensitivity is expectedly high, which highlighted the need to constrain this value as best as possible from ancillary observations. We found that the lapse rate constrained on the permanent network gave good results: the performance, as you suggest, can be specifically assessed against the end of winter mass balance data (bw), which was done in section 4.4 and former figure 5a (now Fig 6a): the lapse rate used yielded satisfactory bw simulations up to the highest stake/snowpit, which gives confidence on the lapse rate used. We did discuss the limitation of using a time-invariant lapse rate, which gave poorer results in the dry 2016 year (Fig6a), but now emphasize this point better in the discussion, especially mentioning the unknown biases for the highest reaches of the glacier not covered by snowpits/stakes. This area only represents 8.8% of the total glacier area. These considerations have been included in the manuscript.

4. The precipitation lapse rate that is used is based on the reference climate station data from November to March (l.206). This does not coincide with the accumulation season on the glacier, which is more like September to May. Is this same precipitation lapse rate used for April to October, and is there objective support for that? This needs to be discussed and addressed, perhaps with an examination of the primary data or perhaps by bringing in the winter mass balance data from the glacier, if there is some from the 2014-2016 study. November to March is relevant for the lower-elevation snow season, but not that of the glacier, where autumn and spring often bring a lot of snow.

The current model code only allowed using a single value for the precipitation lapse rate, which is a limitation. We sought the most representative winter value that influences most the accumulation period. The precipitation gradient is best defined (strongest relationship between precipitation and elevation) during November-March which is also the period when most snow accumulation occurs, i.e. when the whole glacier is above the zero-degree isotherm (Fig S1). The lapse rate value for this period (mean = 15.6 % per 100 m) yielded satisfactory simulations of the end of winter mass balance (Figure 6 a), but we do show that there is a high sensitivity of the simulated mass balance to the uncertainty in this lapse rates (± 4%/100 m: see figure 8c and discussion). In fact, this was a strong point made in the paper, that it is crucial to constrain this lapse rate to obtain good mass-balance simulations, but we realize that we did not express it clearly enough, so its is now better emphasized in the discussion. I also agree that snow accumulation occurs from September to May, for the higher parts of the glacier, and that the lapse rate starts decreasing in these shoulder seasons (April-May and September-Oct.: Fig. S1). Yet the bulk of snow accumulation is rather from Nov. to March and the ± 4% sensitivity used in Figure 6 captures the uncertainty of adding Sept-Oct and May in the mean value). Most importantly, the value used yielded good end of winter mass balance, which is dominated by accumulation.

5.  Wind speed results on ll.393-395. These are extremely high average wind speeds, an annual average of 16 m/s and up to 23 m/s in February. I appreciate this is likely a windy site, and there are katabatic winds here, but those are typically stronger in summer. Are the authors confident that these units are correct - is this perhaps km/hr, or are these maximum (vs. mean) wind speeds that are reported here and plotted in Figure 2?  An average monthly wind speed of 23 m/s equates to 83 km/hr, which is not plausible.  Values reported and used later in the manuscript (e.g., from NARR, means closer to 5 m/s) are more reasonable. I would also add that I have spent some time on this glacier, and there is a steady and reliable down-glacier wind, but not of the knock-you-over variety.

Thanks for spotting this. Indeed, quite high… The wind speeds on Figure 2d were in km/h, not m/s as labelled, but they had been correctly converted to m/s before bias-correcting the NARR wind speeds. This is why the resulting NARR forcings are within expected values, as you spotted correctly (former Fig3, now Fig.4.)  Figure 2d was modified to present the wind speed in m/s rather than km/h.

6.  I am not sure what 'homogenized' means here in the context of the observational precipitation records that are spliced. Homogenized has a very specific meaning for meteorological data sets, involving corrections for discontinuities associated with station moves or changing conditions/instruments/methods at an observation site.  The precipitation data also seem to have a lot of gaps, which makes me worry about the time series of mean annual values. It seems best from about 1972 to 1994, not for the full period plotted in Figure 2. What methods were used to gap-fill this data for missing months?  Apologies if I missed this. My sense is that it would be best to use these data for long-term mean monthly values from 1979-1994, using all available monthly data over this period. This can then inform a bias-adjustment of NARR mean monthly values for the same period, 1979-1994. Then go with bias-adjusted monthly NARR (or ERA5) precipitation for the study. Just my surficial thoughts on inspection of the observational data in Figure 2.

We rephrased to 'merged' instead of 'homogenized'. Yes, there are many gaps, as discussed in the MS. Recall that we used the daily observations from the 'merged' Parker Ridge/Columbia station to bias-correct the daily NARR precipitation, using all common days with available data. So we did not gap-filled the observations, we used whatever was available to bias-correct the NARR data on a daily time scale. The numerous gaps were the prime reason to use the NARR precipitation as forcings. We have made this clearer in the revised text. Asrev1 also requested changes to the figure, the new Figure 2 now

presents the raw daily precipitation records from Parker Ridge and Columbia, since the monthly and annual values derived from them did not make much sense…

7. Perhaps my most significant concern: the sensible heat fluxes seem far too high for a mid-latitude continental glacier, and compared with other data from the region (Peyto, Haig Glaciers). Also, it is surprising and unusual that latent heat fluxes are positive. I don't trust either of these results. Are the erroneously high winds speeds (point 6) the reason for this? This could explain the high values of sensible heat flux, though it is still odd that latent heat flux is positive. What is the basis for determining the snow/ice surface temperature in these calculations? This is critical to the turbulent heat flux calculations, and I did not see a discussion of this in the paper - apologies if I missed it. Is a melting glacier surface assumed in the summer? What is assumed through the rest of the year? I wonder too if the snow roughness value is appropriate for winter conditions - 6 mm is high, perhaps more reflective of sun cups than the smooth winter and spring snow surface. Snow roughness values closer to 1 mm are commonly adopted in glacier modelling. The sensitivity to this variable could be more thoroughly explored, perhaps considering order of magnitude rather than \pm 1 mm variations.

First, the wind speeds are correct, as mentioned in point 7. We were also a little surprised at first by the large sensible heat fluxes and the small but positive latent heat fluxes. We have double checked all calculations, and everything appears to be correct. We think they are correct and that the climate of the Columbia icefield (as you may have experienced) is actually quite different than Peyto and the other glaciers you refer to, which have more continental climates. The icefield is higher and has its own weather, more humid than Peyto, and Haig. This is why the Columbia icefield is often wrapped in clouds while the valley is clear (and explains why one can get stuck a long time on the icefield waiting for a helicopter while it is sunny below the icefield…). But you are correct that this needs to be better discussed, and thank you for suggesting references. Some were already quoted, and some were now added.

You are right about surface temperature (Ts), we did not mention how they were calculated. This is now done in section 3.3.3:

*'Outgoing longwave radiation (LW↑) is calculated from the Stefan Boltzmann equation and the simulated surface temperature (Ts). Ts is obtained in an iterative process by lowering surface temperatures in case of negative energy balances until the energy balance equals zero (Hock and Holmgren, 2005).'*

Regarding the roughness values, we used the closest analogs on Peyto (Munro, 1989) to guide theses parameters. The snow value is indeed on the high side but still closer to 'glacier snow' values (eg Brock et al., 2006) during the snowmelt period. Since we used a time-invariant value, a value close to metamorphized snow is better than mid winter snow when ablation is limited. We have expanded the range for sensitivity analyses of z0 to ±one order of magnitude (updated figure 8). The simulated cumulative balance is expectedly very sensitive to such a broad range. We now discuss this. While a one order magnitude range is possible across a glacier surface, the uncertainty around the mean value (for ice/snow) will be much less than the spatial variability.

About the high contribution of sensible flux. We believe that the often-reported high contribution of net radiation to melting energy in the several past studies is biased by the season (values are commonly reported for the ablation season only – typically July/August – when net radiation is high), and to the

ablation zone of glaciers, where most micrometeorological studies have been done and where again net radiation is higher due the lower albedo.

For example, at the location of the AWS (midway up the ablation zone), the sensible heat flux contributes 32% of the melt energy in July and August, while net radiation contributes 57% and latent heat flux 11%. So, while this is higher than common values for mid latitude temperate glaciers with a continental climate, these values are not uncommon in more humid climates (see compilation in Smith et al 2020). Sublimation does happen during the day, but deposition occurs at night when the glacier surface cools (Figure 10b) and this dominates the overall latent heat budget.

We have included a whole new section to the discussion ('5.4 Energy balance regime') to discuss the energy balance results in the context of other studies on mid latitude glaciers, which addresses your comment.

8.  AWS snow accumulation is reported on l%.485. How is this recorded?  Isn't this just a tipping bucket rain gauge at this site?  Or is this based on measurements from site visits?  It would be helpful, per the comment above, to report winter snow accumulations on the glacier and use this data to help evaluate the precipitation modelling.

Using an ultrasonic gauge on the AWS. It is now mentioned in the. End of winter mass-balance (bw) at stakes and snowpits/soundings between stakes are presented in Figure 6a.

9.  Agree with R1 that it will be really helpful to use mm or m w.e. throughout for mass balance, rather than a mix of mm, cm and m.

Agreed, all mass balance are now reported in m w.e.

10. l.494, the balance gradients. Please give in units of m or mm w.e. per m. This is an interesting result, though I worry that the unusually high value (steep gradient) on the upper glacier is in part due to the unconstrained precipitation/accumulation gradient on the glacier. Looking at the available data in Figure 5 from 2015 and 2016 (2014 data are not sufficient), it would be hard to justify a bi-linear vs. linear relationship for ba. This is purely a model result then, as I understand it - can it be explained via mass balance processes here, since the authors describe the balance ratio as an unusual result?  Is it reflected in the geodetic mass balance profiles?  This is a significant point, as the balance gradients (values, linear vs. bi-linear) are potentially significant for regional-scale mass balance modelling - I can imagine other authors using the values that are reported in this study.  Perhaps it is early to think about this too much, as the results may change upon revisiting of the wind speeds, modelled surface temperatures, and modelled turbulent fluxes.

It is true that the modelled gradient is not constrained by point mass balance beyond 2900 m. Geodetic gradients include ice flux, which would complicate deriving mass balance gradient from them, no? The area of the glacier above 2900 m represents only 8.8 % of the total area, so extrapolation errors in this unsampled area would have a small impact on glacier-wide balances. Yet I agree that we should perhaps not use the modelled gradient beyond the last validation point at 2900 m. We have recalculated the gradient from the ELA up to 2900: this slightly changes the gradient, from 0.31 cm w.e./m (whole accumulation zone) to 0.32 for the period of observation, and from 0.29 to 0.31 for the long-term (1979-2016) period. The revised balance ratio is 3.10 in place of 3.34, still high compared eastern Rockies

'drier' glaciers, and closer to 'wetter' West coast glaciers. Values and calculation methods were updated in the paper. This point, along with the larger simulated sensible heat flux and positive latent heat flux, suggest that the weather of the Columbia icefield is indeed distinct from the more continental types glaciers of the Rockies. This point is emphasized in the discussion (section 5.4).

---

## Referee Report (RR1)

**Intro paragraph:**

The manuscript *Modelling glacier mass-balance and climate sensitivity in a context of observations: applications to Saskatchewan Glacier, western Canada* addresses the research question of glacier mass balance modelling with sparse observations and the applicability of reanalysis climate model data to simulate glacier mass balance and its sensitivity to future climate. The authors break down the components of the surface energy balance to understand the sensitivities of glacier mass balance to future climate scenarios represented by the IPCC RCP scenarios. The modelling work in the manuscript have previously been used but the novel aspects of their work assess how a continental mountain glacier will respond to a warmer and wetter/drier climate future. They use the case study of Saskatchewan Glacier and aggregate several in situ datasets to calibrate and validate their model results.

The handling of the precipitation gradient remains a concern for the study as they use off glacier precipitation data and over smaller elevation range to derive their precipitation lapse rates. However, considering the limitation of in situ data, the authors do a good job on constraining their errors and discussing the limitations of outlined methods. Overall, the manuscript adds progress to reconciling glacier mass balance with sparse and in consistent datasets combined with remote sensing data to achieve holistic results of the future of mountain glaciers in a warming climate.

**General Comments**

-Throughout the manuscript there was no mention of handling of inversions for the calculated lapse rates from the AWS on the glacier and from the permanent weather stations. Is this due to the authors not finding the occurrence of inversions in their study area. Please add in the discussion the implications of inversions in the calculated lapse rates.

-Elevations of the permanent weather stations barely covers the elevation gradients over Saskatchewan Glacier. The authors do a good job of discussing this and pointing out that the higher elevation above 2900 m represents only 8% of the accumulation area and therefore has small impact on the overall simulated mass balance. But it remains a weakness of paper. In the discussion it would be prudent to compare results of precipitation downscaling from other studies such as Jarosch et al. 2012 to understand if more complex methods would better resolve precipitation trends for a further justification of the use of a statistical downscaling method.

-Presentation of the results between fixed and dynamic glacier mass balance results remains unclear throughout the manuscript. Earlier on when discussing the topographic data, it should be mentioned the negligible effect of the conventional glacier simulation and therefore only the reference mass balance simulation results are presented for final glacier mass balance results.

**Specific (Line by line):**

Title: Modelling glacier mass-balance and climate sensitivity in a context of observations: applications to Saskatchewan Glacier, western Canada -> Modelling glacier mass-balance and climate sensitivity in a context of sparse (or limited) observations: applications to Saskatchewan Glacier, western Canada

Line 19: was little -> was a little

Line 120: (ii) should this objective also include the air humidity and albedo feedback as they are the major conclusions of the paper?

Line 89+120: spare -> sparse

Figure 1: Reduce the interval of labeled contours. Increase font size on Fig. 1c legend. It is not immediately clear the location of the air temperature points, since the color of the star is overlapped with the snow survey points – change the symbol of the air temperature point or increase the size of the symbol.

Line 195: Why were the precipitation records from the other five permanent weather stations not used?

Line 209: State the temporal and spatial resolutions of ERA interim and NCEP reanalyse products.

Line 337: te -> the

Line 353: Says depth scales was calibrated with snow depth at AWS but section 3.2.1 does not describe recording snow depth measurements. Although the supplementary material describes snow depth sounding measurements. Clarify where the snow depth measurements are coming from.

Line 358: bias correction -> downscaling?

Line 419: I think this should be from 0 to 7 °C to be consistent with results and abstract.

Line 421: Define GCM at first mention

Figure 5: Are the values correct for relative wind direction on Fig. 5a? If so, why do they vary from the monthly wind directions?

Line 546: Mention the ultrasonic snow depth sounder in section 3.2.1

Figure 6: check the figure caption for correct lettering of figure numbers

Figure 6d: Shows the limitations of the precipitation gradient since the gradient derived is not within the same elevational ranges and should be discussed further as per previous comments.

Line 580: Dynamical adjustment explanation should be explained in the methods somewhere between lines 180 and 185.

Line 595: The use of lapsed interpolated should be reworded for clarity to 'lapse rate corrected'.

Line 611: 'an even more so ice surface morphology' reword to clarify if you mean that the surface morphology is more uneven than the snow surface or less uneven.

Line 800: Include that there was difference in elevational ranges used for precipitant gradient compared to the elevation of the Sask. Glacier.

Line 941: The air humidity feedback is one of the main findings from the paper, expand on the implications for glacier mass balance at Sask. Glacier with increasing atmospheric warming from this feedback.

**Supplementary Martial:**

- 'Errors in glacier outline delineation were not considered' Please provide justification for why they were not considered.

-Figure S1: interesting to see Lake Louise precipitation data included here. Did you compare the record with Columbia and Park Ridge to show the variation? Include a few sentences to say why the Lake Louise and other precip record was not included in the study.

----------------------------------------------------------END OF REVIEW--------------------------------------------------------

---

## Author Response (AR2)

**Response to reviewers**

**Anonymous referee #3**

We thank the anonymous reviewer for his review and constructive comments on our work. We reply to his/her general and specific comments below.

**General Comments**

-Throughout the manuscript there was no mention of handling of inversions for the calculated lapse rates from the AWS on the glacier and from the permanent weather stations. Is this due to the authors not finding the occurrence of inversions in their study area. Please add in the discussion the implications of inversions in the calculated lapse rates.

Reply: we did not have vertical temperature profiles to address this directly. But from the on-glacier temperature sensors distributed across the glacier we found no evidence of systematic inversions (i.e. positive lapse rate) during summer (May-August, see our Figure 5a). Outside the glacier, monthly lapse rates were also consistently negative.

We checked lapse rates calculated on an hourly basis on the glacier. Inversions were found to occur only 1.7% of the time during the May-August period. We added this information in section 4.3 (lapse rates):

Stronger daytime down-glacier winds, possibly driven by a larger thermal gradient between the lower ice-free valley and the glacier, could result in down-glacier cooling and correspondingly shallower near-surface lapse rates or **even inverted lapse rates**, as shown on neighbouring Athabasca glacier (Conway et al., 2021). **Closer inspection of hourly lapse rates revealed that inversions only occurred 1.7% of the time between May and August on Saskatchewan Glacier and that the mean diurnal cycles represented well the bulk of lapse rate variability.**

-Elevations of the permanent weather stations barely covers the elevation gradients over Saskatchewan Glacier. The authors do a good job of discussing this and pointing out that the higher elevation above 2900 m represents only 8% of the accumulation area and therefore has small impact on the overall simulated mass balance. But it remains a weakness of paper. In the discussion it would be prudent to compare results of precipitation downscaling from other studies such as Jarosch et al. 2012 to understand if more complex methods would better resolve precipitation trends for a further justification of the use of a statistical downscaling method.

We added the following section to the discussion:

The station-free, linear orographic model for precipitation (LOP) method used by Jarosch et al. (2012) might perhaps be better suited than station-based downscaling in steep topography. The authors reported an improvement of the median relative error ($M$ = -3.1 to -20.9%) with respect to monthly precipitation totals in the Canadian Rockies, compared to raw NARR which underestimated station precipitation ($M$ = -9.5 to -42.6%). However, the median absolute error ($MAD$) of the relative error did not change much, and even increased in some instances, i.e. from 13.5-31.3% for the raw NARR compared with 19-29.5% for LOP (see table 3 in Jarosch et al., 2012). The station-based scaling used in this study resulted in $M$ = 3.8% and $MAD$ = 33%, compared to $M$ = 27% and $MAD$ = 41% for the raw monthly NARR precipitation. Hence the improvement seen is greater than that reported for the station-free LOP model by Jarosch et al (2012) in the Rockies.

-Presentation of the results between fixed and dynamic glacier mass balance results remains unclear throughout the manuscript. Earlier on when discussing the topographic data, it should be mentioned the negligible effect of the conventional glacier simulation and therefore only the reference mass balance simulation results are presented for final glacier mass balance results.

The text explicitly states when the reference vs conventional mass balance is used. We have reviewed the text to add some more clarifications, e.g., in section 4 and in Figure 8 caption to make it clear that long-term simulation of past mass balance are conventional balances, i.e. including changes in glacier area.

Specific (Line by line):

Title: Modelling glacier mass-balance and climate sensitivity in a context of observations: applications to Saskatchewan Glacier, western Canada -> Modelling glacier mass-balance and climate sensitivity in a context of sparse (or limited) observations: applications to Saskatchewan Glacier, western Canada

Yes that was actually the title… not sure why the word 'sparse' was not there. It is there now.

Line 19: was little -> was a little

Changed to 'not very sensitive'

Line 120: (ii) should this objective also include the air humidity and albedo feedback as they are the major conclusions of the paper?

I agree, changed to: 'quantify the respective contributions of energy balance, precipitation phase **and humidity** feedbacks to the mass balance climate sensitivity warming scenarios';

Line 89+120: spare -> sparse

corrected

Figure 1: Reduce the interval of labeled contours. Increase font size on Fig. 1c legend. It is not immediately clear the location of the air temperature points, since the color of the star is overlapped with the snow survey points – change the symbol of the air temperature point or increase the size of the symbol.

Done

Line 195: Why were the precipitation records from the other five permanent weather stations not used?

Because they were the two closest, and highest elevation ones. All 7 stations were used to derive lapse rates, as mentioned in that section. We modified the sentence to emphasize: the choice of stations to use for downscaling:

'As precipitation was not measured at the AWS site, a historical precipitation record was produced using data from the two weather stations closest to Saskatchewan Glacier and highest in elevation ….'

Line 209: State the temporal and spatial resolutions of ERA interim and NCEP reanalyse products.

Modified to: 'ERA interim (6-hourly, ~80 km resolution) and NCEP (6-hourly, ~ 600 km resolution) reanalyses'

Line 337: te -> the

corrected

Line 353: Says depth scales was calibrated with snow depth at AWS but section 3.2.1 does not describe recording snow depth measurements. Although the supplementary material describes snow depth sounding measurements. Clarify where the snow depth measurements are coming from.

Information was added to methods section 3.2.1: Recorded variables include air temperature ($T_a$), relative humidity ($RH$), incoming global ($G$) and reflected ($SW\uparrow$) solar radiation, wind speed ($WS$) and direction ($WD$) and **snow depth using an ultrasonic sensor.**

Line 358: bias correction -> downscaling?

No, we used the word 'downscaling' to describe the spatial interpolation of NARR to the station and the bias correction of the NARR. That section describes the bias correction step.

Line 419: I think this should be from 0 to 7 °C to be consistent with results and abstract.

Yes. Corrected

Line 421: Define GCM at first mention

Done

Figure 5: Are the values correct for relative wind direction on Fig. 5a? If so, why do they vary from the monthly wind directions?

Yes they are correct. Panel A is the mean diurnal cycle averaged over the whole period, panel B are monthly cycle averaged over the same period. The average of the two curve is the same (12 degrees).

Line 546: Mention the ultrasonic snow depth sounder in section 3.2.1

Done

Figure 6: check the figure caption for correct lettering of figure numbers.

Corrected

Figure 6d: Shows the limitations of the precipitation gradient since the gradient derived is not within the same elevational ranges and should be discussed further as per previous comments.

Yes, but Figure 6d also shows that except for 2014 the observations fit the simulated profile. This is extensively discussed in section 5.2

Line 580: Dynamical adjustment explanation should be explained in the methods somewhere between lines 180 and 185.

The description of the calculation was moved to there as suggested

Line 595: The use of lapsed interpolated should be reworded for clarity to 'lapse rate corrected'.

Corrected as suggested

Line 611: 'an even more so ice surface morphology' reword to clarify if you mean that the surface morphology is more uneven than the snow surface or less uneven.

Corrected to: and even more so on rougher ice surface morphology

Line 800: Include that there was difference in elevational ranges used for precipitant gradient compared to the elevation of the Sask. Glacier.
I do not think it is relevant there as we are discussing downscaling performance at the Columbia/Parker Ridge station so that lapse rate extrapolation is not concerned here.

Line 941: The air humidity feedback is one of the main findings from the paper, expand on the implications for glacier mass balance at Sask. Glacier with increasing atmospheric warming from this feedback.

We added this sentence: Under a stable atmospheric moisture regime, increasing atmospheric warming would lead to an increasing humidity feedback on ablation (Table 1).

Supplementary Martial:

- 'Errors in glacier outline delineation were not considered' Please provide justification for why they were not considered.

Changed to: Errors in glacier outline delineation were not considered as they are assumed small compared to other sources.

-Figure S1: interesting to see Lake Louise precipitation data included here. Did you compare the record with Columbia and Park Ridge to show the variation? Include a few sentences to say why the Lake Louise and other precip record was not included in the study.

The station is somewhat far from the glacier and at a comparatively low elevation to be used as model forcing. Figure S1 does just that (the comparison with Columbia/Parker), on a climatological scale (seasonal).

We added this sentence: 'Only the Columbia and Parker Ridge station, located close to the glacier and the highest elevations (2000 m), were used for downscaling NARR precipitations, while the other stations were used to constrain the precipitation lapse rate.'

----------------------------------------------------END OF REVIEW------------------------------------

**Referee #4: Andrew MacDougall,**

Overall evaluation:

The study uses a distributed energy balance melt model forced with reanalysis data to simulate the mass balance of Saskatchewan Glacier. The model is able to reasonably reproduce the mass balance for the years that data is available, and longer-term averages of mass balance derived from geodetics. The paper is well written and has of complete description of the data-set, models, and analysis used. The authors have responded well to the previous set of comments from other reviewers have found no scientific flaws in the analysis. It is good to see progress being made on glacier melt modelling. Overall I recommend that the paper undergo minor revisions.

We thank Dr. MacDougall for his review of our work and previous evaluation reports. We reply to his general and specific comments below.

Specific comments:

Line 19: "was little sensitive" is unclear. Rewrite for clarity.

Changed to: 'Not very sensitive'

Line 29: Seems like there should be a time unit as part of the temperature sensitivity e.g. 'm w.e. ºC a-1'?

Correct, we added the $a^{-1}$

Line 34: Climate warming increases the global total amount of precipitation. So whether a region gets more snow or less snow is complex and is expected to vary byregion.

Correct. We reworded to : reduced precipitation as snowfall in cold regions

Line 89: Should 'spare' be 'sparse'?

Yes, corrected

Figure 1: Legend is panel C is too low-resolution to read. Need a higher resolution (or vector format) version of this figure.

The legend was improved, font sizes were increased

Line 145: -10.1?

Yes, corrected

Line 210: Change 'represent well' to 'well represent'

Change done

Figure 5: Be careful here with colour-blind compliance. The green and the red are probably to close.

We checked and it seems the contrast seems compliant…

Figure 9: Restate the abbreviations in the figure caption.

Done

Figure 12: Two panels are labelled 'a'

Corrected

Line 842: Change 'reached in' to 'reached for'

Corrected